# ADAPTIVE CANONICALIZATION WITH APPLICATION TO INVARIANT ANISOTROPIC GEOMETRIC NETWORKS

**Ya-Wei Eileen Lin**[1,2] **and Ron Levie**[3]
[1] Technical University of Munich, School of Computation, Information and Technology
[2] Munich Center for Machine Learning
[3] Technion - Israel Institute of Technology, Faculty of Mathematics

## ABSTRACT

Canonicalization is a widely used strategy in equivariant machine learning, enforcing symmetry in neural networks by mapping each input to a standard form. Yet, it often introduces discontinuities that can affect stability during training, limit generalization, and complicate universal approximation theorems. In this paper, we address this by introducing *adaptive canonicalization*, a general framework in which the canonicalization depends both on the input and the network. Specifically, we present the adaptive canonicalization based on prior maximization, where the standard form of the input is chosen to maximize the predictive confidence of the network. We prove that this construction yields continuous and symmetry-respecting models that admit universal approximation properties.

We propose two applications of our setting: (i) resolving eigenbasis ambiguities in spectral graph neural networks, and (ii) handling rotational symmetries in point clouds. We empirically validate our methods on molecular and protein classification, as well as point cloud classification tasks. Our adaptive canonicalization outperforms the three other common solutions to equivariant machine learning: data augmentation, standard canonicalization, and equivariant architectures.

## 1 INTRODUCTION

Equivariant machine learning (Gerken et al., 2023; Villar et al., 2021; Han et al., 2022; Keriven & Peyré, 2019) has been accentuated in geometric representation learning (Bronstein et al., 2017), motivated by the need to build models that respect symmetry inherent in data. For example, permutation equivariance in graphs (Gilmer et al., 2017; Zaheer et al., 2017; Xu et al., 2019), translation equivariance in images (LeCun & Bengio, 1998; Cohen & Welling, 2016), and SO(3) or SE(3) equivariance for 3D objects and molecules (Thomas et al., 2018; Fuchs et al., 2020; Batzner et al., 2022; Satorras et al., 2021). The symmetry is built into the method so that transforming the input induces a predictable transformation of the output. This inductive bias reduces sample complexity, curbs overfitting to arbitrary poses, and often improves robustness on distribution shifts where the same object appears in a different orientation or ordering (Kondor & Trivedi, 2018; Wang et al., 2022; Park et al., 2022; Bronstein et al., 2021; Bietti & Mairal, 2019; Kaba & Ravanbakhsh, 2023).

There are three principal approaches to handling symmetry in machine learning. The first involves designing equivariant architectures (Cohen & Welling, 2017; Weiler et al., 2018a; Weiler & Cesa, 2019; Geiger & Smidt, 2022; Maron et al., 2019a; Lippmann et al., 2025): neural network layers are constructed to commute with the symmetry. The second approach is data augmentation, where each datapoint is presented to the model at an arbitrary pose (Chen et al., 2020; Brandstetter et al., 2022b). The third strategy is canonicalization (Kaba et al., 2023; Ma et al., 2023; 2024; Lim et al., 2023b;a; Mondal et al., 2023; Lawrence et al., 2025; Sareen et al., 2025; Luo et al., 2022): each input is mapped to a standard form and then processed by a non-equivariant network. Another common approach to equivariant machine learning is frame averaging (Puny et al., 2022), which averages the network's output over a set of input transformations.

A well-known problem in canonicalization is that in many cases it unavoidably leads to an end-to-end architecture which is discontinuous with respect to the input (Dym et al., 2024; Zhang et al., 2020; Lim et al., 2023b). This inevitably leads to problems in stability during training and in generalization,

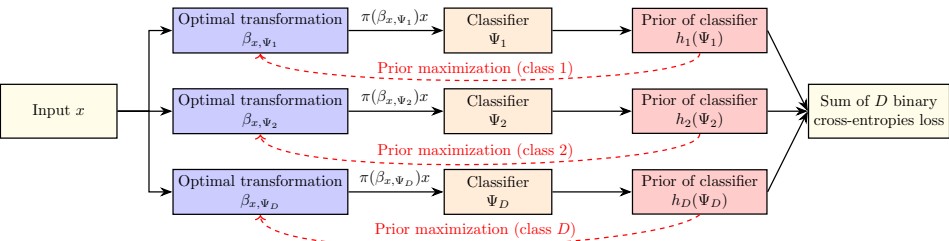

Figure 1: Illustration of prior maximization adaptive canonicalization in classification. The adaptive canonicalization optimizes the transformations $\beta_{x,\Psi_j}$ of the inputs $x$ to the classifiers $\Psi_j$, while, during training, $\Psi_j$ are simultaneously trained w.r.t. the adaptively canonicalized inputs $\pi(\beta_{x,\Psi_j})x$.

as very similar inputs can lead to very different outputs (Dym et al., 2024; Tahmasebi & Jegelka, 2025a;b). Moreover, the discontinuity of the network makes universal approximation properties less natural, as one approximates continuous symmetry preserving functions with discontinuous networks (Dym et al., 2024; Kaba et al., 2023; Wagstaff et al., 2022).

**Our Contribution.** In this paper, we show that the continuity problem in canonicalization can be solved if, instead of canonicalizing only as a function of the input, one defines a canonicalization that depends both on the input and the network. We propose such a general setting, which we call *adaptive canonicalization*, and show that it leads to continuous end-to-end models that respect the symmetries of the data and have universal approximation properties. Our theory does not only lead to superior theoretical properties w.r.t. standard canonicalization, but often also to superior empirical performance, specifically, in molecular, protein, and point cloud classification.

We focus on a specific class of adaptive canonicalizations that we call *prior maximizers*. To explain these methods, we offer the following illustrative example. Suppose that we would like to train a classifier of images into *cats*, *dogs* and *horses*. Suppose as well that each image $x$ can appear in the dataset in any orientation, i.e., as $\pi(\alpha)x$ for any $\alpha \in [0, 2\pi]$ where $\pi(\alpha)$ is rotation by $\alpha$. One standard approach for respecting this symmetry is to design an *equivariant architecture* $\Theta$, which gives the same class probabilities to all rotations of the same image, i.e., $\Theta(\pi(\alpha)x) = \Theta(\pi(\alpha')x)$ for any to angles $\alpha, \alpha'$. Another simple approach for improving the classifier is to train a symmetryless network $\Psi$, and *augment* the training set with random rotations $\pi(\alpha)x$ for each input $x$. Yet another standard approach is to *canonicalize* the input, namely, to rotate each input image $x$ by an angle $\beta_x$ that depends on $x$ in such a way that all rotated versions of the same image would have the exact same standard form, i.e., $\pi(\beta_{\pi(\alpha)x})\pi(\alpha)x = \pi(\beta_{\pi(\alpha')x})\pi(\alpha')x$ for any two angles $\alpha, \alpha'$. Then, the canonicalized image $\pi(\beta_x)x$ is plugged into a standard symmetryless neural network $\Psi$, and the end-to-end architecture $\Psi(\pi(\beta_x)x)$ is guaranteed to be invariant to rotations. We propose a fourth approach, where the canonicalized rotation $\pi(\beta_{x,\Psi})$ depends both on the image $x$ and on the (symmetryless) neural network $\Psi$.

To motivate this approach, consider a neural network $\Psi$ which, by virtue of being symmetriless, may perform better on some orientations of $x$ than others. For illustration, it is easier for humans to detect an image as a horse if its limbs point downwards. Suppose that $\Psi(x) = (\Psi_j(x))_{j=1}^3 = (\Psi_{dog}(x), \Psi_{cat}(x), \Psi_{horse}(x))$ is a sequence of binary classifiers with values in $[0, 1]$ each. The output of $\Psi$ is defined to be the class with highest probability. Suppose moreover that for each $j$, the network $\Psi_j$ is granted the ability to rotate $x$ freely, and probe the output $\Psi_j(\pi(\alpha)x)$ for each $\alpha$. The network then chooses the orientation $\alpha_*$ such that $\Psi_j(\pi(\alpha_*)x)$ is maximized. As an analogy, one can imagine an image on a piece of paper being handed at a random orientation to a person with a visual system $\Psi$. To detect if there is a horse in the image, the person would rotate the paper, searching for an orientation which looks like a horse. Namely, if there is an orientation $\alpha_*$ where $\Psi_{horse}(\pi(\alpha_*)x)$ is high then there is a horse in the image, and otherwise there is none. This process would be repeated for all other classes, and eventually the image would be classified as the $j_*$ such that $\max_\alpha \Psi_{j_*}(\pi(\alpha)x)$ is greater than $\max_\alpha \Psi_j(\pi(\alpha)x)$ for all other $j \neq j_*$. This is the process that we call *prior maximization adaptive canonicalization*. This process is inspired by ideas from cognitive psychology, where the human visual system is believed to learn canonical mental models of objects and to discard redundant variation due to symmetries by mentally "rotating" perceived stimuli into alignment with these canonical views (Shepard & Metzler, 1971; Cooper & Shepard, 1973; Tarr & Pinker, 1989). Our example of a person rotating a sheet of paper to recognize whether it contains a

horse is directly inspired by this line of work: prior maximization adaptive canonicalization can be viewed as a neural analogue of mental rotation, where the network searches over transformations to align inputs with its learned canonical views. We note that some previous models in machine learning were also inspired by this process (Palmer, 1981; Harris et al., 2001; Graf, 2006; Gomez et al., 2008; Konkle & Oliva, 2011; Risko & Gilbert, 2016; Tacchetti et al., 2018; Schmidt & Stober, 2024).

We note that the inputs are adaptively canonicalized also during training, so $\Psi$ needs not learn to respect any symmetry on its own. In fact, $\Psi$ can benefit from being symmetryless. For example, it may search for "horse head" patterns only diagonally above where it detects "horse limbs" patterns, and rely on the prior maximization to orient horses accordingly.

We show that adaptive canonicalization leads to a continuous symmetry preserving end-to-end classifier that can approximate any symmetry preserving continuous function when $\Psi$ are non-equivariant neural networks. As an application, we propose adaptive canonicalization methods for 1) spectral graph neural networks, where the symmetry is in the choice of the eigenbasis of the graph shift operator, and 2) point clouds, with rotation symmetries. We show that adaptive canonicalization in these cases outperforms both standard canonicalization and equivariant networks, as well as augmentation methods. See Figure 1 for an illustration of prior maximization.

## 2 RELATED WORK

Canonicalization has been studied in several forms. For example, in computer vision and geometric deep learning methods, inputs are often first transformed into a standardized pose or reference frame before classification (Lowe, 2004; Jaderberg et al., 2015). More recent work formalizes this as an explicit canonicalization map feeding a downstream network (Lim et al., 2023b; Ma et al., 2024) or as energy-based canonicalization (Kaba et al., 2023) in which one learns an energy over group elements and takes the minimizer as the canonical transformation. The latter has been further developed on symmetries defined by general Lie group actions (Shumaylov et al., 2025). Canonicalization has also been used for data alignment (Mondal et al., 2023; Schmidt & Stober, 2025) and for test-time optimization over transformations, where one searches over group actions to select a canonical representation before downstream inference (Singhal et al., 2025; Schmidt & Stober, 2024). A related line of work is frame averaging (Puny et al., 2020), which averages a network's output over a set of group transformations, and its extension to weighted frame averaging (Dym et al., 2024), where each datapoint is equipped with a probability distribution over the group and averaging is performed with respect to this measure, yielding continuity guarantees. In our work, we instead study canonicalization as a function of both the input and the network, and we establish continuity guarantees for symmetry-preserving continuous functions realized by our construction. Moreover, our approach is not restricted to symmetries defined via group actions, and allows working with more general augmentations for transforming datapoints. We refer to Appendix B for further discussion and comparison with extended related work.

## 3 ADAPTIVE CANONICALIZATION

In this section, we develop the general theory of adaptive canonicalization, and prove that it leads to continuous symmetry preserving networks with universal approximation properties.

### 3.1 BASIC DEFINITIONS AND BACKGROUND

The function that maps each input $x$ to the output $f(x)$ is denoted by $x \mapsto f(x)$. The free variable of a univariate function is denoted by $(\cdot)$, and by $(\cdot, \cdot\cdot)$ for a function of two variables. For example, the function $(x, y) \mapsto \sin(x)\exp(y)$ is also denoted by $\sin(\cdot)\exp(\cdot\cdot)$. We denote the infinity norm of $x = (x_d)_{d=1}^D \in \mathbb{R}^D$ by $|x| := \max_{1 \le d \le D} |x_d|$. We define the infinity norm of a continuous function $f : \mathcal{K} \to \mathbb{R}^D$ over a topological space $\mathcal{K}$ by $\|f\|_\infty = \sup_{x \in \mathcal{K}} |f(x)|$. If $\|f - y\|_\infty < \epsilon$ we say that $y$ approximates $f$ uniformly up to error $\epsilon$. The set of all subsets of a set $\mathcal{K}$, i.e., the *power set*, is denoted by $2^{\mathcal{K}}$. When defining general metric spaces, we allow the distance between points to be $\infty$. This does not affect most of the common properties of metric spaces (see Burago et al. (2001)).

**Function Spaces.** We develop the definitions of adaptive canonicalization in general locally compact Hausdorff spaces. Two important examples of such a space is a compact metric space or $\mathbb{R}^J$.

**Definition 1.** *Let $\mathcal{K}$ be a locally compact Hausdorff space, and $D \in \mathbb{N}$.*

- *A function $f : \mathcal{K} \to \mathbb{R}^D$ is said to vanish at infinity if for every $\epsilon > 0$ there exists a compact set $K \subset \mathcal{K}$ such that $|f(x)| < \epsilon$ for every $x \in \mathcal{K} \setminus K$.*
- *The space of all continuous functions $f : \mathcal{K} \to \mathbb{R}^D$ that vanish at infinity, with the supremum norm $\|f\|_\infty = \max_{x \in \mathcal{K}} |f(x)|$ is denoted by $C_0(\mathcal{K}, \mathbb{R}^D)$.*

In adaptive canonicalization, we consider families of continuous functions where the $\epsilon - \delta$ formulation of continuity is uniform over the whole family, as defined next.

**Definition 2.** *Let $\mathcal{X}$ and $\mathcal{Y}$ be two metric spaces with metrics $d_\mathcal{X}$ and $d_\mathcal{Y}$ respectively. A family $\mathcal{F}$ of function $f : \mathcal{X} \to \mathcal{Y}$ is called* equicontinuous *if for every $x \in \mathcal{X}$ and every $\epsilon > 0$, there exists $\delta > 0$ such that every $z \in \mathcal{X}$ which satisfies $d_\mathcal{X}(x, z) < \delta$ also satisfies*

$$\forall f \in \mathcal{F} : \quad d_\mathcal{Y}(f(x), f(z)) < \epsilon.$$

**Universal Approximation.** *Universal approximation theorems (UAT)* state that any continuous function over some topological space can be approximated by a neural network. In such a case, the neural networks are said to be *universal approximators*, as defined next.

**Definition 3.** *Let $\mathcal{K}$ be a locally compact Hausdorff space and $D \in \mathbb{N}$. A set of continuous functions $\mathcal{N}(\mathcal{K}, \mathbb{R}^D) \subset C_0(\mathcal{K}, \mathbb{R}^D)$ is said to be a* universal approximator *of $C_0(\mathcal{K}, \mathbb{R}^D)$ if for every $f \in C_0(\mathcal{K}, \mathbb{R}^D)$ and $\epsilon > 0$ there is a function $\theta \in \mathcal{N}(\mathcal{K}, \mathbb{R}^D)$ such that*

$$\forall x \in \mathcal{K} : \quad |f(x) - \theta(x)| < \epsilon.$$

In the above definition, we interpret $\mathcal{N}(\mathcal{K}, \mathbb{R}^D)$ as a space of neural networks. A UAT is hence any theorem which shows that some set of neural networks is a universal approximator. Two examples of UATs are: 1) multilayer perceptrons (MLP) are universal approximators of $C_0(\mathcal{K}, \mathbb{R}^D)$ for compact subset $\mathcal{K}$ of the Euclidean space $\mathbb{R}^D$ (Hornik et al., 1989; Cybenko, 1989), and 2) DeepSets (Zaheer et al., 2017) are universal approximators of continuous functions from multi-sets to $\mathbb{R}^d$. See Appendix D for more details.

## 3.2 ADAPTIVE CANONICALIZATION

In the general setting of adaptive canonicalization, we have a domain of inputs $\mathcal{G}$ which need not have any structure apart for being a set, e.g., the set of graphs. We consider continuous functions $f : \mathcal{K} \to \mathbb{R}^L$ over a "nice" domain $\mathcal{K}$, e.g., $\mathcal{K} = \mathbb{R}^J$. Such functions can be approximated by neural networks. We then pull-back $f$ to be a function from $\mathcal{G}$ to $\mathbb{R}^L$ using a mapping $\rho_f : \mathcal{G} \mapsto \mathcal{K}$ that depends on (is adapted to) $f$. Namely, we consider $f(\rho_f(\cdot)) : \mathcal{G} \to \mathbb{R}^D$. The following definitions assure that such a setting leads to functions with nice properties, as we show in subsequent sections.

**Definition 4.** *Let $\mathcal{K}$ be a locally compact Hausdorff space, $\mathcal{G}$ be a set, and $D \in \mathbb{N}$. A mapping $\rho = \rho_{(\cdot)}(\cdot\cdot) : C_0(\mathcal{K}, \mathbb{R}^D) \times \mathcal{G} \to \mathcal{K}, (f, g) \mapsto \rho_f(g)$, is called an* adaptive canonicalization *if the set of functions*

$$\{f \mapsto f \circ \rho_f(g) \mid g \in \mathcal{G}\}$$

*is equicontinuous (as functions $C_0(\mathcal{K}, \mathbb{R}) \to \mathbb{R}^D$). Here, $f \circ \rho_f(g) := f(\rho_f(g))$.*

Next, we define the function space that we would like to approximate using adaptive canonicalization.

**Definition 5.** *Let $\mathcal{K}$ be a locally compact Hausdorff space, $\mathcal{G}$ be a set, and $D \in \mathbb{N}$. Let $\rho$ be an adaptive canonicalization, and let $f \in C_0(\mathcal{K}, \mathbb{R}^D)$. The function*

$$f \circ \rho_f : \mathcal{G} \to \mathbb{R}^D, \quad g \mapsto f\big(\rho_f(g)\big)$$

*is called an* adaptive canonicalized continuous function, *or a* canonicalized function *in short.*

In Section 3.4, we show that for an important class of adaptive canonicalizations the set of adaptive canonicalized continuous functions is exactly the set of all symmetry preserving continuous functions.

It is now direct to prove the following universal approximation theorem.

**Theorem 6** (Universal approximation of adaptive canonicalized functions)**.** *Let $\mathcal{N}(\mathcal{K}, \mathbb{R}^D)$ be a universal approximator of $C_0(\mathcal{K}, \mathbb{R}^D)$, and $f \circ \rho_f$ an adaptive canonicalized continuous function. Then, for every $\epsilon > 0$, there exists a network $\theta \in \mathcal{N}(\mathcal{K}, \mathbb{R}^D)$ such that for every $g \in \mathcal{G}$*

$$|f \circ \rho_f(g) - \theta \circ \rho_\theta(g)| < \epsilon.$$

*Proof.* Let $\epsilon > 0$. By Definition 4, there exists $\delta > 0$ such that

$$\forall y \in C_0(\mathcal{K}, \mathbb{R}^D) : \|f - y\|_\infty < \delta \Rightarrow \left(\forall g \in \mathcal{G} : |f \circ \rho_f(g) - y \circ \rho_y(g)| < \epsilon\right). \quad (1)$$

By the universal approximation property, there exists a network $\theta$ such that $\|f - \theta\|_\infty < \delta$. Hence, by (1), $\quad \forall g \in \mathcal{G} : |f \circ \rho_f(g) - \theta \circ \rho_\theta(g)| < \epsilon$. $\qquad\square$

## 3.3 PRIOR MAXIMIZATION ADAPTIVE CANONICALIZATION

Prior maximization is a special case of adaptive canonicalization, where $\rho_f$ is chosen to maximize some prior on the output of $f$. The maximization is done over a space of transformations $\kappa_u : \mathcal{G} \to \mathcal{K}$ parameteried by $u$, i.e., maximizing the prior of $f(\kappa_u(g))$ w.r.t. $u$.

Let $\mathcal{U}$ be metric space, and for every $u \in \mathcal{U}$, let

$$\kappa_{(\cdot)}(\cdot\cdot) : \mathcal{U} \times \mathcal{G} \to \mathcal{K}, \quad (u, g) \mapsto \kappa_u(g) \in \mathcal{K}.$$

Suppose that $\kappa_u(g)$ is continuous in $u$ for every $g \in \mathcal{G}$. We call $\kappa$ a *transformation family*, transforming objects in $\mathcal{G}$ into points in $\mathcal{K}$, where different $u \in \mathcal{U}$ define different transformations. Let $H = (h_1, \ldots, h_D)$, where $h_d : \mathbb{R} \to \mathbb{R}$ for each $d$, be a sequence of continuous monotonic functions, that we call the ensemble of *priors*. We call each $h_d$ a *prior*. We denote $H \circ f := (h_d \circ f_d)_{d=1}^D$.

For every $f = (f_1, \ldots, f_D) \in C_0(\mathcal{K}, \mathbb{R}^D)$, $g \in \mathcal{G}$ and $d$, assume that $h_d \circ f_d(\kappa_{(\cdot)}(g))$ attains a maximum in $\mathcal{U}$. This is the case for example when $\mathcal{U}$ is compact. Define

$$\rho_f(g) = \left(\rho_{f_d}^d(g)\right)_{d=1}^D := \left(\left\{\kappa_{u_*}(g) \mid h_d \circ f_d\left(\kappa_{u_*}(g)\right) = \max_{u \in \mathcal{U}} h_d \circ f_d\left(\kappa_u(g)\right)\right\}\right)_{d=1}^D \in \left(2^\mathcal{K}\right)^D. \quad (2)$$

Note that $\rho_f : \mathcal{G} \to (2^\mathcal{K})^D$. By abuse of notation, we also denote by $\rho_f$ the mapping that returns some arbitrary sequence of points $(x_d \in \rho_{f_d}^d(g))_{d=1}^D \in \mathcal{K}^D$ for each $g \in \mathcal{G}$. The choice of the specific point in $\rho_{f_d}^d(g)$ does not affect the analysis. We interpret $\rho_f$ as a function that takes an input $g$ and canonicalizes it separately with respect to each output channel $f_d$, adaptively to $f_d$.

When used for classification, we interpret each output channel $f_d \circ \rho_{f_d}^d(g) \in [0, 1]$ as a binary classifier, i.e., representing the probability of $g$ being in class $d$ vs. not being in class $d$. This multiclass classification setting is called *one vs. rest*, where a standard loss is a sum of $D$ binary cross-entropies (Rifkin & Klautau, 2004; Galar et al., 2011; Allwein et al., 2000).

**Definition 7.** *Consider the above setting. The mapping $\rho$ defined by (2) is called* prior maximization. *If in addition $\mathcal{G}$ has a metric such that for every $f \in C_0(\mathcal{K}, \mathbb{R}^D)$ the family $\{g \mapsto f(\kappa_u(g))\}_{u \in \mathcal{U}}$ is equicontinuous, $\rho$ is called* continuous prior maximization.

In Theorem 8 we show that prior maximization is indeed adaptive canonicalization.

For example, let $\mathcal{U} = \mathcal{SO}(3)$ be the space of 3D rotations, and $\mathcal{G} = \mathcal{K} = \mathcal{B}^N$ the set of sequences of $N$ points in the 3D unit ball $\mathcal{B}$, i.e., the space of point clouds. We consider the rotation $g \mapsto \kappa_u(g)$ of the point cloud $g$ by $u \in \mathcal{U}$. Since $\mathcal{G}$ and $\mathcal{U}$ are compact metric spaces, and $(g, u) \mapsto \kappa_u(g)$ is continuous, $\kappa$ and $f$ must be uniformly continuous. Hence, $\{g \mapsto f(\kappa_u(g))\}_{u \in \mathcal{U}}$ is equicontinuous. In fact, whenever $\mathcal{G}$ is compact and $\kappa$ continuous w.r.t. $(u, g)$, it is automatically also uniformly continuous, so $\rho$ is a continuous prior maximization. See Appendix F for additional examples of continuous prior maximization.

**Properties of Prior Maximization.**

**Theorem 8.** *In prior maximization, each $\rho^d : C_0(\mathcal{K}, \mathbb{R}) \times \mathcal{G} \to \mathcal{K}$ is adaptive canonicalization.*

*Proof.* Consider without loss of generality the case where the output dimension is $D = 1$. Since the specific values of $H = h_1$ do not matter, only if it is ascending or descending, without loss of generality suppose $H(x) = x$, in which case prior maximization maximizes directly the output of $f \circ \kappa_u(g)$ with respect to $u$. Consider an arbitrary maximizer $\rho_f(g) \in \arg\max_{u \in \mathcal{U}} f(\kappa_u(g))$ for each $f \in C_0(\mathcal{K}, \mathbb{R})$. The choice of the maximizer does not affect the analysis. Now, if $f, y \in C_0(\mathcal{K}, \mathbb{R})$ satisfy $\|f - y\|_\infty < \epsilon$, then also for every $u \in \mathcal{U}$, $|f(\kappa_u(g)) - y(\kappa_u(g))| < \epsilon$. Let $u_0 \in \mathcal{U}$ be a maximizer of $f(\kappa_u(g))$. We have $y(\kappa_{u_0}(g)) > f(\kappa_{u_0}(g)) - \epsilon$, so

$$\max_u y(\kappa_u(g)) > \max_u f(\kappa_u(g)) - \epsilon.$$

Similarly, we have $\max_u f(\kappa_u(g)) > \max_u y(\kappa_u(g)) - \epsilon$. Together,

$$\left| \max_{u \in \mathcal{U}} f \circ \kappa_u(g) - \max_{u \in \mathcal{U}} y \circ \kappa_u(g) \right| < \epsilon. \tag{3}$$

Hence, $f \mapsto f \circ \rho_f(g)$ is Lipschitz continuous with Lipschitz constant 1 for every $g \in \mathcal{G}$, and therefore equicontinuous over the parameter $g \in \mathcal{G}$. □

This immediately gives a universal approximation theorem for prior maximization as a corollary of Theorem 6. Moreover, we can show that continuous prior maximization gives functions continuous in $\mathcal{G}$. This is one of the main distinctions between prior maximization and standard canonicalization.

**Theorem 9.** *Consider a continuous prior maximization $\rho$ (Definition 7). Then, $f \circ \rho_f$ is continuous.*

*Proof.* Let $\epsilon > 0$. For every $g \in \mathcal{G}$ there is $\delta = \delta_{\epsilon,g} > 0$ such that for every $g' \in \mathcal{G}$ with $d(g, g') < \delta$ and every $u \in \mathcal{U}$ we have $|f(\kappa_u(g)) - f(\kappa_u(g'))| < \epsilon$. Now, by the same argument as in (3),

$$\left| \max_u f(\kappa_u(g)) - \max_u f(\kappa_u(g')) \right| < \epsilon.$$

□

### 3.4 SYMMETRY PRESERVING PRIOR MAXIMIZATION

Consider the following additional assumptions on the construction of continuous prior maximization. Suppose that the space $\mathcal{G}$ is a disjoint union of metric spaces $\mathcal{G}_j$ with finite distances, i.e., $\mathcal{G} = \cup_j \mathcal{G}_j$. Here, $j$ may run on a finite or infinite index set. We define the metric $d$ in $\mathcal{G}$ as follows. For $g_j \in \mathcal{G}_j$ and $g_i \in \mathcal{G}_i$, $d(g_j, g_i) = \infty$ if $j \neq i$ and $d(g_j, g_i) = d_j(g_j, g_i) < \infty$ if $i = j$, where $d_j$ is the metric in $\mathcal{G}_j$. In the theory of metric spaces, the spaces $\mathcal{G}_j$ are called *galaxies* of $\mathcal{G}$. This construction is useful for data which does not have a uniform notion of dimension, e.g., graphs. For example, each galaxy in this case can be the space of adjacency matrices of a fixed dimension with a standard matrix distance.

For each $j$, let $\mathcal{U}_j$ be a group acting continuously on $\mathcal{G}_j$ by $\pi_j(u_j)g_j$. Namely, $\pi_j(u_j) : \mathcal{G}_j \to \mathcal{G}_j$ is continuous for every $u_j \in \mathcal{U}_j$, and for every $u'_j \in \mathcal{U}_j$ and $g_j \in \mathcal{G}_j$ we have $\pi_j(u'_j)\pi_j(u_j)g_j = \pi_j(u'_j u_j)g_j$ and $\pi_j(e_j)g_j = g_j$, where $e_j$ is the identity of $\mathcal{G}_j$. Define $\mathcal{U} = \cup_j \mathcal{U}_j$. Namely, $\mathcal{U}$ is the metric space with galaxies $\mathcal{U}_j$ similarly to the construction of $\mathcal{G}$. Let $\pi$ be a mapping that we formally call an action of $\mathcal{U}$ on $\mathcal{G}$, defined for $u = u_i \in \mathcal{U}_i \subset \mathcal{U}$ and $g = g_j \in \mathcal{G}_j \subset \mathcal{G}$ by $\pi(u)g = \pi_i(u_i)(g_j)$ if $i = j$ and $\pi(u)g = g$ if $i \neq j$.

**Definition 10.** *Consider the above setting and a continuous prior maximization $\rho$. Let $P : \mathcal{G} \to \mathcal{K}$ be continuous, and suppose that the transformation family $\kappa$ is of the form $\kappa_u = P \circ \pi(u)$. We call $\kappa$ a* symmetry preserving transformation family, *and $\rho$ a* symmetry preserving prior maximization.

Note that whenever the spaces $\mathcal{U}_j$ are compact, the functions $u \mapsto h_d \circ f_d(\kappa_u(g))$ of (2) are guaranteed to attain a maximum, even though the space $\mathcal{U} = \cup_j \mathcal{U}_j$ is in general not compact. Hence, the above setting with compact $\mathcal{U}_j$ is an example of prior maximization. More generally, if the restriction of the setting to $\mathcal{U}_j$ and $\mathcal{G}_j$ is (continuous) prior maximization for a single $j$, then the setting for $\mathcal{U}$ and $\mathcal{G}$ is also a (continuous) prior maximization.

**Definition 11.** *We call a function $Q : \mathcal{G} \to \mathbb{R}^D$ continuous symmetry preserving if there exists $F : \mathcal{K} \to \mathbb{R}^D$ in $C_0(\mathcal{K}; \mathbb{R}^D)$ such that for all $u \in \mathcal{U}$ and $g \in \mathcal{G}$, $\quad Q(g) = F(P(\pi(u)(g)))$.*

When $\mathcal{K} = \mathcal{G} = \mathcal{G}_1$ and $P$ is the identity, a symmetry preserving continuous function is a continuous function which is invariant to the action of $u$, i.e., the classical case in equivariant machine learning.

**Properties of Symmetry Preserving Prior Maximization.** We already know by Theorem 9 that $f \circ \rho_f$ is a continuous function when $\rho$ is a symmetry preserving prior maximization. We next show that the set of functions of the form $f \circ \rho_f$ exhaust the space of all continuous symmetry preserving functions.

**Theorem 12.** *Let $\rho$ be a symmetry preserving adaptive canonicalization. Then,*

1. *Any continuous symmetry preserving function can be written as $f \circ \rho_f$ for some $f \in C_0(\mathcal{K}, \mathbb{R}^D)$.*
2. *For any $f \in C_0(\mathcal{K}; \mathbb{R}^D)$, the function $f \circ \rho_f : \mathcal{G} \to \mathbb{R}^D$ is continuous symmetry preserving.*

*Proof.* Without loss of generality, consider the case $D = 1$ and $H(x) = x$.

1. For $Q(g) = F(P(\pi(u)(g)))$, take $f = F$. Then, by definition of symmetry preservation, for every $u \in \mathcal{U}$: $\quad f \circ \rho_f(g) = \max_v F(P \circ \pi(v)g) = F(P \circ \pi(u)g) = Q(g)$.
2. For any $u = u_i \in \mathcal{U}_i$, since $\pi_i(u_i)$ is an action, for any $g = g_j \in \mathcal{G}_j$,

$$f \circ \rho_f(\pi(u)g) = \max_{v_j \in \mathcal{U}_j} f(P \circ \pi_j(v_j)\pi_i(u_i)g_j) = \max_{v_j \in \mathcal{U}_j} f(P \circ \pi_j(v_j)g_j) = f \circ \rho_f(g).$$

$\square$

This leads to the following UAT.

**Theorem 13.** *Consider a symmetry preserving prior maximization and let $\mathcal{N}(\mathcal{K}, \mathbb{R}^D)$ be a universal approximator of $C_0(\mathcal{K}, \mathbb{R}^D)$. Then, any continuous symmetry preserving function can be approximated uniformly by $\theta \circ \rho_\theta$ for some network $\theta \in \mathcal{N}(\mathcal{K}, \mathbb{R}^D)$.*

# 4 APPLICATION OF ADAPTIVE CANONICALIZATION TO ANISOTROPIC GEOMETRIC NETWORKS

In this section, we propose two architectures based on adaptive canonicalization which can be interpreted as anisotropic. First, a spectral graph neural network (GNN) which is sensitive to directionality within eigenspaces. Then, a 3D point cloud network which is sensitive to 3D directions. In Appendix A, we give a detailed tutorial for our proposed architectures. Below, we give a concise derivation.

**Basic Notations for Graphs and Vectors.** We denote by $\mathbb{N}_0$ the set of nonnegative integers. We denote matrices and 2D arrays by boldface capital letters, e.g., $\mathbf{B} \in \mathbb{R}^{N \times T}$. We denote $[N] = \{1, \dots, N\}$ for $N \in \mathbb{N}$. We denote by $\mathbf{B}(:, j)$ and $\mathbf{B}(j, :)$ the $j$'th column and row of the matrix $\mathbf{B}$ respectively. A graph is denoted by $G = ([N], \mathbf{A}, \mathbf{S})$, where $[N]$ is the set of $N$ vertices, $\mathbf{A} \in \mathbb{R}^{N \times N}$ is the adjacency matrix, and $\mathbf{S} \in \mathbb{R}^{N \times T}$ is an array representing the signal. Namely, $\mathbf{S}(n, :) \in \mathbb{R}^T$ is the feature at node $n$. A graph shift operator (GSO) is a self-adjoint operator that respects in some sense the graph's connectivity, e.g., a graph Laplacian or the adjacency matrix.

## 4.1 ANISOTROPIC NONLINEAR SPECTRAL FILTERS

Consider graphs with a self-adjoint GSO $\mathcal{L}$ (e.g., a graph Laplacian or the adjacency matrix) and $T$-channel signals (over the nodes). The number of nodes $N$ varies between graphs. Consider predefined bands $b_0 < b_1 < \dots < b_B \in \mathbb{R}$, and their indicator functions $P_k := \mathbb{1}_{[b_{k-1}, b_k)} : \mathbb{R} \to \mathbb{R}_+$[1]. For each $k \in [B]$, consider the space of signals $\mathcal{X}_k$ in each band $[b_{k-1}, b_k)$, namely, the range of the orthogonal projection $P_k(\mathcal{L})$. Let $M_k$ be the dimension of $\mathcal{X}_k$. We also call $\mathcal{X}_k$ the $k$'th band. See Appendix C for details on how to plug operators into functions via functional calculus. Consider an orthogonal basis $\mathbf{X}_k = (\mathbf{X}_k(:, j))_j \in \mathbb{R}^{N \times M_k}$ for each band $\mathcal{X}_k$. In this setting, the symmetry is the choice of the orthogonal basis within each band $\mathcal{X}_k$. Namely, for $\mathbf{X}_k \in \mathbb{R}^{N \times M_k}$ and orthogonal matrix $\mathbf{U}_k \in \mathbb{R}^{M_k \times M_k}$, the bases $\mathbf{X}_k$ and $\mathbf{X}_k \mathbf{U}_k$ are treated as different vectorial representations of the same linear space $\mathcal{X}_k$.

In the above setting, the galaxies $\mathcal{G}_j$ are defined as follows. Each galaxy is indexed by $j = (N, \mathbf{M}) := (N; M_1, \dots, M_B)$ where $N \in \mathbb{N}$ is the size of the graph, and $M_k \in \mathbb{N}_0$ for $k \in [B]$ are the dimensions of the bands, satisfying $\sum_k M_k \leq N$. Each $\mathcal{G}_{N,\mathbf{M}}$ is the space of pairs of a signal $\mathbf{S} \in \mathbb{R}^{N \times T}$ and a sequence of $B$ orthogonal matrices of height $N$ and widths $M_1, \dots, M_B$. We denote the matrix with dimension 0 by $\emptyset$. For two signals $\mathbf{S}, \mathbf{Q}$ and orthogonal bases sequences $\mathbf{X}, \mathbf{Y}$, the metric $d((\mathbf{X}, \mathbf{S}), (\mathbf{Y}, \mathbf{Q}))$ is defined to satisfy

$$d(\mathbf{X}, \mathbf{Y})^2 = \|\mathbf{S} - \mathbf{Q}\|_2^2 + \sum_{k=1}^{B} \sum_{j=1}^{M_k} \|\mathbf{X}_k(:, j) - \mathbf{Y}_k(:, j)\|_2^2 = \|\mathbf{S} - \mathbf{Q}\|_2^2 + \|\mathbf{X} - \mathbf{Y}\|_2^2,$$

where for $\mathbf{z} = (z_n)_n$, $\|\mathbf{z}\|_2^2 = \sum_n z_n^2$. By definition, the distance between any two points from two different galaxies is $\infty$.

---

[1] $\mathbb{1}_{[b_{k-1}, b_k)}(x)$ is the function that returns 1 if $x \in [b_{k-1}, b_k)$ and 0 otherwise.

The transformation family $\kappa_{\mathbf{U}}$ apply unitary operators $\mathbf{U}_k \in \mathcal{O}(M_k)$ form the right on each $\mathbf{X}(:,k)$, where $\mathcal{O}(M_k)$ is the group of orthogonal matrices in $\mathbb{R}^{M_k \times M_k}$. Namely, $\mathcal{U}_{N;\mathbf{M}} := \prod_{k=1}^{B} \mathcal{O}(M_k)$. We now define the operator $C$ that computes the spectral coefficients of the signal $\mathbf{S}$ with respect to the spectral basis $\mathbf{X}$. Namely,

$$C(\mathbf{X}, \mathbf{S}) = \left(C_k(\mathbf{X}, \mathbf{S})\right)_{k=1}^{B} := \left(\mathbf{X}(:,k)^{\top}\mathbf{S} \in \mathbb{R}^{M_k \times T}\right)_{k=1}^{B}.$$

In *anisotropic nonlinear spectral filters (A-NLSF)*, we consider a symmetryless neural network $\Psi$ that operates on the space of spectral coefficients of the signal, e.g., a multilayer perceptron (MLP). To define $\Psi$ consistently, over a Euclidean space of fixed dimension, we extend or truncate the sequence of spectral coefficients as follows. Let $J_1, \ldots, J_B$ be a predefined sequence of integers, and denote $J = \sum_k J_k$. We define $P$ as the mapping that takes $(\mathbf{X}, \mathbf{S})$ as input, first compute the spectral coefficients $\left(C_k(\mathbf{X}, \mathbf{S})\right)_{k=1}^{B}$, and then truncates or pads with zeros each $C_k(\mathbf{X}, \mathbf{S})$ to be in $\mathbb{R}^{J_k \times T}$. Namely, $P(\mathbf{X}, \mathbf{S}) = \left(P_k(\mathbf{X}, \mathbf{S}) \in \mathbb{R}^{J_k \times T}\right)_{k=1}^{B} \in \mathbb{R}^{J \times T}$. Here, the matrix $\emptyset$ is padded to the zero matrix $\mathbf{0} \in \mathbb{R}^{J_k \times T}$. Hence, the symmetryless network $\Psi$ maps $\mathbb{R}^{J \times T}$ to $\mathbb{R}^D$. A more detailed derivation of the construction is given in Appendix A.3.

We call this architecture anisotropic for the following reason. Consider for example the grid graph with $N \times N$ vertices. Since spectral filters are based on functional calculus (see Appendix C), they are invariant to graph automorphism, and hence to rotations. This means that spectral filters treat the $x$ and $y$ axes equally, and any filter is isotropic in the spatial domain. On the other hand, our symmetryless network $\Psi$ can operate differently on the $x$ and $y$ axes, and we can implement general directional filters on images with $\Psi$. In the general case, $\Psi$ can operate differently on Fourier modes from the same eigenspace, which we interpret as directions within the eigenspace, while standard GNNs cannot. See Appendix H.2 for more details.

## 4.2 ANISOTROPIC POINT CLOUD NETWORKS

Here, we present a point cloud network which is a combination of an equivariant network with adaptive canonicalization. Namely, we consider a permutation invariant network $\Psi$ like DeepSet (Zaheer et al., 2017) or DGCNN (Wang et al., 2019), and to attain 3D rotation invariance in addition we incorporate prior maximization. Together, the method is invariant both to permutations and rotations. We call this method anisotropic since $\Psi$ does not respect the rotation symmetries, and is hence sensitive to directions in the $x, y, z$ space.

We restrict the analysis to multi-sets of a fixed number of points $N$. Multisets are sets where repetitions of elements are allowed. Here, we formally define a multi-set as an equivalence class of arrays up to permutation. To define this, let $\mathcal{S}_N$ be the symmetric group of $N$ elements, i.e., the group of permutations. Given $s \in \mathcal{S}$ and $\mathbf{X} \in \mathbb{R}^{N \times J}$, let $\rho(s)\mathbf{X}$ be the permutation that changes the order of the rows $\mathbf{X}$ according to $s$. We say that $\mathbf{X} \sim \mathbf{Y}$ if there is $s \in \mathcal{S}_N$ such that $\mathbf{X} = \rho(s)\mathbf{Y}$. The equivalence class $[\mathbf{X}]$ is defined as $\{\mathbf{Y} \in \mathbb{R}^{N \times J} \mid \mathbf{Y} \sim \mathbf{X}\}$, and the space of equivalence classes, also called the *quotient space*, is denoted by $(\mathbb{R}^{N \times J}/\sim) := \{[\mathbf{X}] \mid \mathbf{X} \in \mathbb{R}^{N \times J}\}$. We identify the space $(\mathbb{R}^{N \times J}/\sim)$ with the space of multisets.

In Appendix D.2 we show that the quotient space has a natural metric. We hence take $\mathcal{G} = \mathcal{K}$ consisting of a single galaxy $\mathcal{G}_N = (\mathbb{R}^{N \times J}/\sim)$. We moreover show in Appendix D.2 that any universal approximator of permutation invariant functions in $C_0(\mathbb{R}^{N \times J}, \mathbb{R}^D)$, e.g., DeepSet (Zaheer et al., 2017), canonically gives a universal approximator of general continuous functions in $C_0(\mathbb{R}^{N \times J}/\sim, \mathbb{R}^D)$.

The symmetry in our adaptive canonicalization is 3D rotations $\pi(u) \in \mathbb{R}^{3 \times 3}$, where $u$ is in the rotation group $\mathcal{SO}(3)$. Namely, we consider $J = 3$, and rotate the rows of $\mathbf{X} \in \mathbb{R}^{N \times 3}$ via $\mathbf{X}\pi(u^{-1})$. We take $P$ as the identity. Note that this construction can be easily extended to multisets of arbitrary sizes, by considering the galaxies $\mathcal{G}_N = \mathbb{R}^{N \times 3}/\sim$ for all $N \in \mathbb{N}$, and $\mathcal{U}_N = \mathcal{SO}(3)$ applied from the right on $\mathbb{R}^{N \times 3}$. For details on the theoretical construction see Appendix D.2, and for details on the architecture see Appendix A.4.

Table 1: Classification performance on the grid signal orientation task and graph classification benchmarks from TUDataset. The highest accuracy is in ▉ and the second highest is in ▉.

| | Toy Example | TUDataset | | | | |
|---|---|---|---|---|---|---|
| | | MUTAG | PTC | ENZYMES | PROTEINS | NCI1 |
| MLP | $50.03_{\pm 0.1}$ | $79.31_{\pm 3.5}$ | $63.98_{\pm 2.0}$ | $42.17_{\pm 2.8}$ | $75.08_{\pm 3.4}$ | $77.34_{\pm 1.6}$ |
| GCN | $50.01_{\pm 0.1}$ | $81.63_{\pm 3.1}$ | $60.22_{\pm 1.9}$ | $43.66_{\pm 3.4}$ | $75.17_{\pm 3.7}$ | $76.29_{\pm 1.8}$ |
| GAT | $49.95_{\pm 0.0}$ | $83.17_{\pm 4.4}$ | $62.31_{\pm 1.4}$ | $39.83_{\pm 3.7}$ | $74.72_{\pm 4.1}$ | $74.01_{\pm 4.3}$ |
| GIN | $50.00_{\pm 0.1}$ | $83.29_{\pm 3.6}$ | $63.25_{\pm 2.3}$ | $45.69_{\pm 2.6}$ | $76.02_{\pm 2.9}$ | $79.84_{\pm 1.2}$ |
| ChebNet | $50.12_{\pm 0.1}$ | $82.15_{\pm 1.6}$ | $64.06_{\pm 1.2}$ | $50.42_{\pm 1.4}$ | $74.28_{\pm 0.9}$ | $76.98_{\pm 0.7}$ |
| FA+GIN | $49.99_{\pm 0.1}$ | $84.07_{\pm 2.4}$ | $66.58_{\pm 1.8}$ | $52.64_{\pm 2.2}$ | $79.53_{\pm 2.5}$ | $80.23_{\pm 0.9}$ |
| OAP+GIN | $50.03_{\pm 0.0}$ | $84.95_{\pm 2.0}$ | $67.35_{\pm 1.1}$ | $58.40_{\pm 1.6}$ | $83.41_{\pm 1.4}$ | $80.97_{\pm 1.1}$ |
| NLSF | $50.07_{\pm 0.1}$ | $84.13_{\pm 1.5}$ | $68.17_{\pm 1.0}$ | $65.94_{\pm 1.6}$ | $82.69_{\pm 1.9}$ | $80.51_{\pm 1.2}$ |
| $S^2$GNN | $49.93_{\pm 0.1}$ | $82.70_{\pm 2.1}$ | $67.34_{\pm 1.5}$ | $63.26_{\pm 2.8}$ | $78.52_{\pm 1.9}$ | $75.62_{\pm 2.0}$ |
| A-NLSF | $99.38_{\pm 0.2}$ | $87.94_{\pm 0.9}$ | $73.16_{\pm 1.2}$ | $73.01_{\pm 0.8}$ | $85.47_{\pm 0.6}$ | $82.01_{\pm 0.9}$ |

### 4.3 ADDITIONAL APPLICATIONS OF ADAPTIVE CANONICALIZATION

Adaptive canonicalization can be used to model image truncation augmentations. Here, the "symmetry" corresponds to different crops and prior maximization selects the crop on which the classifier is most confident (see Appendix A.5). Our formulation also accommodates various image settings with rotation symmetries and other image transformations (see Appendix F). Note that our setting also applies to pretrained networks. Finally, to broaden the applicability of our approach beyond classification, we explore using the adaptive canonicalization mechanism for point cloud segmentation (see Appendix H.10).

## 5 EXPERIMENTS

We evaluate the anisotropic nonlinear spectral filters (Section 4.1) on toy problems and graph classification, and test the anisotropic point cloud network (Section 4.2) on point cloud classification. The experimental details, including experimental setups and hyperparameters, are in Appendix G. Additional experiments, e.g., ablation study, are in Appendix H.

**Maximization Method.** We approximate the prior maximization by sampling a finite set of transformations from a probability measure on the transformation space, evaluating the prior for all candidates, and retaining the transformation that yields the largest prior value with the one-vs-rest classification objective. We then refine the selected candidate locally by running a few steps of gradient descent (see Appendix A.2).

### 5.1 EXPERIMENTAL EVALUATION OF ANISOTROPIC NONLINEAR SPECTRAL FILTERS

**Illustrative Toy Problems: Grid Signal Orientation Task.** To showcase the effectiveness of A-NLSF, we study a toy classification task: *grid-split channel orientation*. We consider a square grid on the torus, and each node has two channels. The grid is partitioned vertically into two equal disjoint halves. Channel 1 is nonzero only on the left half, and Channel 2 is only nonzero on the right half. In class 0, both channels are 1-frequency pure harmonic signals along $x$, and in class 1, Channel 1 is 1-frequency along $x$ and Channel 2 is 1-frequency along $y$. The task is to decide if the oscillations at the two channels are in the same orientation. See Appendix G for further details.

We compare A-NLSF with the following baselines: (i) MLP, (ii) GCN (Kipf & Welling, 2017), (iii) GAT (Veličković et al., 2018), (iv) GIN (Xu et al., 2019), (v) ChebNet (Defferrard et al., 2016a), (vi) NLSF (Lin et al., 2024a), and (vii) $S^2$GNN (Geisler et al., 2024). In addition, we test canonicalization baselines by combining GIN with frame averaging (FA) (Puny et al., 2022) and orthogonalized axis projection (OAP) (Ma et al., 2024). Table 1 reports the classification results. We see that competing methods remain at chance level while A-NLSF achieves high accuracy by adaptively resolving ambiguities, showing the advantage of adaptive canonicalization, which can internally break the rotation symmetry and use anisotropic classifiers.

**Graph Classification on TUDataset.** We further evaluate A-NLSF on graph classification benchmarks from TUDataset (Morris et al., 2020): MUTAG, PTC, ENZYMES, PROTEINS, and NCI1,

and follow the experimental setup (Ma et al., 2019; Ying et al., 2018; Zhang et al., 2019b) (see Appendix G). We compare with the same baselines as in the grid signal orientation tasks. Table 1 summarizes the classification performance. Canonicalization baselines generally improve over GIN. Notably, we observe that A-NLSF outperforms competing baselines, suggesting that our method provides more informative representations compared to a fixed, precomputed canonical form or isotropic filters.

**Molecular Classification on OGB Datasets.** To further assess the effectiveness of A-NLSF, we evaluate on large-scale molecular and protein benchmarks from Open Graph Benchmark (OGB) (Hu et al., 2020): ogbg-molhiv, ogbg-molpcba, and ogbg-ppa. We compare with GCN, GIN, GatedGCN (Bresson & Laurent, 2017), PNA (Corso et al., 2020), GraphTrans (Wu et al., 2021), SAT (Chen et al., 2022), GPS (Rampášek et al., 2022), SAN (Kreuzer et al., 2021), and the canonicalization method OAP+GatedGCN. The molecular classification results are reported in Table 2. We see that our method achieves

Table 2: Molecular and protein classification performance on OGB datasets.

| | ogbg-molhiv | ogbg-molpcba | ogbg-ppa |
|---|---|---|---|
| | AUROC ↑ | Avg. Precision ↑ | Accuracy ↑ |
| GCN | $0.7599_{\pm 0.0119}$ | $0.2424_{\pm 0.0034}$ | $0.6857_{\pm 0.0061}$ |
| GIN | $0.7707_{\pm 0.0149}$ | $0.2703_{\pm 0.0023}$ | $0.7037_{\pm 0.0107}$ |
| GatedGCN | $0.7687_{\pm 0.0136}$ | $0.2670_{\pm 0.0020}$ | $0.7531_{\pm 0.0083}$ |
| PNA | $0.7905_{\pm 0.0132}$ | $0.2838_{\pm 0.0035}$ | - |
| GraphTrans | - | $0.2761_{\pm 0.0029}$ | - |
| SAT | - | - | $0.7522_{\pm 0.0056}$ |
| GPS | $0.7880_{\pm 0.0101}$ | $0.2907_{\pm 0.0028}$ | $0.8015_{\pm 0.0033}$ |
| SAN | $0.7785_{\pm 0.2470}$ | $0.2765_{\pm 0.0042}$ | - |
| OAP+GatedGCN | $0.7802_{\pm 0.0128}$ | $0.2783_{\pm 0.0024}$ | $0.7745_{\pm 0.0098}$ |
| A-NLSF | $0.8019_{\pm 0.0152}$ | $0.2968_{\pm 0.0022}$ | $0.8149_{\pm 0.0067}$ |

consistent improvements across these datasets, leading to improved generalization in classification.

## 5.2 EXPERIMENTAL EVALUATION OF ANISOTROPIC POINT CLOUD NETWORKS

To evaluate adaptive canonicalization on point cloud classification, we test ModelNet40 (Wu et al., 2015). The dataset consists of 12,311 shapes from 40 categories, with 9,843 samples for training and 2,468 for testing. We build on two point cloud architectures, PointNet (Qi et al., 2017a) and DGCNN (Wang et al., 2019), and apply adaptive canonicalization into their pipeline, denoted respectively AC-PointNet and AC-DGCNN (see Section 4.2 and Appendix A.4). Following Esteves et al. (2018); Deng et al. (2021), we perform on-the-fly rotation augmentation during training, where the dataset size remains unchanged, and for the test set, each test example is arbitrarily rotated as well. This produces a data distribution where each shape appears in any orientation with the same probability. We compare with PointNet, DGCNN, equivariant networks VN-PointNet and VN-DGCNN from Deng et al. (2021), canonicalization methods CN-PointNet and CN-DGCNN from Kaba et al. (2023), and traditional augmentation baselines where the training set is statically expanded with pre-generated rotated samples, denoted PointNet-Aug and DGCNN-Aug. For further details, see Appendix G. Table 3 shows the classification performance. We observe that our method outperforms the competing baselines and it allows the model to learn both local geometric features and optimal reference alignments for each class.

Table 3: Classification results on ModelNet40. Results of competing methods marked with * are taken from Deng et al. (2021); Luo et al. (2022); Kaba et al. (2023).

| | Accuracy |
|---|---|
| PointNet | 74.7* |
| DGCNN | 88.6* |
| PointNet-Aug | $75.8_{\pm 0.9}$ |
| DGCNN-Aug | $89.0_{\pm 1.0}$ |
| VN-PointNet | 77.2* |
| VN-DGCNN | 90.2* |
| CN-PointNet | $79.7_{\pm 1.3}$ * |
| CN-DGCNN | $90.0_{\pm 1.1}$ * |
| AC-PointNet | $81.1_{\pm 0.7}$ |
| AC-DGCNN | $91.6_{\pm 0.6}$ |

## 6 CONCLUSION

We introduce adaptive canonicalization based on prior maximization, a general framework for equivariant machine learning in which the standard form depends on both the input and the network. We prove that our method is continuous, symmetry preserving, and has universal approximation properties. We demonstrate the applicability of our theory in two main settings: resolving eigenbasis ambiguities in spectral graph neural networks, and handling rotational symmetries in point clouds.

**Limitations and Future Work.** Our framework is naturally suited to classification tasks, and at the current scope of the paper, we did not address regression tasks theoretically. We will extend the adaptive canonicalization to regression in future work. Another limitation of our approach is that prior maximization requires solving $D$ optimizations at runtime for the $D$ classes. In future work, we will reduce this to a single optimization to improve efficiency.

## ETHICS STATEMENT

This work introduces a novel method for handling symmetry for equivariant machine learning, with a focus on theory and with application to spectral graph neural networks and point cloud networks. The experiments are conducted using simulated toy problems and public datasets, and therefore there is no concerns related to privacy, consent, or potential harm to living subjects. As the data employed are technical and free from sensitive or identifiable content, the research does not raise any apparent ethical concerns. Accordingly, no additional ethical approval was required for this study.

## REPRODUCIBILITY STATEMENT

For the theoretical results, we include the main proofs in the paper and present additional analysis and illustrative examples in Appendix D and Appendix F. For the empirical study, the implementation details are reported in Appendix G. Our code is available at `https://github.com/ywelld/_ac`.

## USE OF LARGE LANGUAGE MODELS

Following the ICLR 2026 policy that requires disclosure of use of Large Language Models (LLMs), we state that an LLM was used for editing purposes, such as grammar, spelling, phrasing, and stylistic polish.

## ACKNOWLEDGMENTS

We thank the anonymous reviewers for their insightful feedback. RL was supported by a grant from the United States-Israel Binational Science Foundation (BSF), Jerusalem, Israel, and the United States National Science Foundation (NSF), (NSF-BSF, grant No. 2024660), and by the Israel Science Foundation (ISF grant No. 1937/23). YEL acknowledges funding by the Alexander von Humboldt Foundation and the Munich Center for Machine Learning (MCML).

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

# Appendix

# A TUTORIAL FOR PRACTITIONERS: APPLICATIONS OF ADAPTIVE CANONICALIZATION

In this section, we present the construction details of the application of adaptive canonicalization to anisotropic geometric networks. We start by describing the one vs. rest classifier and our maximization method.

## A.1 ONE VS. REST CLASSIFIERS

In our setting, each output channel $f_d \circ \rho_{f_d}^d(g) \in [0, 1]$ is a binary classifier, i.e., representing the probability of $g$ being in class $d$ vs. not being in class $d$ (Rifkin & Klautau, 2004; Galar et al., 2011; Allwein et al., 2000). The per-class score is obtained by $\hat{y}_d = \sigma(f_d \circ \rho_{f_d}^d(g))$, where $\sigma$ is a sigmoid function. Note that the vector $(\hat{y}_1, \dots, \hat{y}_D)$ is not a probability measure, since each entry $\hat{y}_d$ represents a separate probability, and these probabilities do not sum to one in general. We use binary cross-entropy per class and sum over classes $\sum_{d=1}^{D} (-y_d \log \hat{y}_d - (1 - y_d) \log(1 - \hat{y}_d))$, where $y_{d^*} = 1$ on the true class $d^*$, and $y_d = 0$ for $d \neq d^*$.

## A.2 RANDOM MAXIMIZATION

We estimate the prior maximization by searching over the transformation space (e.g., unitary transformations per spectral band or rotation for point clouds) and selecting the transformation that maximizes the prior on the output of the network (i.e., one vs. rest in classification). Specifically, we consider a probability measure on the space $\mathcal{U}_j$ and draw i.i.d samples $\{u_j^i\}_{i=1}^{K}$ from it. The argmax of $\{h_d \circ f_d(\kappa_{u_j}(g))\}_{u_j \in \mathcal{U}_j}$ of Eq. (2) is estimated as the argmax of

$$\{h_d \circ f_d(\kappa_{u_j^i}(g))\}_{i=1}^{K}.$$

For example, in anisotropic nonlinear spectral filters, we draw a finite pool of candidate transformations from the Haar measure on $\mathcal{U}_j$ (Mezzadri, 2006). For anisotropic point cloud networks, the search over rotations can be implemented with quaternion (Shoemake, 1992) or Rodrigues' formula (Rodrigues, 1840). For each input, we evaluate the prior objective for all candidates in the pool and keep only the maximizing orientation when computing the forward pass and gradients. In this way, the prior maximization is implemented as a randomized search: we sample a set of transformations, apply them in parallel, and pick the one giving the best prior value. Note that this sampling-based maximization is best thought of as one convenient implementation of our framework, and other optimization strategies over $\mathcal{U}_j$ could be implemented and plugged in as long as they approximately solve the same maximization problem.

In Appendix E we analyze the approximation properties of this maximization method. Specifically, we show that the randomized maximization method approximates the true maximum in high probability.

We note that prior maximization can be strengthened by locally refining the argmax sample point $u_j^i$. Specifically, we initialize with random transformations drawn from the probability measure, perform the sampling-based prior maximization to select a candidate, and then run a few steps of gradient descent to further decrease the objective locally. If $\mathcal{U}$ is a manifold embedded in Euclidean space, one can apply the vanilla Euclidean gradient descent and then project the obtained transformation back to $\mathcal{U}$, or apply Riemannian gradient descent (Absil et al., 2008) to stay on $\mathcal{U}$. We summarize the random maximization in Algorithm 1. In our experiments, we use 32 sampled transformations for anisotropic nonlinear spectral filters and use 50 sampled transformations for point clouds.

## A.3 CONSTRUCTION DETAILS FOR ANISOTROPIC NONLINEAR SPECTRAL FILTERS

In spectral methods for graphs, we use eigenvectors as a core component for graph representation learning. However, these eigenvectors are not uniquely defined. For each eigenvector we can flip its sign and obtain another eigenvector of the same eigenvalue. Moreover, when an eigenvalue has multiplicity larger than one, any orthogonal basis of its eigenspace gives a valid set of eigenvectors. This means that the same graph can lead to different eigenvectors depending on the eigensolver or numerical details. Hence, a spectral graph neural network that takes these eigenvectors as an input may give different outputs for the same graph, which affects stability and can reduce performance.

---

**Algorithm 1** Random maximization

**Input:** Input $g$, backbone network $f$, scalar prior $h(x)$, sampler `Sample_U()` for $u \sim P$ sampled from a probability measure over $\mathcal{U}$, number of random samples $K$, gradient descent step `GD_Step`

**function** `Random_Maximization`$(g, f, h, \texttt{Sample\_U}, K, \texttt{GD\_Step})$
$\quad \{u^1, \ldots, u^K\} \leftarrow \texttt{Sample\_U}(K)$
$\quad u^* \leftarrow \underset{\{u_1, \ldots, u_K\}}{\arg\max} \ h \circ f(\kappa_{u^i}(g))$
$\quad u^* \leftarrow \texttt{GD\_Step}(u^*)$

$\quad$ **return** $u^*$

---

Since the model should depend only on the graph and node features, and not on arbitrary choices of eigenvector bases, i.e., it should be invariant to the choice of the eigenbasis, it is important to remove such ambiguities. In this work, we focus on graph classification tasks with $D$ classes and study how to resolve eigenbasis ambiguities using prior maximization adaptive canonicalization.

**Setup and Eigendecomposition.** To study this, we work in the following setup. Consider a graph $G = ([N], \mathbf{A}, \mathbf{S})$, where $[N]$ is the set of $N$ vertices, $\mathbf{A} \in \mathbb{R}^{N \times N}$ is the adjacency matrix, and $\mathbf{S} \in \mathbb{R}^{N \times T}$ is an array of node features (row $n$ is the $T$-dimensional feature at node $n$). We consider the normalized graph Laplacian $\mathcal{L} = \mathbf{I} - \mathbf{D}^{-\frac{1}{2}} \mathbf{A} \mathbf{D}^{-\frac{1}{2}}$ as the graph shift operator (GSO) in our experiments,[2] where $\mathbf{D}$ is the diagonal degree matrix with diagonal entries $(d_i)_i \in \mathbb{R}^N$, where $d_i$ is the degree of node $i$. The eigendecomposition of $\mathcal{L}$ is given by $\mathcal{L} = \mathbf{V}^{(G)} \mathbf{\Lambda}^{(G)} \mathbf{V}^{(G)\top}$, where $\mathbf{V}^{(G)} = (\mathbf{v}_i)_i \in \mathbb{R}^{N \times N}$ is an orthogonal matrix of eigenvectors as the columns (i.e., an eigenbasis) and $\mathbf{\Lambda}^{(G)} = \operatorname{diag}(\lambda_1, \ldots, \lambda_N)$ is the diagonal matrix of eigenvalues, where $0 \le \lambda_1 \le \ldots \le \lambda_N \le 2$.

**Band Partition and Band-Preserving Eigenbasis Ambiguity.** We then group the spectrum into predefined bands with boundaries $b_0 < b_1 < \cdots < b_B$ contained in $[0, 2]$, where $B \in \mathbb{N}$ is the total number of bands. The total band $[b_0, b_B]$ is chosen as a subset of $[0, 2]$ since the spectrum of the normalized Laplacian is guaranteed to lie in this interval[3]. In our implementation, we use a dyadic partitioning scheme. Given a decay rate $0 < r < 1$, we set

$$b_0 = 0, \ b_k = 2r^{B-k} \text{ for } k = 1, \ldots, B-1, \ b_B = 2,$$

and define the $k$-th band as $[b_{k-1}, b_k)$. A larger $B$ and smaller $r$ yields narrower bands and hence finer spectral resolution. For example, taking $B = 5$ and $r = 0.5$ gives five bands: $[0, 0.125), [0.125, 0.25), [0.25, 0.5), [0.5, 1)$ and $[1, 2)$.

We now make explicit the symmetry we want the model to respect. Recall that the eigendecomposition of the normalized Laplacian is $\mathcal{L} = \mathbf{V}^{(G)} \mathbf{\Lambda}^{(G)} \mathbf{V}^{(G)\top}$, where $\mathbf{V}^{(G)} = (\mathbf{v}_i)_i \in \mathbb{R}^{N \times N}$ collects the eigenvectors and $\mathbf{\Lambda}^{(G)} = \operatorname{diag}(\lambda_1, \ldots, \lambda_N)$ the eigenvalues. Given the band boundaries $b_0 < \cdots < b_B$, we define for each band $k$ the index set $I_k(G) := \{i \in [N] : \lambda_i \in [b_{k-1}, b_k)\}$, and let $\mathbf{V}_k^{(G)} \in \mathbb{R}^{N \times M_k(G)}$ be the submatrix of $\mathbf{V}^{(G)}$ whose columns are the eigenvectors $(\mathbf{v}_i)_{i \in I_k(G)}$. Therefore, we can write $\mathbf{V}^{(G)} = [\mathbf{V}_1^{(G)} | \cdots | \mathbf{V}_B^{(G)}]$. For each band we denote the associated Paley-Wiener space by $\mathcal{X}_k(G) := \operatorname{span}\{\mathbf{v}_i : i \in I_k(G)\} = \operatorname{Im}(\mathbf{V}_k^{(G)})$. The ambiguity we want to handle comes from changing the orthonormal basis inside each band. Any other basis of the $k$-th band has the form $\widetilde{\mathbf{V}}_k^{(G)} = \mathbf{V}_k^{(G)} \mathbf{U}_k$, where $\mathbf{U}_k \in \mathbb{R}^{M_k(G) \times M_k(G)}$ is a unitary matrix. Equivalently, we can write a full orthonormal basis of $\mathbb{R}^N$ whose columns are partitioned into subsequences corresponding to the bands, and each subsequence spans the associated *Paley-Wiener space* $\mathcal{X}_k(G)$. We can write this condition in matrix form as: $\mathbb{R}^{N \times N} \ni \widetilde{\mathbf{V}}^{(G)} = \mathbf{V}^{(G)} \mathbf{U}^{(G)}$, where $\mathbf{U}^{(G)} := \operatorname{diag}(\mathbf{U}_1, \ldots, \mathbf{U}_B) \in \mathbb{R}^{N \times N}$, i.e., the a block matrix with diagonal blocks $\mathbf{U}_k$. The set of block-diagonal unitary matrices $\mathcal{U}^{(G)} = \{\mathbf{U}^{(G)} = \operatorname{diag}(\mathbf{U}_1, \ldots, \mathbf{U}_B)\}$ constitutes the band-preserving transformations, i.e., the

---

[2]Note that any other self-adjoint GSO can be chosen in our method.
[3]For other GSOs, different bands can be chosen to match their spectral range, or to cover a subset of the spectrum.

unitary matrices that keep the Paley-Wiener spaces (or bands) invariant. The space of these matrices constitutes the symmetry space in our setting.

**Spectral Coefficients and Fixed-Dimensional Standardization.** The above construction is defined for a single graph, but in our setting we work with a collection of graphs (i.e. a data distribution). The spectral support of the bands, that is, the intervals $[b_{k-1}, b_k)$, is fixed across all graphs. However, for a given graph $G$, the $k$-th band, i.e, the linear span of the eigenvectors whose eigenvalues lie in $[b_{k-1}, b_k)$, is a subspace that depends on $G$. Moreover, different graphs can have different numbers of eigenvalues in the same interval, so the dimension of the $k$-th band is not constant across graphs. We denote by $M_k(G)$ the dimension of the $k$-th band for the graph $G$.

Given a graph signal $\mathbf{S} \in \mathbb{R}^{N \times T}$ the band-wise spectral coefficients are defined to be

$$C_k(\mathbf{V}_k^{(G)}, \mathbf{S}) := \mathbf{V}_k^{(G)\top} \mathbf{S} \in \mathbb{R}^{M_k(G) \times T}$$

for $k = 1, \ldots, B$. Note that applying a unitary transformation $\mathbf{U}_k \in \mathbb{R}^{M_k(G) \times M_k(G)}$ on the Paley-Wiener basis and then computing coefficients is equivalent to computing coefficients in the original basis and multiplying by $\mathbf{U}_k^\top$, i.e.,

$$C_k(\mathbf{V}_k^{(G)} \mathbf{U}_k, \mathbf{S}) = (\mathbf{V}_k^{(G)} \mathbf{U}_k)^\top \mathbf{S} = \mathbf{U}_k^\top (\mathbf{V}_k^{(G)\top} \mathbf{S}) = \mathbf{U}_k^\top C_k(\mathbf{V}_k^{(G)}, \mathbf{S}).$$

In order to apply a task network given by a standard neural network architecture, such as an MLP, we require that all inputs lie in a common vector space of a fixed dimension, independent of the particular graph $G$. However, the band dimensions $M_k(G)$ vary across graphs. We therefore introduce a band-wise padding-or-truncating operator. For each band $k$, we predefine a target size $J_k$, which is a hyperparameter. For a general coefficient matrix $\mathbf{C} \in \mathbb{R}^{M \times T}$, we define $\bar{M}_k = \min\{M, J_k\}$ and let $\mathbf{C}_k = \mathbf{C}(1 : \bar{M}_k, :) \in \mathbb{R}^{\bar{M} \times T}$ denote the truncation to the first $\bar{M}_k$ rows of $\mathbf{C}$. Then, the padding-or-truncating operator is defined as $P_k(\mathbf{C}) := \mathbf{C}_k$ if $\bar{M}_k = J_k$, and otherwise

$$P_k(\mathbf{C}) := \begin{bmatrix} \mathbf{C}_k \\ \mathbf{0}_{(J_k - \bar{M}_k) \times T} \end{bmatrix} \in \mathbb{R}^{J_k \times T}.$$

Thus, $P_k$ pads with zeros when $M < J_k$ and truncates when $M > J_k$. Applying this to each band and stacking, we obtain the fixed-dimensional spectral representation

$$P(\mathbf{V}_1^{(G)}, \cdots, \mathbf{V}_B^{(G)}, \mathbf{S}) := \begin{bmatrix} P_1(C_1(\mathbf{V}_1^{(G)}, \mathbf{S})) \\ \vdots \\ P_B(C_B(\mathbf{V}_B^{(G)}, \mathbf{S})) \end{bmatrix} \in \mathbb{R}^{J \times T},$$

where $J = \sum_{k=1}^{B} J_k$. Then, it can be fed to a task network $\Psi : \mathbb{R}^{J \times T} \to \mathbb{R}^D$ across all graphs.

**Anisotropic Nonlinear Spectral Filters via Adaptive Canonicalization.** We now resolve the band-wise eigenbasis ambiguity by prior maximization adaptive canonicalization. Consider the priors $h_d(x) = x$. Let $\Psi : \mathbb{R}^{J \times T} \to \mathbb{R}^D$ be a task network and we write $\Psi(\cdot) = (\Psi_d(\cdot))_{d=1}^D$. For each class $d \in \{1, \ldots, D\}$ and each graph-signal pair $(G, \mathbf{S})$, we select band-wise unitary transforms $\mathbf{U}_1^{(d)}, \ldots, \mathbf{U}_B^{(d)}$ (with $\mathbf{U}_k^{(d)} \in \mathbb{R}^{M_k(G) \times M_k(G)}$) by prior maximization:

$$\{\mathbf{U}_1^{\square(d)}, \ldots, \mathbf{U}_B^{\square(d)}\} = \underset{\mathbf{U}_1^{(d)}, \ldots, \mathbf{U}_B^{(d)}}{\arg\max} \; h_d \left( \Psi_d \left( \begin{bmatrix} P_1(C_1(\mathbf{V}_1^{(G)} \mathbf{U}_1^{(d)}, \mathbf{S})) \\ \vdots \\ P_B(C_B(\mathbf{V}_B^{(G)} \mathbf{U}_B^{(d)}, \mathbf{S})) \end{bmatrix} \right) \right).$$

This is implemented more efficiently by

$$\{\mathbf{U}_1^{\square(d)}, \ldots, \mathbf{U}_B^{\square(d)}\} = \underset{\mathbf{U}_1^{(d)}, \ldots, \mathbf{U}_B^{(d)}}{\arg\max} \; h_d \left( \Psi_d \left( \begin{bmatrix} P_1(C_1(\mathbf{V}_1^{(G)}, \mathbf{S}) \mathbf{U}_1^{(d)}) \\ \vdots \\ P_B(C_B(\mathbf{V}_B^{(G)}, \mathbf{S}) \mathbf{U}_B^{(d)}) \end{bmatrix} \right) \right).$$

The second implementation is more efficient, as typically $N \gg J_k$, and the projections $C_k(\mathbf{V}_k^{(G)}, \mathbf{S}))$ are only computed once before applying the optimization algorithm. Once the maximizing transforms

are obtained, the class-$d$ logit is

$$s_d = \Psi_d \left( \begin{bmatrix} P_1(C_1(\mathbf{V}_1^{(G)}, \mathbf{S}) \mathbf{U}_1^{\square(d)}) \\ \vdots \\ P_B(C_B(\mathbf{V}_B^{(G)}, \mathbf{S}) \mathbf{U}_B^{\square(d)}) \end{bmatrix} \right),$$

and we apply a sigmoid to obtain a per-class probability. The training loss is the sum of $D$ binary cross-entropies (one-vs-rest).

This construction is seen as anisotropic because $\Psi_d$ is symmetryless, and can operate differently along different directions within the same eigenspace.

## A.4 CONSTRUCTION DETAILS FOR ANISOTROPIC POINT CLOUD NETWORKS

We next apply adaptive canonicalization in the spatial domain, where inputs are point clouds in $\mathbb{R}^3$ represented by their $(x, y, z)$ coordinates. We focus on point-cloud classification with multiset neural networks.

**Multiset Neural Networks and Orientation Sensitivity.** A point cloud is represented as $\mathbf{X} \in \mathbb{R}^{N \times 3}$ whose $N$ rows are points. By a multiset neural network we mean a mapping $\Psi : \mathbb{R}^{N \times 3} \to \mathbb{R}^D$ that is invariant to permutations of the $N$ points, i.e.,

$$\Psi(\mathbf{PX}) = \Psi(\mathbf{X}) \quad \text{for all permutation matrices } \mathbf{P} \in \{0, 1\}^{N \times N}.$$

Architectures such as DeepSet (Zaheer et al., 2017) and PointNet (Qi et al., 2017a) satisfy this permutation invariance by construction. When these models use the 3D coordinates as input features, they are not invariant to 3D rotations (Wiersma et al., 2022; Qian et al., 2021): for a rotation $\mathbf{R} \in \mathcal{SO}(3)$, the logits produced for $\mathbf{X}$ and for its rotated version $\mathbf{XR}^\top$ need not coincide. In this work, we use such multiset architectures out-of-the-box and address this orientation sensitivity by canonicalizing the 3D rotation of the input.

**Adaptive Canonicalization over $\mathcal{SO}(3)$.** Let $\Psi : \mathbb{R}^{N \times 3} \to \mathbb{R}^D$ be a multiset neural network producing $D$ class logits. In classification, $D$ is the number of classes. We denote $\Psi(\mathbf{X}) = (\Psi_d(\mathbf{X}))_{d=1}^D$ with $\Psi_d : \mathbb{R}^{N \times 3} \to \mathbb{R}$. Here, $\Psi$ is invariant to permutations of the $N$ points, but not invariant to 3D rotations of the point cloud. The prior maximization for each class $d \in \{1, \ldots, D\}$ is performed by

$$\mathbf{R}_d^\square = \underset{\mathbf{R} \in \mathcal{SO}(3)}{\arg\max} \, h_d(\Psi_d(\mathbf{XR}^\top)),$$

where class-$d$ prior is $h_d(x) = x$. Once we found the canonical form $\mathbf{R}_d^\square$, the class scores in our anisotropic point cloud network are computed as $s_d = \Psi_d(\mathbf{XR}_d^{\square\top})$ and are then passed through a sigmoid nonlinearity to obtain class probabilities. The training loss is the sum of $D$ binary cross-entropies. Importantly, our adaptive canonicalization does not modify the underlying architecture of $\Psi$, it only evaluates $\Psi$ on rotated inputs and selects an orientation. In other words, the canonicalization setting is independent of the specific multiset neural network architecture. Below, we present three widely used multiset neural networks equipped with adaptive canonicalization.

**Adaptive Canonicalization Applied to DeepSet (AC-DeepSet).** DeepSet (Zaheer et al., 2017) defines permutation-invariant mappings on arrays via pointwise encoding and summation aggregation. It has a universal approximation property (Wagstaff et al., 2022). Given $\mathbf{X} \in \mathbb{R}^{N \times 3}$ with rows $\mathbf{x}_1, \ldots, \mathbf{x}_N$, the DeepSet has the form

$$\Psi(\mathbf{X}) = \xi \left( \sum_{i=1}^N \phi(\mathbf{x}_i) \right) \in \mathbb{R}^D,$$

where $\phi$ and $\xi$ are MLPs and $\xi = (\xi_1, \ldots, \xi_D)$ outputs $D$ logits. Applying adaptive canonicalization, for each class $d$ we select

$$\mathbf{R}_d^\square = \underset{\mathbf{R} \in \mathcal{SO}(3)}{\arg\max} \, \xi_d \left( \sum_{i=1}^N \phi(\mathbf{x}_i \mathbf{R}^\top) \right)$$

and then compute the class scores as $s_d = \xi_d \left( \sum_{i=1}^N \phi(\mathbf{x}_i \mathbf{R}_d^{\square\top}) \right)$.

**Adaptive Canonicalization Applied to PointNet (AC-PointNet).** PointNet (Qi et al., 2017a) applies a shared point encoder with a symmetric pooling operator:

$$\Psi(\mathbf{X}) = \xi\left(\text{Pool}(\phi(\mathbf{x}_1), \ldots, \phi(\mathbf{x}_N))\right) \in \mathbb{R}^D,$$

where Pool is typically max-pooling and $\xi = (\xi_1, \ldots, \xi_D)$ outputs $D$ logits. Adaptive canonicalization selects, for each class $d$,

$$\mathbf{R}_d^{\square} = \underset{\mathbf{R} \in \mathcal{SO}(3)}{\arg\max} \, \xi_d\left(\text{Pool}(\phi(\mathbf{x}_1\mathbf{R}^{\top}), \ldots, \phi(\mathbf{x}_N\mathbf{R}^{\top}))\right),$$

and the class score $s_d = \xi_d\left(\text{Pool}(\phi(\mathbf{x}_1\mathbf{R}^{\square\top}), \ldots, \phi(\mathbf{x}_N\mathbf{R}^{\square\top}))\right)$.

**Adaptive Canonicalization Applied to DGCNN (AC-DGCNN).** DGCNN (Wang et al., 2019) builds a dynamic $k$NN graph in feature space and applies edge convolution (EdgeConv) before global pooling and a final classifier head. Let $\mathbf{X} \in \mathbb{R}^{N \times 3}$ be the input shape with rows $\mathbf{x}_i \in \mathbb{R}^3$. Given point features $\mathbf{h}_i^{(\ell)} \in \mathbb{R}^{F_\ell}$ at layer $\ell$ (with $\mathbf{h}_i^{(0)} = \mathbf{x}_i$), DGCNN constructs a dynamic $k$NN graph as $\mathcal{N}_k^{(\ell)}(i) = \text{kNN}(\mathbf{h}_i^{(\ell)}, \{\mathbf{h}_j^{(\ell)}\}_{j=1}^N; k)$, and computes the edge features for neighbors $j \in \mathcal{N}_k^{(\ell)}(i)$ via a shared MLP $\varphi_\ell$:

$$\mathbf{e}_{ij}^{(\ell)} = \varphi_\ell([\mathbf{h}_i^{(\ell)} \, \| \, (\mathbf{h}_j^{(\ell)} - \mathbf{h}_i^{(\ell)})]) \in \mathbb{R}^{F_{\ell+1}}.$$

The updated node feature is updated by symmetric aggregation $\mathbf{h}_i^{(\ell+1)} = \text{Pool}_{j \in \mathcal{N}_k^{(\ell)}(i)} \mathbf{e}_{ij}^{(\ell)}$. Stacking $L$ such EdgeConv block yields features $\{\mathbf{h}_i^{(\ell)}\}_{\ell=1}^L$. DGCNN then form a global shape representation by pooling across points, given by

$$\mathbf{g}(\mathbf{X}) = \text{Pool}_{i \in [N]}\left(\text{concat}\left(\mathbf{h}_i^{(1)}, \ldots, \mathbf{h}_i^{(L)}\right)\right) \in \mathbb{R}^{F_{\text{glob}}},$$

followed by an MLP classifier head $\rho : \mathbb{R}^{F_{\text{glob}}} \to \mathbb{R}^D$:

$$\Psi(\mathbf{X}) = \rho(\mathbf{g}(\mathbf{X})) \in \mathbb{R}^D,$$

and $\Psi(\mathbf{X}) = (\Psi_d(\mathbf{X}))_{d=1}^D$. We apply adaptive canonicalization by rotating the input coordinates. For each class $d \in \{1, \ldots, D\}$ we select

$$\mathbf{R}_d^{\square} = \underset{\mathbf{R} \in \mathcal{SO}(3)}{\arg\max} \, \Psi_d(\mathbf{X}\mathbf{R}^{\top}),$$

and compute the class score as $s_d = \Psi_d(\mathbf{X}\mathbf{R}_d^{\square\top})$. As above, we apply sigmoids to $(s_d)_{d=1}^D$ and train with the sum of $D$ binary cross-entropies (one-vs-rest).

**Remark: Canonicalization on Inputs vs. Vector-Valued Representations.** For PointNet and DGCNN, adaptive canonicalization can also be applied to vector-valued internal representations, e.g., using the Vector Neuron (VN) representation (Deng et al., 2021). In VN representations, features are organized as collections of 3D vectors (rather than scalar channels). This perspective provides additional flexibility and motivation for our approach: adaptive canonicalization can be applied either (i) at the input level, by rotating $\mathbf{X}$ as in the main text, or (ii) at the representation level, by rotating a VN feature tensor (or a designated VN layer output) before the downstream classifiers. In the latter case, one canonicalizes the orientation of a learned geometric representation, which can be advantageous when early layers extract more stable directions than the input itself. In both settings, the core mechanism of adaptive canonicalization is the same: we search over $\mathcal{SO}(3)$ to select the rotation that maximizes the chosen prior score.

### A.5 Truncation Canonicalization

We introduce truncation-based prior maximization on images as an additional application of our adaptive canonicalization framework. Intuitively, many image classes are invariant under removing uninformative regions: as long as the object of interest remains in the field of view, the class label should not change. We consider, in this case, the truncation as our "symmetry" and apply prior maximization over this family of transformations to select a canonical truncation. We note that this transformation family is not based on a group action.

**Theory.** We model images as elements of a set of $\mathcal{G}$ in the following way. Given $N \in \mathbb{N}$, consider the regular grid on the unit square $[0,1]^2$ given by the pixels $Q_{i,j} = [\frac{i}{N}, \frac{i+1}{N}) \times [\frac{j}{N}, \frac{j+1}{N})$ for

$0 \leq i, j < N$. We define $\mathcal{G}$ to be the set of all functions $g \in L_2(\mathbb{R}^2)$ such that $\mathrm{supp}(g) \subset [0,1]^2$ and $g$ is piecewise constant on the grid, i.e., there exist coefficients $v_{i,j} \in [0,1]$ such that $g(x) = v_{i,j}$ for all $x \in \{Q_{i,j}\}$. Since each image $g \in \mathcal{G}$ is uniquely determined by its collection of pixel values $(v_{i,j})_{0 \leq i,j < N} \in [0,1]^{N^2}$, we can identify (via an isometric isomorphism) $\mathcal{G}$ with with the box $[0,1]^{N^2} \subset \mathbb{R}^{N^2}$. We consider the standard Euclidean metric and topology on $[0,1]^{N^2}$, which is hence a compact space. Because $\mathcal{G}$ is isometrically isomorphic to $[0,1]^{N^2}$, the set $\mathcal{G}$ is also compact with respect to the $L^2(\mathbb{R}^2)$ metric (and therefore also locally compact).

The truncation is parameterized by four coordinates collected in a vector $u = (x_{\mathrm{L}}, y_{\mathrm{T}}, x_{\mathrm{R}}, y_{\mathrm{B}}) \in \mathcal{U} \subset [0,1]^4$, where we impose the constraints $0 \leq x_{\mathrm{L}} \leq x_{\mathrm{R}} \leq 1$ and $0 \leq y_{\mathrm{T}} \leq y_{\mathrm{B}} \leq 1$. We consider the standard Euclidean metric in $\mathcal{U}$. Since $\mathcal{U}$ is a closed and bounded subset of $[0,1]^4$, it is compact.

We now define the truncation transformation. For a parameter $u = (x_{\mathrm{L}}, y_{\mathrm{T}}, x_{\mathrm{R}}, y_{\mathrm{B}}) \in \mathcal{U}$, we view $u$ as encoding the top-left and bottom-right corners of an axis-aligned rectangle: $R(u) = [x_{\mathrm{L}}, x_{\mathrm{R}}] \times [y_{\mathrm{T}}, y_{\mathrm{B}}] \subset [0,1]^2$. As $u$ varies in the compact space $\mathcal{U}$, this truncation window moves continuously inside the domain containing all images. Given an image $g \in \mathcal{G}$, we define the truncation by

$$x \mapsto \begin{cases} g(x), & x \in R(u), \\ 0, & x \notin R(u). \end{cases}$$

After truncation, we include a dilation step that maps the truncated window back to the full image domain. Let $\tau_u : [0,1]^2 \to R(u)$ be an affine map defined by

$$\tau_u(x_1, x_2) = (x_{\mathrm{L}} + (x_{\mathrm{R}} - x_{\mathrm{L}})x_1, \ y_{\mathrm{T}} + (y_{\mathrm{B}} - y_{\mathrm{T}})x_2).$$

The function lies in $L_2(\mathbb{R}^2)$ but not in $\mathcal{G}$ in general. We then project it back onto the finite-dimensional subspace $\mathcal{G}$ of piecewise constant functions by the orthogonal projection $Q : L_2(\mathbb{R}^2) \to \mathcal{G}$ defined by

$$(Qg)(x) = \sum_{i,j=1}^{N} \bar{g}_{ij} \mathbb{1}_{Q_{ij}}(x), \qquad \bar{g}_{ij} = N^2 \int_{Q_{ij}} g(y) dy.$$

We then define $\kappa_u : \mathcal{G} \to \mathcal{G}$ by

$$(\kappa_u(g)) = Q((g \, \mathbb{1}_{R(u)}) \circ \tau_u).$$

We now verify that this truncation family satisfies our assumptions. By construction, $\mathcal{U}$ is compact, and the image space $\mathcal{G}$ is compact. In our setting we have $\mathcal{K} = \mathcal{G}$, so $\mathcal{K}$ is also compact. The map

$$\kappa_{(\cdot)}(\cdot\cdot) : \mathcal{U} \times \mathcal{G} \to \mathcal{K}, \quad (u, g) \mapsto \kappa_u(g) \in \mathcal{K}$$

is continuous, and hence, by compactness of the product $\mathcal{U} \times \mathcal{G}$, it is uniformly continuous. This implies that $\{g \mapsto f(\kappa_u(g))\}_{u \in \mathcal{U}}$ is equicontinuous. Thus, the truncation transformation family satisfies the conditions for continuous canonicalization.

**Implementation.** We work with array images $\mathbf{X} \in \mathbb{R}^{N \times N}$ and use a finite set of admissible truncations. Instead of allowing truncation at fractions of pixels, in practice we only allow truncation at the boundaries of the pixels. We also exclude very small windows, since they are likely to contain only a part of the object to be classified. Formally, this means that we consider a finite transformation set $\mathcal{T}$ and endow it with the trivial metric. That is, the distance is 1 between distinct truncations. Hence, $\mathcal{T}$ is compact and the truncation map is trivially continuous under this metric, so the attainment arguments for prior maximization adaptive canonicalization apply verbatim.

Concretely, we parameterize pixel-aligned windows by a side-length $s \in [s_{\min}, s_{\max}] \subset [0.5, 1.0]$ and a discrete top-left corner. For a given $s$, we set the truncation side length to $M(s) = \lfloor sN \rfloor$, and require the window to lie inside the image domain, i.e., $1 \leq i_0 \leq N - M(s) + 1$ and $1 \leq j_0 \leq N - M(s) + 1$. The corresponding axis-aligned window is $\{(i,j) \in \{1, \ldots, N\}^2 \mid i_0 \leq i \leq i_0 + M(s) - 1, \ j_0 \leq j \leq j_0 + M(s) - 1\}$. The truncation then keeps only the pixels inside the window and zeros out the rest, which yields a truncated patch $C_{(i_0, j_0, s)}(\mathbf{X}) \in \mathbb{R}^{M(s) \times M(s)}$. To keep the input dimension fixed, we then rescale this cropped patch back to the original resolution $N \times N$ using a standard interpolation scheme. We denote this resizing operator by $R_s : \mathbb{R}^{M(s) \times M(s)} \to \mathbb{R}^{N \times N}$. The truncation operator is therefore defined as the composition

$$T_{(i_0, j_0, s)}(\mathbf{X}) = R_s(C_{(i_0, j_0, s)}(\mathbf{X})) \in \mathbb{R}^{N \times N}.$$

The transformation family used in our method is
$$\mathcal{T} = \{T_{(i_0,j_0,s)} \mid s \in [s_{\min}, s_{\max}], \ (i_0, j_0) \text{ admissible as above}\},$$
and random truncations are obtained by sampling $s$ and $(i_0, j_0)$ from a suitable distribution under these constraints.

**Truncation Canonicalization by Prior Maximization.** We consider image classification with $D$ classes. Let $\Psi : \mathbb{R}^{N \times N} \to \mathbb{R}^D$ be a symmetryless neural network and we write $\Psi(\mathbf{X}) = (\Psi_d(\mathbf{X}))_{d=1}^D$. For each class $d \in \{1, \ldots, D\}$ we select the canonical truncation by
$$T_{(i_0,j_0,s)}^{\square} = \underset{T_{(i_0,j_0,s)} \in \mathcal{T}}{\arg\max} \ h_d(\Psi_d(T_{(i_0,j_0,s)}(\mathbf{X}))),$$
and compute the class-$d$ score as $s_d = \Psi_d(T_{(i_0,j_0,s)}^{\square}(\mathbf{X}))$. The scores $(s_d)_{d=1}^D$ are passed through sigmoids to obtain per-class probabilities, and training uses the sum of $D$ binary cross-entropies (one-vs-rest).

## B   EXTENDED RELATED WORK

We provide an extended discussion of related work for equivariant machine learning.

### B.1   CANONICALIZATION

Canonicalization (Babai & Luks, 1983; Palais & Terng, 1987; Ma et al., 2023; 2024) is a classical strategy for handling symmetry in data, especially for tasks where invariance or equivariance to group actions is desirable (Gerken et al., 2023). It preprocesses each input by mapping it to a standard form prior to downstream learning and inference, so that all symmetry-equivalent inputs are treated identically by the subsequent model. There are two common ways for canonicalization: fixed or learned approaches.

- **Fixed Canonicalization.** Fixed canonicalization uses deterministic and often analytic procedures to assign a unique representative to each symmetry orbit. For example, principal component analysis (PCA) (Jolliffe & Cadima, 2016) alignment canonically orients an object (e.g., a point cloud or molecule) by rotating it so its principal components align with the coordinate axes (Kazhdan et al., 2003). Procrustes analysis (Gower, 1975) canonically orients sets of points by finding the optimal rotation, translation, and scale that minimizes squared point-to-point distances to a reference. For sets and graphs, canonicalization can be achieved by reordering nodes, atoms, or features so that isomorphic inputs share a single labeling. In spectral methods and spectral graph neural networks where eigenvectors are fundamental (Kipf & Welling, 2017; Defferrard et al., 2016a; Von Luxburg, 2007; Belkin & Niyogi, 2003; Dwivedi et al., 2023; Maskey et al., 2022; Bracha et al., 2024; Velich et al., 2025), canonicalization of spectral decomposition (Lim et al., 2023b;a; Ma et al., 2023; 2024) addresses eigenbasis ambiguity (Chung, 1997; Spielman, 2012) by processing each eigenspace independently and selecting representative eigenvectors or directions by applying orthogonal or axis-based projections, typically as a graph preprocessing step. An alternative approach is eigenbasis canonicalization via the input signal, where the signal itself is used to define a canonical spectral representation, making the spectral transformation independent of the arbitrary choice of eigenvectors (Lin et al., 2024a; Geisler et al., 2024).
- **Learned Canonicalization.** Learned canonicalization (Zhang et al., 2019a; Kaba et al., 2023; Luo et al., 2022) seeks to overcome the rigidity and inflexibility of fixed rules with a trainable mapping that selects a representative for each symmetry orbit. The canonicalizer is parameterized (typically as a neural network and trained to produce canonical forms. For example, Kaba et al. (2023) developed a neural network that learns the canonicalization transformation, which enables plug-and-play equivariance, e.g., orthogonalizing learned features via the Gram-Schmidt process (Trefethen & Bau, 2022). Their results show that the learned canonicalizers outperform fixed canonicalizers.

However, Dym et al. (2024) pointed out that regardless of whether the canonicalizations are learned or not, a continuous canonicalization does not exist for many common groups (e.g. $S_n$, SO(d), O(d) on point clouds $n \geq d$). Therefore, while learned canonicalization improves empirical performance,

it remains generally discontinuous and can induce instability, hinder generalization, limit model reliability on boundary cases, or out-of-distribution data. In contrast, our adaptive canonicalization framework learns the optimal transformation for each input by maximizing the predictive confidence of the network, resulting in a continuous and symmetry preserving mapping. We include a detailed comparison of most related canonicalization work with our method below.

### B.1.1 Equivariance with Learned Canonicalization Functions

Recently, Kaba et al. (2023) introduced energy-based canonicalization. The central idea is to learn an energy function over samples and group elements, and define the canonicalization as minimizing this energy with respect to the group, given a fixed datapoint. Specifically, their energy minimization is related to our prior maximization adaptive canonicalization method. However, there are several important conceptual and technical differences between their approach and our adaptive canonicalization.

First, their energy $s$ is not the task neural network like in our analysis, but rather some other trainable neural network. Similarly to our approach, training $s$ end-to-end with the task neural network can be seen as canonicalization that depends on the task network, if one considers the full end-to-end architecture consisting both of the energy minimization and the task network. However, the approach in Kaba et al. (2023) does not give continuity guarantees as opposed to our approach. Second, in their work, they consider symmetries based on group actions, while we consider a more general setting of canonicalization transformations (or augmentations) that need not be based on groups. This makes our approach much more applicable across different domains. Moreover, in their framework, there is one canonical form for each datapoint and network, while in our approach, each output channel of the network defines a different canonical form of the datapoint. This allows our approach to preserve continuity. Notably, Kaba et al. (2023) do not attempt to study the continuity of the end-to-end predictor.

Another difference in Kaba et al. (2023) is that when training is initialized, the canonicalizing energy $s$ is random. This leads each datapoint to be randomly transformed, so the task neural network initially has to perform well at all orientations of the data. This can lead the task network to ultimately learn an "average behavior," not specializing in any special orientation but rather performing reasonably well on all orientations of the data. In other words, the limited set of trainable parameters has to simultaneously specialize in many orientations, which reduces the network's expressive power. In contrast, in our prior maximization approach, from the beginning of training, the network only pays a price for not performing well on the single best orientation per datapoint (on which the network performs the best). This encourages the network to specialize on one canonical orientation per datapoint, and not learn an average behavior. Hence, in our approach, all trainable parameters of the task network can focus on performing well only on the sole canonical orientation of each datapoint.

### B.1.2 Canonicalization and Data Re-Alignments

As discussed in Appendix B.1.1, in the energy-based canonicalization framework of (Kaba et al., 2023), the canonicalizing energy $s$ is random at initialization. As a result, it leads each datapoint to be randomly transformed and the task neural network initially has to perform well at all orientations. This can lead the task network to ultimately learn an "average behavior," not specializing in any special orientation but rather performing reasonably well on all orientations of the data. To address this effect, Mondal et al. (2023) biases the canonical transformation of each datapoint to be the identity, assuming that the datapoints in the training set already have a small orientation variance. On top of that, Schmidt & Stober (2025) iteratively reduces the orientation variance of the training set by iteratively reorienting datapoints that lead to a large loss. We note that these approaches are rather different from our prior maximization method, and they do not try to address the continuity problem in canonicalization.

### B.1.3 Weighted Canonicalization

The energy-based canonicalization (Kaba et al., 2023) was further explored and extended in (Shumaylov et al., 2025) on symmetries defined by general Lie group actions. Similarly to our work, Shumaylov et al. (2025) also discusses continuity preservation, but their approach is different from ours. In the work of Shumaylov et al. (2025), they define the notion of weighted canonicalization,

which is a similar concept to the weighted frame introduced by Dym et al. (2024). Here, to each data-point there is an assigned probability measure over the orbit of the datapoint. Namely, the distribution is over the space of data instead of over the group like in the weighted frames of Dym et al. (2024). With respect to energy minimization, this approach is not very different from Kaba et al. (2023). The main difference is in the minimization algorithm, which minimizes over the Lie algebra instead of the Lie group. It is important to note that their work does not train the energy end-to-end with the neural network. Hence, in their work, the canonicalization depends on the whole training set, but not on the task network, which is quite different from our approach. Moreover, in their setup, one has to learn an approximation of the data distribution, which is invariant to the group action. This is a highly nontrivial approach to implement. In contrast, our prior maximization is simple and direct.

### B.1.4 TEST-TIME CANONICALIZATION

Another recent extension of canonicalization, (Singhal et al., 2025) , explores a set of transformations at test time and uses the scoring functions of large pre-trained foundation models like CLIP (Radford et al., 2021) or SAM (Kirillov et al., 2023) to select the most "canonical" representation upon which downstream inference is performed. Note that the work in Singhal et al. (2025) does not involve network retraining, uses the foundation models as is, and performs canonicalization entirely at inference by optimizing over transformations. While it achieves strong empirical performance, its canonicalization mapping is not guaranteed to be continuous, and in fact, continuity is not discussed. Therefore, small input changes may cause abrupt switches in the selected canonical view.

A related line of work is inverse transformation search (Schmidt & Stober, 2024), which also performs test-time optimization over transformations to exploit invariances. Their method focuses on the standard action of the special linear group on images (rotations, scalings, and shear transformations), i.e., purely group-based symmetries, and similar to Singhal et al. (2025), does not address continuity. In addition, their method does not train the model simultaneously with the canonicalization. While their energy-induced confidence is similar to our prior-maximization formulation in the classification setting, it does not lead to a continuity guarantee. In contrast, our work rigorously develops sufficient conditions on the canonicalizer for the canonicalized network to be continuous and have a universal approximation property. Specifically, our one-vs.-all setting is a different type of energy that does lead to continuous end-to-end classifiers.

### B.2 FRAME AVERAGING

Frame averaging (Puny et al., 2022) achieves equivariance to group symmetries by averaging a network's output over a set of group transformations (known as a "frame"). It is built on the classical group averaging operator, which guarantees symmetries by summing a function over all group elements. Frame averaging has two main advantages: 1) it allows adaptation of standard non-equivalent network architectures to handle symmetry, similar to canonicalization methods, and 2) it avoids computational intractability of full group averaging, especially for large or continuous groups. Recent work (Lin et al., 2024b) proposes minimal frame averaging that attains strong symmetry coverage with small frames. Domain-specific frame averaging methods (Duval et al., 2023b; Atzmon et al., 2022) show that it can be deployed in material modeling and geometric shape analysis. However, it requires a careful selection of a suitable frame. In addition, frame averaging uses a fixed set of transformations independent of the input or task, potentially leading to sub-optimal or less discriminative feature representations. In contrast, our adaptive canonicalization learns the optimal transformation for each input in a data- and network-dependent way, yielding symmetry preserving continuous functions that can improve representation quality and empirical task performance.

A related work that addresses continuity is the weighted frame averaging proposed by Dym et al. (2024). They first prove that in many well-known cases, continuous canonicalization is impossible. This does not contradict our work, as in Dym et al. (2024) the canonicalization is a function solely of the datapoint, and not the task network. Then, they define a variant of frame averaging, called weighted frame averaging, in which to each datapoint there is an associated probability distribution over the group, and the frame averaging is performed with respect to this measure. This construction yields continuity guarantees. However, their focus is fundamentally different from ours: they study frame averaging while we focus on canonicalization. Moreover, the weighted frame averaging is a function only of the data, not the neural network, as opposed to our method. We next compare our framework to weighted frame averaging in more detail.

- In our method, data need not come from a vector space of a fixed dimension (see e.g., the application for graphs). In contrast, weighted frame averaging requires working with data that comes from a vector space of a fixed dimension.
- In our work, symmetries need not be based on group representations. Our notion of symmetry is called a transformation family, and our "symmetries" are not even required to be invertible or based on a group action. See, for example, the image truncation transformation in Appendix A.5. On the other hand, the symmetries in weighted frame averaging are required to be representations of compact groups.
- In the framework of weighted frame averaging, the requirement that the weighted frame is robust is sufficient for continuity of the canonicalized function. However, this requirement is quite strong, and constructing robust frames could be challenging. In contrast, we achieve continuity of the canonicalized function even though the maximizer in prior maximization need not be continuous in any sense (as a function of the datapoint). The maximizer need not even be uniquely defined. Hence, practitioners can use our method out of the box on new domains with new symmetries without having to prove any nontrivial mathematical results. In weighted frame averaging, employing the method in a new setting typically requires carefully constructing a problem-specific weighted frame and proving that it is robust. It often involves nontrivial mathematical work and there is no general recipe for doing so. In our case, once an architecture and a symmetry setting are chosen, prior-maximization adaptive canonicalization is straightforward to implement and does not require additional sophisticated proofs from the practitioner. The only assumption that needs to be checked is continuity of the chosen transformations, which is usually easy to verify.
- Robust weighted frames often require averaging the predicting network over many transformations of the input. In practice, our prior maximization approach works with fewer random argmax candidates. For comparison, frame averaging for rotations of $n$ points in a 3D point cloud requires an order of $n^2$ transformations (where $n = 1024$ in the ModelNet40 dataset for example), while, in practice, prior maximization performs well with a total of 50 random transformations.

These differences imply that our framework is more flexible in terms of the data types and symmetry structures it can accommodate, while also imposing a lighter mathematical burden on practitioners who wish to apply it. This gives much more freedom for future practitioners, and may lead to a wider adaptation of the method.

We note that the earlier work (Basu et al., 2023) proposed a similar idea to weighted frame averaging, but did not study continuity preservation.

### B.3 EQUIVARIANT ARCHITECTURES

Equivariant architectures are a class of models explicitly designed to respect symmetry groups acting on the data. Formally, a network is equivariant to a group of transformations if, when the input is transformed by a symmetry group action, the output transforms via the same group action. That is, for a group $G$ and a function $f$, equivariance guarantees $f(g \cdot x) = g \cdot f(x)$ for all $g \in G$ and input $x$. It has been developed across images (Cohen & Welling, 2016; 2017; Kondor & Trivedi, 2018; Worrall et al., 2017; Weiler et al., 2018b), graphs (Bronstein et al., 2017; Zaheer et al., 2017; Gilmer et al., 2017; Maron et al., 2019a; Kofinas et al., 2024; Thiede et al., 2020; Vignac et al., 2020; Keriven & Peyré, 2019), molecules (Thomas et al., 2018; Brandstetter et al., 2022a; Anderson et al., 2019; Fuchs et al., 2020; Satorras et al., 2021; Duval et al., 2023a; Schütt et al., 2017; Liao & Smidt, 2023; Hordan et al., 2025; Du et al., 2022; Passaro & Zitnick, 2023), and manifolds (Masci et al., 2015; Monti et al., 2017; Cohen et al., 2019; Weiler et al., 2021). Notably, Maron et al. (2019b) studies the universal approximation property for equivariant architectures. Notably, equivariant networks often demand group-specific designs that rely on group theory, representation theory, and tensor algebra. This can reduce flexibility and raise compute and memory costs (Pertigkiozoglou et al., 2024; Wang et al., 2024; Liao & Smidt, 2023). Making nonlinearities, pooling, and attention strictly equivariant further constrains layer choices and can increase parameter count and runtime. Moreover, imposing symmetry throughout the stack may limit the expressivity when data only approximately respect the assumed symmetry or contain symmetry-breaking noise (Wang et al., 2022; Lawrence et al., 2025). In contrast, our approach handles symmetry by learning an input- and task-dependent canonical form through prior maximization and applying a standard backbone. By construction, this mapping is

continuous and symmetry preserving. Our adaptive canonicalization removes heavy group-specific layers, reduces per-layer equivariant cost, and keeps ordinary nonlinearities and pooling.

### B.4 DATA AUGMENTATION

Data augmentation (Shorten & Khoshgoftaar, 2019; Mumuni & Mumuni, 2022; Yang et al., 2022) incorporates symmetry by explicitly applying symmetry transformations to training inputs, thereby encouraging the model to make predictions that are stable (or transform predictably) under those transformations. In this way, augmentation promotes invariance or equivariance by exposing the model to multiple transformed views of the same input (i.e., broader coverage of the transformation orbit). In a sense, augmentation changes the data distribution rather than the model. For example, if the original data consists of images of axis-aligned rectangles, then applying rotation augmentation induces a data distribution containing rectangles at general orientations. Recent work has analyzed augmentation through multiple lenses, including training dynamics, implicit regularization, group-theoretic foundations, and learnable augmentation strategies (Gens & Domingos, 2014; Chen et al., 2020; Yang et al., 2023; Hernández-García & König, 2018; Hoffer et al., 2019; Shen et al., 2022; Dao et al., 2019; Santos-Escriche et al., 2025; Zhang et al., 2018; Zhong et al., 2020; DeVries & Taylor, 2017; Hendrycks et al., 2020; Tahmasebi et al., 2025). While both data augmentation and our adaptive canonicalization use transformations, they do so in fundamentally different ways. Augmentation adds transformed samples and implicitly trains a task network to behave consistently across them, so symmetry is encouraged indirectly through the distributional shift entailed by augmentation. Adaptive canonicalization instead selects (for each input) a canonical transformation from the same family by optimizing a prior score, and then evaluates the task network on the resulting canonical representative. Thus, augmentation spreads supervision across many transformed versions, whereas canonicalization explicitly reduces transformation ambiguity by mapping each example to a preferred representative. Moreover, canonicalization can be anisotropic, selecting different canonical transformations for different decision heads, which is not captured by standard augmentation strategies.

## C FUNCTIONAL CALCULUS AND SPECTRAL FILTERS

In this section, we recall the theory of plugging self-adjoint operators inside functions.

**Spectral Theorem.** Let $\mathcal{L}$ be a self-adjoint operator on a finite-dimensional Hilbert space (e.g., $\mathcal{L} \in \mathbb{C}^{N \times N}$ with $\mathcal{L} = \mathcal{L}^*$). There exists a unitary matrix $V$ and a real diagonal matrix $\Lambda = \mathrm{diag}(\lambda_1, \ldots, \lambda_N)$ such that $\mathcal{L} = V \Lambda V^*$. The columns $v_i$ of $V$ form an orthonormal eigenbasis with $\mathcal{L} v_i = \lambda_i v_i$.

**Functional Calculus.** For any function $f : \mathbb{R} \to \mathbb{C}$ defined on the spectrum $\sigma(\mathcal{L}) = \{\lambda_1, \ldots, \lambda_N\}$:
$$f(\mathcal{L}) := V f(\Lambda) V^*, \qquad f(\Lambda) = \mathrm{diag}\big(f(\lambda_1), \ldots, f(\lambda_N)\big).$$
Equivalently, we can write the spectral projections $P_i := v_i v_i^*$,
$$f(\mathcal{L}) = \sum_{i=1}^{N} f(\lambda_i) \, P_i.$$
We can plug a self-adjoint matrix into a function by: 1) diagonalize $\mathcal{L}$, 2) apply $f$ to the eigenvalues, and 3) conjugate back.

Take $f = \mathbb{1}_I$ for a Borel set $I \subset \mathbb{R}$. The indicator function $\mathbb{1}_I(\mathcal{L})$ is an orthogonal projection, since $\mathbb{1}_I(\mathcal{L})^2 = \mathbb{1}_I(\mathcal{L})$ and $\mathbb{1}_I(\mathcal{L})^* = \mathbb{1}_I(\mathcal{L})$.

**Spectral Graph Filters.** A *graph shift operator (GSO)* is a self-adjoint matrix that reflects the graph's connectivity, such as a (normalized) graph Laplacian or a symmetrized adjacency. Let $\mathcal{L}$ be such a GSO with eigenpairs $\{(\lambda_i, v_i)\}_{i=1}^{N}$ and $V = [v_1, \ldots, v_N]$. For a $T$-channel node signal $\mathbf{X} \in \mathbb{R}^{N \times T}$ and a matrix-valued frequency response $g : \mathbb{R} \to \mathbb{R}^{d' \times T}$, the spectral filter
$$g(\mathcal{L})\mathbf{X} := \sum_{i=1}^{N} v_i v_i^\top \, \mathbf{X} \, g(\lambda_i)^\top$$

applies the graph convolution theorem (Bracewell & Kahn, 1966) with $d'$ output channel: each Fourier mode $v_i$ is preserved in space, while channels are mixed by $g(\lambda_i)$ in the spectral domain. In the scalar case ($T = d' = 1$) with $f : \mathbb{R} \to \mathbb{R}$, the spectral filter simply reduces to the functional-calculus operator acting on $\mathbf{X}$:

$$f(\mathcal{L})\mathbf{X} = \sum_{i=1}^{N} f(\lambda_i)\, v_i v_i^\top \mathbf{X} \; = \; V f(\Lambda) V^\top \mathbf{X}.$$

Spectral graph neural networks (Defferrard et al., 2016b; Kipf & Welling, 2017; Levie et al., 2018) compose such filters with pointwise nonlinearities, using trainable $g$ at each layer.

# D  UNIVERSAL APPROXIMATION THEOREMS

## D.1  UNIVERSAL APPROXIMATION OF EUCLIDEAN FUCNTIONS

Here, we cite a classical result stating that multilayer perceptrons (MLPs) are universal approximators of functions over compact sets in Euclidean spaces.

**Theorem 14** (Universal Approximation Theorem (Leshno et al., 1993)). *Let $\sigma : \mathbb{R} \to \mathbb{R}$ be a continuous, non-polynomial function. Then, for every $M, L \in \mathbb{N}$, compact $\mathcal{K} \subseteq \mathbb{R}^M$, continuous function $f : \mathcal{K} \to \mathbb{R}^L$ , and $\varepsilon > 0$, there exist $D \in \mathbb{N}$, $\mathbf{W}_1 \in \mathbb{R}^{D \times M}$, $\mathbf{b}_1 \in \mathbb{R}^D$ and $\mathbf{W}_2 \in \mathbb{R}^{L \times D}$ s.t.*

$$\sup_{\boldsymbol{x} \in \mathcal{K}} |f(\boldsymbol{x}) - \boldsymbol{W}_2 \sigma (\boldsymbol{W}_1 \boldsymbol{x} + \boldsymbol{b}_1)| \le \varepsilon. \tag{4}$$

## D.2  UNIVERSAL APPROXIMATION OF MULTI-SETS FUNCTIONS

Multisets are sets where repetitions of elements are allowed. Formally, a multiset of $N$ elements in $\mathbb{R}^J$ can be defined as a set of pairs $(x, i)$ where $x$ denotes a point and $i$ the number of times the point $x$ appears in the multiset.

Standard universal approximation analysis of multi-set functions goes along the following lines. First, we represent multisets as 2D arrays $\mathbf{X} \in \mathbb{R}^{N \times J}$, where each row $\mathbf{X}_{n,:} \in \mathbb{R}^J$ represents one point in the multiset. Note that the same multi-set can be represented by many arrays. In fact, two arrays $\mathbf{X}$ and $\mathbf{X}'$ represent the same multi-set if and only if one is a permutation of the other.

To formulate this property, let $\mathcal{S}_N$ be the symmetric group of $N$ elements, i.e., the group of permutations of $N$ elements. Given $s \in \mathcal{S}$ and $\mathbf{X} \in \mathbb{R}^{N \times J}$, let $\rho(s)\mathbf{X}$ denote the permutation of $\mathbf{X}$ via $s$. By convention, permutations change the order of the $N$ rows of $\mathbf{X}$, and keeps each row intact. Now, $\mathbf{X}$ and $\mathbf{X}'$ represent the same multi-set if and only if there is a permutation $s \in \mathcal{S}_N$ such that $\mathbf{X} = \rho(s)\mathbf{X}'$.

Now, for a function $y : \mathbb{R}^{N \times J} \to \mathbb{R}^D$ to represent a multi-set function, it should be invariant to permutations, i.e., for every $s \in \mathcal{S}_N$ we have $y(\rho(s)\mathbf{X}) = y(\mathbf{X})$. Hence, standard UATs of multi-set functions are formulated based on the following notion of universal approximation.

**Definition 15.** *Let $\mathcal{K} \subset \mathbb{R}^{N \times J}$ be an invariant compact domain, i.e., $\rho(s)\mathbf{X} \in \mathcal{K}$ for every $\mathbf{X} \in \mathcal{K}$ and $s \in \mathcal{S}_N$. A set of invariant functions $\mathcal{N}(\mathcal{K}, \mathbb{R}^D) \subset C_0(\mathcal{K}, \mathbb{R}^D)$ is called an invariant universal approximator of $C_0(\mathcal{K}, \mathbb{R}^D)$ equivariant functions if for every invariant function $y \in C_0(\mathcal{K}, \mathbb{R}^D)$ and $\epsilon > 0$ where is $\theta \in \mathcal{N}(\mathcal{K}, \mathbb{R}^D)$ such that for every $\mathbf{X} \in \mathcal{K}$*

$$|\theta(\mathbf{X}) - y(\mathbf{X})| < \epsilon.$$

For example, DeeptSets are universal approximators of invariant $C_0(\mathcal{K}, \mathbb{R}^D)$ functions (Wagstaff et al., 2022).

In this section, we describe an alternative, but equivalent, approach to model multisets of size $N$ and their universal approximation theorems using the notion of quotient. The motivation is that our main UAT, Theorem 13, is based on the standard symmetryless notion of universal approximation, Definition 3. While it is possible to develop our adaptive canonicalization theory for functions that preserve symmetries, and obtain an analogous theorem to Theorem 13 based on invariant universal approximators, there is no need for such complications. Instead, we can use the standard definition of universal approximation (Definition 3), and directly encode the symmetries in the domain using quotient spaces, as we develop next.

**Quotient Spaces.** Let $\mathcal{X}$ be a topological space, and $x \sim y$ an equivalence relation between pairs of points. The equivalence class $[x]$ of $x \in \mathcal{X}$ is defined to be the set

$$[x] := \{y \in X \; x \sim y\}.$$

**Definition 16** (Quotient topology). *Let $\mathcal{X}$ be a topological space, and $\sim$ an equivalence relation on $\mathcal{X}$. The* quotient set *is defined to be*

$$\mathcal{X}/\sim := \{[x] \mid x \in \mathcal{X}\}.$$

*The quotient set is endowed with the* quotient topology. *The quotient topology is the finest (largest) topology making the mapping $\nu : x \mapsto [x]$ continuous. In other words, the open sets $B \subset (\mathcal{X}/\sim)$ are those sets such that $\cup_{[x] \in B}[x]$ is open in $\mathcal{X}$.*

The mapping $\nu : \mathcal{X} \to (\mathcal{X}/\sim)$, defined by $\nu(x) = [x]$, is called the *canonical projection.*

**Multi-Sets as Equivalence Classes and UATs.** Define the equivalence relation: $\mathbf{X} \sim \mathbf{Y}$ if there exists $s \in \mathcal{S}_N$ such that $\mathbf{X} = \rho(s)\mathbf{Y}$. Now, a multi-set of $N$ elements can be defined as $\mathbb{R}^{N \times J}/\sim$. As opposed to the definition of multisets as sets of pairs $(x, i)$, the quotient definition automatically gives a topology to the sets of multisets, namely, the quotient topology. In fact, it can be shown that the quotient topology is induced by the following metric.

**Definition 17** (Multi-Set Metric). *Given $[\mathbf{X}], [\mathbf{Y}] \in (\mathbb{R}^{N \times J}/\sim)$, their distance is defined to be*

$$d([\mathbf{X}], [\mathbf{Y}]) := \min_{\mathbf{X}' \in [\mathbf{X}], \mathbf{Y}' \in [\mathbf{Y}]} \|\mathbf{X}' - \mathbf{Y}'\|.$$

It is easy to see that

$$d([\mathbf{X}], [\mathbf{Y}]) := \min_{s \in \mathcal{S}_N} \|\mathbf{X} - \rho(s)\mathbf{Y}\|.$$

One can show that the above distance is indeed a metric, and that the topology induced by this metric is exactly the quotient topology, i.e., $d$ metrizes $(\mathbb{R}^{N \times J}/\sim)$.

**Theorem 18.** *The metric $d([\mathbf{X}], [\mathbf{Y}])$ metrizes the quotient topology $\mathbb{R}^{N \times J}/\sim$.*

A set $\mathcal{K} \subset \mathbb{R}^{N \times J}$ is called invariant if for every $\mathbf{X} \in \mathcal{K}$ and $s \in \mathcal{S}_N$ we have $\rho(s)\mathbf{X} \in \mathcal{K}$. Consider the quotient space

$$\mathcal{K}/\sim = \{[\mathbf{X}] \mid \mathbf{X} \in \mathcal{K}\} \subset \mathbb{R}^{N \times J}/\sim .$$

We now have the following proposition about the continuity of symmetric functions.

**Proposition 19.** *Let $\mathcal{K} \subset \mathbb{R}^{N \times J}$ be an invariant compact domain. For every continuous invariant mapping $y : \mathcal{K} \to \mathbb{R}^D$ there exists a unique continuous mapping $\overline{y} : (\mathcal{K}/\sim) \to \mathbb{R}^D$ such that*

$$y = \overline{y} \circ \nu,$$

*where $\nu$ is the canonical projection. On the other hand, for every function $z \in C_0(\mathcal{K}/\sim, \mathbb{R}^D)$, we have that $z \circ \nu$ is a continuous invariant function in $C_0(\mathcal{K}, \mathbb{R}^D)$.*

Let $\mathcal{K} \subset \mathbb{R}^{N \times j}$ be an invariant domain. For a set of continuous invariant functions $\mathcal{N} \subset C_0(\mathcal{K}, \mathbb{R}^D)$, we denote

$$(\mathcal{N}/\sim) := \{\overline{y} \mid y \in \mathcal{N}\} \subset C_0(\mathcal{K}/\sim, \mathbb{R}^D).$$

Note that by Proposition 19

$$\left(C_0(\mathcal{K}, \mathbb{R}^D)/\sim\right) = C_0\left(\mathcal{K}/\sim, \mathbb{R}^D\right).$$

This immediately leads to a UAT theorem for multi-set continuous functions in which every continuous multi-set function can be approximated by a neural network.

**Theorem 20.** *Let $\mathcal{K} \subset \mathbb{R}^N$ be an invariant compact domain, and let $\mathcal{N}(\mathcal{K}, \mathbb{R}^D) \subset C_0(\mathcal{K}, \mathbb{R}^D)$ be an invariant universal approximator of $C_0(\mathcal{K}, \mathbb{R}^D)$ equivariant functions. Then $\mathcal{N}(\mathcal{K}, \mathbb{R}^D)/\sim$ is a universal approximator $C_0(\mathcal{K}/\sim, \mathbb{R}^D)$.*

Note that in the above UAT the symmetries are directly encoded in the quotient spaces, and, hence, there is no need to encode any symmetry in the spaces of functions. Hence, Theorem 20 is based on the standard symmetryless definition of universal approximation – Definition 3 – rather than the symmetry driven construction of Definition 15. As a result, we can directly use our theory of adaptive

canonicalization on multi-set functions. Specifically, we can use Theorem 13, where the space $\mathcal{K}$ in the theorem is taken as $\mathcal{K}/\sim$ in our above analysis.

Now, we immediately obtain that the set of neural networks $\overline{\theta}$ where $\theta$ is a DeepSet is a universal approximator of the space of continuous multi-set functions.

# E    THEORETICAL ANALYSIS OF RANDOMIZED PRIOR MAXIMIZATION

Consider the randomized maximization method described in Appendix A.2. To understand how well this random maximization approximates the ideal maximization over the full transformation space, we can adopt the analysis from Cordonnier et al. (2024). Their results provide a tail bound for approximating a maximum over a probability space by the maximum over random i.i.d. samples.

We first recall the notion of the volume retaining probability space introduced by Cordonnier et al. (2024).

**Definition 21** (Volume retaining property (Cordonnier et al., 2024)). *Let $X \subset \mathbb{R}^d$ and let $P$ be a probability measure on $X$. We say that the probability space $(X, P)$ has the $(r_0, \kappa)$-volume retaining Lebesgue measure if there exist constants $r_0 > 0$ and $\kappa > 0$ such that for any $r \leq r_0$ and any $x \in X$*

$$P(B(x,r) \cap X) \geq \kappa \lambda_d(B(x,r)),$$

*where $\lambda_d$ is the $d$-dimensional Lebesgue measure and $B(x,r)$ is the ball center at $x$ with radius $r$.*

In our case, points are randomly sampled from some canonical measure over $\mathcal{U}_j$ (i.e., the Haar measure), and in all of our example applications $\mathcal{U}_j$ has the volume retaining property.

On a volume retaining space, Cordonnier et al. (2024) prove the following concentration inequality for maxima.

**Lemma 22** (Concentration inequality for volume retaining space (Cordonnier et al., 2024)). *Let $(X, P)$ be a probability space with the $(r_0, \kappa)$-volume retaining property and let $g : X^2 \to \mathbb{R}^q$ be $K_g$-Lipschitz. For any $\rho \geq \exp(-n\kappa r_0^d 2^d)$, for any random variables $X_1, \ldots, X_n \overset{i.i.d.}{\sim} P$, with probability at least $1 - \rho$, it holds*

$$\| \max_{1 \leq i \leq n} g(x, X_i) - \sup g(x, \cdot) \|_\infty \leq \frac{K_g}{2} \left( \frac{\ln(q/\rho)}{n\kappa} \right)^{1/d}.$$

Applying this lemma to our setting, $g$ is the output of the task neural network on some class. Since typical neural networks are Lipschitz continuous (e.g., any multilayer perceptron based on a Lipschitz activation function), this immediately gives a guarantee that our random maximization method approximates the true maximum. We plan to extend this analysis for future work.

Finally, we note that when the prior maximization has error $e$ (which is a random variable), this leads to an additive term $e$ in universal approximation. Namely, for any $\epsilon > 0$, any continuous function can be approximated by $\theta \circ \rho_\theta$ up to error $\epsilon + e$ instead of $\epsilon$ in Theorem 6.

# F    ADDITIONAL EXAMPLES OF CONTINUOUS PRIOR MAXIMIZATION

We first note that when $\mathcal{K}$ is a locally compact metric space, functions in $C_0(\mathcal{K}, \mathbb{R}^D)$ must be uniformly continuous.

## F.1    UNBOUNDED POINT CLOUDS AND ROTATIONS

Let $\mathcal{U} = \mathcal{SO}(3)$ be the space of 3D rotations, and $\mathcal{G} = \mathcal{K} = \mathbb{R}^{N \times 3}$ the set of sequences of $N$ points in $\mathbb{R}^3$, i.e. the space of point clouds. Consider the $\mathcal{L}_2$ metric in $\mathcal{G}$. Consider the rotation $g \mapsto \kappa_u(g)$ of the point cloud $g$ by $u \in \mathcal{U}$.

Let $f \in C_0(\mathcal{G}, \mathbb{R}^D)$. Next we show that $f$ must be uniformly continuous. Let $\epsilon > 0$. By the fact that $f$ vanishes at infinity, there exists a compact domain $\mathcal{K} \subset \mathbb{R}^{N \times 3}$ such that for every $x \notin \mathcal{K}$ we have $|f(x)| < \epsilon/2$. By the fact that $\mathcal{U} \times \mathcal{K}$ is compact and $\kappa$ continuous, $\kappa$ is uniformly continuous on $\mathcal{U} \times \mathcal{K}$. Hence, there exists $\delta_\epsilon > 0$ such that every $g, g' \in \mathcal{K}$ with $d(g, g') < \delta_\epsilon$ satisfy $d(f(g), f(g')) < \epsilon$.

Let $\kappa'$ be the compact space consisting of all point of distance less or equal to $\delta_\epsilon$ from $\mathcal{K}$. There exists $0 < \delta'_\epsilon < \delta_\epsilon$ such that every $g, g' \in \mathcal{K}'$ with $d(g, g') < \delta'_\epsilon$ satisfy $d(f(g), f(g')) < \epsilon$.

Now, let $g, g' \in \mathcal{G}$ satisfy $d(g, g') < \delta'_\epsilon$. If one of the point $g$ or $g'$ lies outside $\mathcal{K}'$, then both of them lie outside $\mathcal{K}$, so

$$d(f(g), f(g')) \leq \|f(g)\| + \|f(g')\| < \epsilon.$$

Otherwise, both lie in $\mathcal{K}'$, so $d(f(g), f(g')) < \epsilon$. Both cases together mean that $f$ is uniformly continuous.

As a result of uniform continuity, $\{g \mapsto f(\kappa_u(g))\})u \in \mathcal{U}$ is equicontinuous, and this is a setting of continuous prior maximization.

In fact, this analysis shows that whenever $\mathcal{G} = \mathcal{K}$ and $\kappa$ is continuous in $(u, g)$, then it's corresponding $\rho$ is continuous prior maximization.

### F.2 Continuous to Discrete Images with Rotations and Other Image Transformations

Consider the "continuous" space of images $\mathcal{G} = \mathcal{L}_2(\mathbb{R}^2)$ and the discrete space $\mathcal{K} = \mathbb{R}^{N \times N}$ of images of $N \times N$ pixels.

Let $\mathcal{U}$ be the unit circle. For $g \in \mathcal{G}$ and $u \in \mathcal{U}$ let $\pi(u)g$ be the rotation of the image $g$ by angle $u$. To define the discretizing mapping $P : \mathcal{L}_2(\mathbb{R}^2) \to \mathbb{R}^{N \times N}$, consider the partition of $[-1, 1]$ into the $N$ intervals

$$I_n = [-1 + 2n/N, -1 + 2(n+1)/N), \quad n = 0, \dots N - 1.$$

Consider the closed linear subspace $\mathcal{D}(\mathbb{R}^2) \subset \mathcal{L}_2(\mathbb{R}^2)$ consisting of images that are zero outside $[-1, 1]^2$ and piecewise constant on the squares $\{I_n \times I_m\}_{n,m=0}^{N-1}$. Now, $P$ is the operator that takes $g \in \mathcal{L}_2(\mathbb{R}^2)$ first orthogonally projects it upon $\mathcal{D}(\mathbb{R}^2)$ to get $g'$, and then returns

$$P(g) = \{g'(-1 + 2n/N, -1 + 2m/N)\}_{n,m=0}^{N-1} \in \mathbb{R}^{N \times N}.$$

Define the mapping $\kappa_u$ as follows: $\kappa_u(g) = P(\pi(u)g)$. By the fact that $\pi_u$ is an isometry for every $u \in \mathcal{U}$ and $P$ is non-expansive (as an orthogonal projection), $\kappa_u : \mathcal{G} \to \mathbb{R}^{N \times N}$ is Lipschitz 1 for every $u$. Hence, $\{g \mapsto f(\kappa_u(g))\}_{u \in \mathcal{U}}$ is equicontinuous.

As a result, the corresponding prior minimization is a continuous prior minimization.

This setting can be extended to other image deformation based on diffeomorphisms of the domain $\mathbb{R}^2$, parameterized by compact spaces $\mathcal{U}$, For example, one can take dilations up to some uniformly bounded scale. More generally, one can consider a compact set $\mathcal{U}$ of matrices in $\mathbb{R}^{2 \times 2}$, and define $\pi(u)g(\mathbf{x}) = g(\S\mathbf{U})$ for $\mathbf{U} \in \mathcal{U}$, $g \in L_2(\mathbb{R}^2)$ and $\mathbf{x} \in \mathbb{R}^2$.

### F.3 Discrete to Discrete Images with Rotations and Other Image Transformations

One can replace $\mathcal{G}$ by $\mathcal{D}(\mathbb{R}^2)$ in the above analysis. Since now $\mathcal{D}(\mathbb{R}^2)$ is compact, $\rho$ must be a continuous prior maximization.

The above examples can be naturally extended to additional image transformations, like translations and dilations. Notably, translations and dilations do not form a compact group, but still they satisfy the conditions of our theory, which requires no compactness assumptions.

## G Experimental Details

In this section, we describe the experimental setups and implementation details used in Section 5.

### G.1 Illustrative Toy Problems: Grid Signal Orientation Tasks

**Toy Problem and Experimental Setup.** We consider a square grid on the torus with a 2-channel signal. The first channel contains a sinusoidal signal aligned with the $x$-axis, given by $\sin(2\pi x/T)$.

Table 4: Datasets statistics.

| Dataset | # Graphs | # Classes | Avg.# Nodes | Avg.# Edges |
|---|---|---|---|---|
| MUTAG | 188 | 2 | 17.93 | 19.79 |
| PTC | 344 | 2 | 14.29 | 14.69 |
| ENZYMES | 600 | 6 | 32.63 | 64.14 |
| PROTEINS | 1113 | 2 | 39.06 | 72.82 |
| NCI1 | 4110 | 2 | 29.87 | 32.30 |
| ogbg-molhiv | 41127 | 2 | 25.5 | 27.5 |
| ogbg-molpcba | 437929 | 128 | 26.0 | 28.1 |
| ogbg-ppa | 158100 | 37 | 243.4 | 2266.1 |

The second channel depends on the class label: in Class 0 it is aligned with the $x$-axis, while in Class 1 it encodes a sinusoidal signal along the $y$-axis, $\sin(2\pi y/T)$. Independent Gaussian noise with variance $\sigma^2$ is added to each channel. In addition, it introduces an additional challenge by spatially restricting the support of the channels. The grid is vertically partitioned into two disjoint halves. The first channel is supported only on the left half. The second channel is supported only on the right half. The task is to decide if the frequency at the two channels is in the same orientation. The grid size is fixed at $N = 40^2$, the sinusoidal period is set to $T = 20$, and the noise level is chosen as $\sigma = 0.1$. We generate 1000 samples. Evaluation is carried out using 10-fold cross-validation.

**Competing Methods.** The competing methods include: MLP, GCN, GAT, GIN, ChebNet, NLSF, $S^2$GNN, FA+GIN, and OAP+GIN.

**Hyperparameters.** We use a three-layer network with a hidden feature dimension chosen from $\{32, 64, 128\}$ and ReLU activation functions. The learning rate is selected from $\{10^{-3}, 10^{-4}, 10^{-5}\}$. Batch size 100. All models are implemented in PyTorch and optimized with the Adam optimizer (Kingma & Ba, 2014). Experiments are conducted on an Nvidia DGX A100. The output of the GNN is then passed to an MLP.

### G.2 Graph Classification on TUDataset

**Datasets and Experimental Setup.** We consider five graph classification benchmarks from TU-Dataset (Morris et al., 2020): MUTAG, PTC, ENZYMES, PROTEINS, and NCI1. The dataset statistic is reported in Table 4. Following the random split protocol (Ma et al., 2019; Ying et al., 2018; Zhang et al., 2019b), we partition the dataset into 80% training, 10% validation, and 10% testing. Results are averaged over 10 random splits, with mean accuracy and standard deviation reported.

**Competing Baselines.** We evaluate on medium-scale graph classification benchmarks from TU-Dataset, using the same set of competing methods as in grid signal orientation tasks. The baselines include MLP, GCN, GAT, GIN, ChebNet, NLSF, $S^2$GNN, FA+GIN, and OAP+GIN.

**Hyperparameters.** The hidden dimension is set to be 128. The models are implemented using PyTorch, optimized with the Adam optimizer (Kingma & Ba, 2014). An early stopping strategy is applied, where training halts if the validation loss does not improve for 100 consecutive epochs. The hyperparameters are selected through a grid search, conducted via Optuna (Akiba et al., 2019), with with the learning rate and weight decay explored in the set $\{1e^{-2}, 1e^{-3}, 1e^{-4}\}$, the pooling ratio varying within $[0.1, 0.9]$ with step 0.1, and the number of layers ranging from 2 to 9 in a step size of 1. The output representations are then passed into an MLP, and predictions are obtained by optimizing a cross-entropy loss function. Experiments are conducted on an Nvidia DGX A100.

### G.3 Molecular Classification on OGB Datasets

**Datasets and Experimental Setup.** We evaluate on larger-scale benchmarks from the Open Graph Benchmark (OGB) dataset (Hu et al., 2020) for classification tasks, including ogbg-molhiv, ogbg-molpcba, and ogbg-ppa. Dataset statistics are summarized in Table 4 The evaluation settings are followed by the OGB protocol (Hu et al., 2020).

**Competing Baselines.** For large-scale graph classification, we include GCN, GIN, GatedGCN, PNA, GraphTrans, SAT, GPS, SAN, and the canonicalization-based variant OAP+GatedGCN as the competing methods. These approaches have previously demonstrated strong performance on OGB benchmarks, and their reported results are taken from prior work[4].

**Hyperparameters.** The models are implemented in PyTorch and optimized with the Adam optimizer, with training capped at a maximum of 1000 epochs and controlled by an early stopping criterion. The hidden dimension is selected from the set $\{128, 256, 512\}$, while the number of layers varies from 2 to 10 in steps of 1. Dropout rates are explored within the range $[0, 0.1, \ldots, 0.5]$, the learning rate is tuned within the interval $[0.0001, 0.001]$, and the warmup is set as 5 or 10. Additionally, the batch size is chosen from $\{32, 64, 128, 256\}$ and the weight decay is chosen from $\{10^{-4}, 10^{-5}, 10^{-6}\}$. All hyperparameters are tuned using Optuna (Akiba et al., 2019). The experiments are conducted on an NVIDIA A100 GPU.

### G.4  ModelNet40 Point Cloud Classification

**Datasets and Experimental Setup.** Our evaluation for point cloud classification was carried out on the ModelNet40 dataset (Wu et al., 2015), which consists of 40 object categories and a total of 12,311 3D models. Following prior studies (Wang et al., 2019; Deng et al., 2021), we allocated 9,843 models for training and 2,468 models for testing in the classification task. For each model, 1,024 points were uniformly sampled from its mesh surface, using only the $xyz$ coordinates of the sampled points. We apply on-the-fly rotation augmentation during training, following Esteves et al. (2018); Deng et al. (2021), such that the dataset size remains unchanged. At test time, each example is rotated by an arbitrary rotation. Note that on-the-fly augmentation essentially changes the training data distribution during the learning process. The purpose of comparing under rotation protocols is to assess a model's invariance to rotational changes.

**Competing Baselines.** For point cloud classification tasks, we compare our anisotropic geometric method with point cloud approaches, including PointNet and DGCNN architectures, as well as equivariant models based on the vector neuron framework, i.e., VN-PointNet and VN-DGCNN. We further include canonicalization baselines, CN-PointNet and CN-DGCNN, and traditional augmentation baselines in which the training set is expanded with pre-generated rotations (PointNet-Aug, DGCNN-Aug) with a factor of five ($\times 5$). The experimental results of PointNet, DGCNN, VN-PointNet, and VN-DGCNN are taken from Wang et al. (2019); Deng et al. (2021).

**Hyperparameters.** We follow the published hyperparameters and training protocol of PointNet and DGCNN. For PointNet, we uses identical channel widths to PointNet (64, 64, 64, 128, 1024). We use Adam optimizer with learning rate 0.001 and batch size 32 with a weight decay $1 \times 10^{-4}$ and dropout 0.3. For DGCNN, each input comprises 1,024 uniformly sampled points, and the k-NN graph uses neighborhood size $k = 20$. DGCNN uses four EdgeConv layers (with per-layer MLPs of sizes 64, 64, 128, 256). We train with stochastic gradient descent (initial learning rate 0.1) and apply a cosine annealing schedule of 0.001. Training runs for 250 epochs with a batch size of 32, and we use a dropout rate of 0.5 in the fully connected layers.

## H  Additional Experimental Results

In this section, we present additional results of our adaptive canonicalization, including experimental trade-offs of sampling-based and optimization-based construction, anisotropic nonlinear spectral filters for node-level representation, and out-of-sample rotation generalization for point clouds.

### H.1  Sampling-based VS Optimization-based Implementation

To evaluate the trade-offs between sampling-based and our sample-and-refine (optimization-based) implementation, we conduct experiments on the TUDataset graph classification benchmarks. Table 5 reports the classification performance of TUDaset under sampling-based and optimization-based adaptive caninocalization. We see that the optimization-based implementation consistently performs

---

[4]https://ogb.stanford.edu/docs/leader_graphprop/

better than the sampling-based one. While increasing the sampling candidates (from $1\times$ to $5\times$ or $10\times$) improves the performance, the sample-and-refine strategy is more memory-efficient than massive sampling approaches. Rather than storing and evaluating hundreds of rotation matrices simultaneously, it processes a smaller working set through iterative refinement, reducing memory pressure (Li et al., 2022). In terms of computation time, the sampling-based method grows linearly with the number of candidates, while the optimization-based method add a small overhead to the inner steps. We see that in practice, a modest refinement (a few steps) surpasses the accuracy of large sampling budgets at a lower time, offering a better accuracy-time trade-off.

Table 5: Comparison of sampling-based vs. optimization-based adaptive canonicalization. Classification accuracy across TUDataset. Sampling methods use different numbers of random candidates, while the optimization approach combines sampling with local refinement via gradient descent.

|  | MUTAG | PTC | ENZYMES | PROTEINS | NCI1 |
|---|---|---|---|---|---|
| A-NLSF (sampling) | $84.23_{\pm1.4}$ | $69.05_{\pm1.8}$ | $70.10_{\pm1.5}$ | $82.94_{\pm1.6}$ | $80.64_{\pm1.2}$ |
| A-NLSF (sampling $\times5$) | $85.17_{\pm1.3}$ | $72.21_{\pm1.3}$ | $71.59_{\pm1.0}$ | $83.57_{\pm1.8}$ | $80.92_{\pm1.3}$ |
| A-NLSF (sampling $\times10$) | $85.54_{\pm1.3}$ | $72.78_{\pm1.5}$ | $72.42_{\pm1.2}$ | $85.03_{\pm1.2}$ | $80.94_{\pm0.8}$ |
| A-NLSF (optimization) | $87.94_{\pm0.9}$ | $73.16_{\pm1.2}$ | $73.01_{\pm0.8}$ | $85.47_{\pm0.6}$ | $82.01_{\pm0.9}$ |

## H.2 NODE-LEVEL ANISOTROPIC NONLINEAR FILTERS

We introduce the adaptive canonicalization applied to spectral graph neural networks for learning graph-level representation in Section 4.1 and Appendix A.3. The adaptive canonicalization can also be applied to node-level representation, where the node-level representation proceeds by mapping the input signal to the spectral domain (Mallat, 2002) in a band-wise manner with an oriented basis within each band's eigenspace, and performing a synthesis step that transforms the learned coefficients back to the node domain.

On a square grid, each $x$ Fourier mode has a corresponding $y$ Fourier mode of the same response. Therefore, standard spectral methods are inherently isotropic as they cannot distinguish between horizontal and vertical directional information. On the other hand, adaptive canonicalization is anisotropic and our method can learn distinct orientations. The resulting spatial operator can therefore implement any directional filter that a convolutional neural network can achieve (Shannon, 2006; LeCun & Bengio, 1998; Freeman et al., 1991; Dagès et al., 2025).

In graph-level tasks, the canonicalized node-level embeddings can serve as the intermediate representation from which graph-level features are derived. Specifically, the resulting node embeddings can be aggregated through standard pooling operations to have a graph-level representation. We evaluate the node-to-graph construction on TUDataset for graph classification tasks. The results are summarized in Table 6. We see that the node-to-graph construction achieves performance closely aligned with, and in some cases approaching, that of the direct graph-level canonicalization. We attribute the slightly worse performance to the potential pooling loss.

Table 6: Graph classification performance on TUDataset using adaptive canonicalization. Comparison between direct graph-level representations (Graph) and node-to-graph constructions (Node-to-graph).

|  | MUTAG | PTC | ENZYMES | PROTEINS | NCI1 |
|---|---|---|---|---|---|
| Node-to-graph | $87.02_{\pm1.1}$ | $72.14_{\pm1.5}$ | $71.26_{\pm1.2}$ | $84.87_{\pm0.8}$ | $81.64_{\pm1.2}$ |
| Graph | $87.94_{\pm0.9}$ | $73.16_{\pm1.2}$ | $73.01_{\pm0.8}$ | $85.47_{\pm0.6}$ | $82.01_{\pm0.9}$ |

## H.3 OUT-OF-SAMPLE ROTATION GENERALIZATION IN POINT CLOUDS

We adopt the $z/\mathcal{SO}(3)$ protocol (Esteves et al., 2018; Deng et al., 2021): training with on-the-fly azimuthal rotations ($z$-axis) augmentation, and evaluation applies under arbitrary rotations to each test shape. In this setting, we assess out-of-sample rotation generalization by constraining training data rotations while testing on the full rotation group. The classification performance on ModelNet40 under $z/\mathcal{SO}(3)$ protocol is reported in Table 7. Standard PointNet and DGCNN drop sharply under this shift. Equivariant vector-neuron variants recover much of the loss, and canonicalization baselines are comparable. Our adaptive canonicalization outperforms both equivariant architecture and canonicalization baselines in both backbones.

Table 7: Classification accuracy on ModelNet40 under $z/\mathcal{SO}(3)$ protocol.

|  | PointNet | DGCNN | VN-PointNet | VN-DGCNN | CN-PointNet | CN-DGCNN | AC-PointNet | AC-DGCNN |
|---|---|---|---|---|---|---|---|---|
| Accuracy | 19.6 | 33.8 | 77.5 | 89.5 | 79.6 | 88.8 | 81.4 | 91.8 |

## H.4 ABLATION STUDIES

We conduct ablation studies on the spectral band partitioning and the choice of GSO for A-NLSF, as well as on the impact of different point cloud backbones in our anisotropic point cloud networks.

### H.4.1 SPECTRAL BAND PARTITION

In our experiment, we adopt a dyadic partitioning scheme (see Appendix A.3). In this section, we conduct an ablation using a uniform partitioning of the eigenvalues with the same number of bands and report the graph classification performance in Table 8. We see that using the dyadic partitions performs better than using the uniform partition. tion provided by dyadic bands, which could more effectively isolate band-wise unitary actions that commute with the chosen GSO. We also note that spectral band design can be realized in more flexible and expressive ways, for example, through attention as in SpecFormer (Bo et al., 2023). Investigating such learned or adaptive band-selection strategies is an important direction for future work and may further strengthen our adaptive canonicalization framework.

Table 8: Graph classification performance using uniform and dyadic spectral band partitioning.

|  | MUTAG | PTC | ENZYMES | PROTEINS | NCI1 |
|---|---|---|---|---|---|
| Uniform | $81.36_{\pm 1.2}$ | $66.20_{\pm 0.8}$ | $62.84_{\pm 1.4}$ | $80.01_{\pm 1.3}$ | $79.62_{\pm 1.0}$ |
| Dyadic | $87.94_{\pm 0.9}$ | $73.16_{\pm 1.2}$ | $73.01_{\pm 0.8}$ | $85.47_{\pm 0.6}$ | $82.01_{\pm 0.9}$ |

### H.4.2 GRAPH SHIFT OPERATOR

We evaluate the graph Laplacian as an alternative GSO. Table 9 reports the graph classification performance of A-NLSF when instantiated with the graph Laplacian versus the normalized graph Laplacian. We observe that using the normalized graph Laplacian in our method yields better performance than the graph Laplacian. We attribute this to the properties of the normalized Laplacian: (i) the normalized Laplacian removes degree-related scaling effects, leading to a comparable spectral domain across graphs with different degree distributions, and (ii) its eigenvalues lie in the fixed interval $[0, 2]$, providing a controlled and interpretable frequency range, and making dyadic partitioning better aligned across different graphs.

Table 9: Graph classification performance of A-NLSF with different GSO.

|  | MUTAG | PTC | ENZYMES | PROTEINS | NCI1 |
|---|---|---|---|---|---|
| Graph Laplacian | $83.76_{\pm 1.0}$ | $67.23_{\pm 1.4}$ | $62.60_{\pm 1.2}$ | $82.64_{\pm 1.6}$ | $78.59_{\pm 0.8}$ |
| Normalized graph Laplacian | $87.94_{\pm 0.9}$ | $73.16_{\pm 1.2}$ | $73.01_{\pm 0.8}$ | $85.47_{\pm 0.6}$ | $82.01_{\pm 0.9}$ |

## H.5 SENSITIVITY ANALYSIS

To examine the effect of different hyperparameters, we conduct a hyperparameter sensitivity study covering grid size, sinusoidal period, noise level, and hidden dimension. For each hyperparameter, we swept over a range of values while keeping all other settings fixed. The results of the sensitivity analysis are summarized in Figure 2. Overall, we observe that our method is reasonably robust. For grid size and sinusoidal period, performance remains stable across the tested ranges. For the noise level, small to moderate noise leads to similar performance, with a degradation only when the noise becomes large enough that it effectively corrupts the underlying structure of the data. For the hidden dimension, small dimensions impact the performance, but performance stabilizes once we enter a standard regime of model capacity.

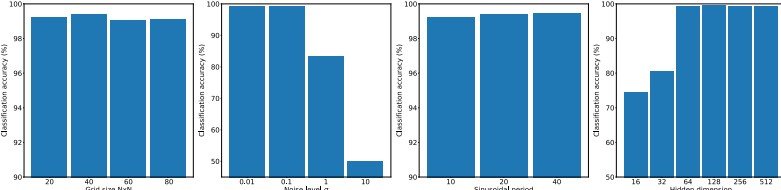

Figure 2: Hyperparameter sensitivity with respect to grid size, noise level, and hidden dimension.

### H.5.1 POINT CLOUD BACKBONES

In order to assess the impact of the backbone choice on the performance of our anisotropic point cloud networks, we extended our experiments to include two additional and widely used point cloud backbones, PointNet++ (Qi et al., 2017b) and RSCNN (Liu et al., 2019), in addition to PointNet and DGCNN reported in Table 3. We denote the corresponding variants by AC-PointNet++ and AC-RSCNN. The ablation results are reported in Table 10. We see that the choice of backbone does influence the overall point cloud classification performance. However, we observe that our adaptive canonicalization framework consistently improves the classification performance across these backbones. Moreover, when comparing methods built on the same backbone (e.g., PointNet or DGCNN), our approach outperforms equivariant models, data augmentation, and standard canonicalization (see Table 3). This indicates that our method is robust across different point cloud backbones and can further benefit from stronger backbones when they are available.

Table 10: Classification results on ModelNet40 for different point cloud backbones. Results of competing methods marked with * are taken from Deng et al. (2021); Luo et al. (2022).

|  | Accuracy |
|---|---|
| PointNet | 74.7* |
| AC-PointNet | 81.1$_{\pm 0.7}$ |
| PointNet++ | 85.0* |
| AC-PointNet++ | 87.4$_{\pm 0.4}$ |
| RSCNN | 82.6* |
| AC-RSCNN | 87.6$_{\pm 0.3}$ |
| DGCNN | 88.6* |
| AC-DGCNN | 91.6$_{\pm 0.6}$ |

### H.6 TRUNCATION CANONICALIZATION WITH A PRETRAINED CLASSIFIER

We introduce in Appendix A.5 an application of our adaptive canonicalization on truncation prior maximization. We now illustrate the applicability of this setup with a pretrained image classifier. Specifically, we take a ResNet-18 (He et al., 2016) pretrained on ImageNet (Deng et al., 2009). We freeze the backbone, and train only the classifier using the CIFAR-10 or CIFAR-100 (Krizhevsky et al., 2009) training set. The experiment is conducted with ten independent runs, and the resulting image classification performance is reported in the Table 11. We see that truncation-based prior maximization improves classification performance over the standard vanilla baseline. This implies that our method enables the model to adaptively select a canonical truncation that enhances downstream performance. In addition, we observe that the selected canonical crops tend to tightly focus on the main object while discarding background and irrelevant context. It matches the intuition behind our prior maximization: by optimizing over the truncation family, the model chooses a representative transformed image that best aligns with its prior over the class. This experiment demonstrates that our adaptive canonicalization framework can be instantiated with a truncation symmetry and benefit from off-the-shelf pretrained models. It also highlights the potential of transformation families as a practical way to improve pretrained models via adaptive canonicalization.

Table 11: Image classification accuracy on CIFAR-10 and CIFAR-100 using a ResNet18 pretrained on ImageNet, with and without truncation canonicalization.

|  | CIFAR-10 | CIFAR-100 |
|---|---|---|
| Vanilla | 72.09$_{\pm 1.0}$ | 56.94$_{\pm 0.8}$ |
| Truncation canonicalization | 74.92$_{\pm 0.6}$ | 60.38$_{\pm 0.5}$ |

### H.7 COMPUTATIONAL REQUIREMENT COMPARISON

Table 12 the training time per epoch with the number of parameters. We see that A-NLSF uses a similar number of parameters as the other methods and fewer than the spectral method. Its computational

requirements are comparable to the other methods and does not rely on a significantly larger training budget than the competing methods.

Table 12: Running time per epoch(s)/number of parameters.

|  | MUTAG | PTC | ENZYMES | PROTEINS | NCI1 |
|---|---|---|---|---|---|
| MLP | 0.07/105K | 0.10/114K | 0.13/125K | 0.37/129K | 1.01/134K |
| GCN | 0.40/116K | 0.66/120K | 0.81/137K | 1.92/142K | 5.84/149K |
| GAT | 0.62/138K | 0.87/149K | 0.96/154K | 2.34/159K | 4.93/167K |
| GIN | 0.14/105K | 0.37/106K | 0.52/107K | 0.94/106K | 1.97/121K |
| ChebNet | 0.79/185K | 1.25/189K | 1.72/191K | 3.64/217K | 11.52/245K |
| FA+GIN | 0.57/120K | 1.04/123K | 1.35/126K | 2.55/130K | 4.31/142K |
| OAP+GIN | 0.22/105K | 0.39/104K | 0.57/109K | 1.21/110K | 2.36/124K |
| A-NLSF | 0.44/132K | 0.89/140K | 1.29/145K | 2.20/148K | 4.37/151K |

## H.8 TRAINING STABILITY

To quantify the training stability of our method, we track the canonical rotations of a subset of 1500 randomly chosen training examples in the point cloud classification experiment. At each epoch, we measure the mean geodesic distance on $\mathcal{SO}(3)$ between the canonicalizations between consecutive epochs. Figure 3 reports the mean geodesic distance between epochs. We observe that this distance decreases during the training and then

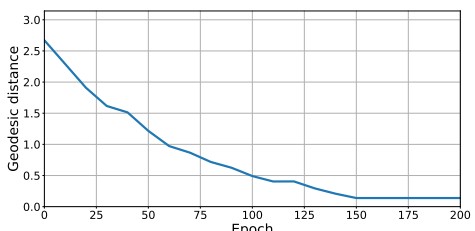

Figure 3: Mean geodesic distance on $\mathcal{SO}(3)$ between the canonicalizations between consecutive epochs.

remains stable, indicating that the canonical representatives stabilize with no rapid switching.

## H.9 CANONICALIZED POINT CLOUDS

Figure 4 shows the canonicalized point clouds for the chair class in the point cloud classification experiment. We randomly select 20 examples from this class and visualize them after applying the optimal transformations. We observe that the examples in this class share a similar orientation after canonicalization.

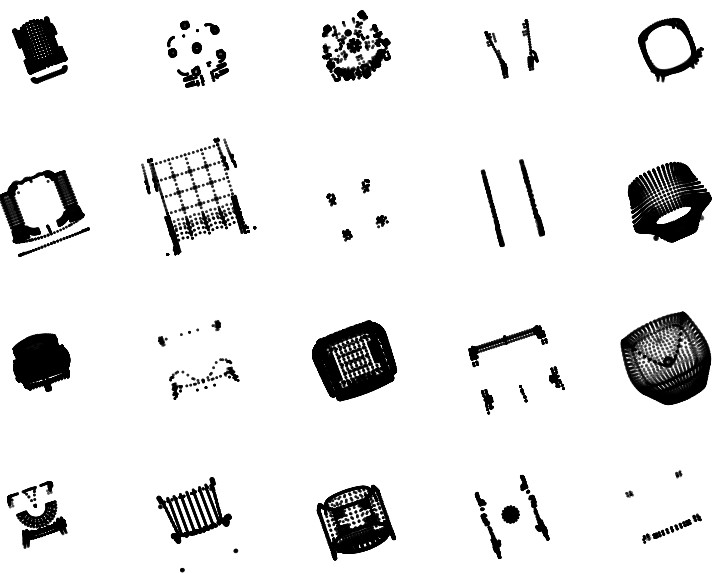

Figure 4: The canonicalized point clouds for the chair class.

## H.10 SHAPENET PART SEGMENTATION

To expand our experimental study on point cloud data, we further conduct experiments on the ShapeNet part segmentation benchmark (Yi et al., 2016). The dataset consists of 16 shape categories annotated with a total of 50 parts, where each category is labeled with between two and six parts. Note that our prior maximization adaptive canonicalization method was naturally suited to classification tasks. Extending it to the segmentation task is not trivial, as the segmentation task requires predicting a label for each point in the point cloud, and applying prior maximization to each point would be computationally inefficient. Therefore, in order to adapt our adaptive canonicalization to the segmentation task, we consider the adaptive canonicalization with the minimal entropy prior summing over nodes. This modification preserves the spirit of the adaptive canonicalization while making it compatible with per-point prediction. Table 13 reports the segmentation performance. For the PointNet backbone, we see that, similar to the point cloud classification task, the entropy-based adaptive canonicalization yields advantageous

Table 13: Part segmentation performance on the ShapeNet part dataset. The metric is reported with the average category mean IoU Results of competing methods marked with * are taken from Deng et al. (2021); Kaba et al. (2023).

| | |
|---|---|
| PointNet | 62.3* |
| DGCNN | 78.6* |
| VN-PointNet | 72.8* |
| VN-DGCNN | 81.4* |
| CN-PointNet | $73.6_{\pm 1.1}$ * |
| CN-DGCNN | $78.5_{\pm 0.9}$ * |
| AC-PointNet | $76.0_{\pm 0.6}$ |
| AC-DGCNN | $80.9_{\pm 0.7}$ |

segmentation performance compared to equivariant architectures and standard canonicalization baselines. For the DGCNN backbone, our method attains performance comparable to equivariant architectures while outperforming existing canonicalization methods. These results demonstrate that our approach has potential beyond classification. Note that one of the main contributions of our work is to construct continuous and symmetry-respecting models. In the entropy prior adaptive canonicalization, the continuity property is not straightforward. We plan to investigate the continuity properties of this adapted approach in future work.

