# OpenReview forum: "Adaptive Canonicalization with Application to Invariant Anisotropic Geometric Networks"
_ICLR.cc/2026/Conference — ICLR 2026 Poster_

### Official Review · Reviewer_48pC · 2025-10-27

**Soundness:** 3
**Presentation:** 3
**Contribution:** 4
**Rating:** 8
**Confidence:** 3

**Summary:**

The paper proposes *adaptive canonicalization*, a form of canonicalization that depends on the input and network to alleviate concerns about continuity (i.e., slight changes in the input do not change the canonical form) and about universal approximators. The paper also shows how to use this canonicalization for molecular, protein, and point cloud classification tasks.

**Strengths:**

- The paper is well-structured, and the motivation is appropriately highlighted.
- The paper constructively develops the theoretical background to prove the claims.
- The experiments consistently show improved performance of using adaptive canonicalization for wide-variety of tasks.

**Weaknesses:**

I have a few minor questions, which are highlighted in the next section.

**Questions:**

- No equivariant tasks (i.e., segmentation) were studied within the paper, which is one of the more relevant tasks where such a method should be evaluated.
- One of the primary benefits of canonicalization-based methods is the adaptation of large pre-trained models in a zero-shot manner to avoid significant re-training cost. However, the experimental designs proposed in the paper require re-design and re-training of the downstream network. It would be great if the authors could add a discussion of this in the paper, along with potential ways to use adaptive canonicalization with pre-trained models.
- Could you discuss and contrast your proposed method with the following lines of work [1, 2]?
- How does the distribution of canonical samples look? Does it retain some structure or orientation (or specific rotations, e.g., for images, say for a horse, we get for humans, as described, with legs down), or is there no such orientation once the canonicalization is *adaptive*?
- Can you provide the average over multiple runs in Table 3?
- Computational requirements are not discussed, i.e., what was the training budget (time and compute requirements) for A-NLSF vs other methods? It would be unfair if the proposed method requires a significantly higher training budget.
- How do the theoretical results in this paper compare with the results of [3]?

1. Equi-Tuning: Group Equivariant Fine-Tuning of Pretrained Models. Basu et al., AAAI 2023.
2. Efficient Equivariant Transfer Learning from Pretrained Models. Basu et al., NeurIPS 2023.
3. On the Universality of Invariant Networks. Maron et al., ICML 2019.

---

> ### Author Response · Authors · 2025-11-28
>
> (This is comment 1 out of 3.)
>
> We thank the reviewer for their valuable comments and for recognizing the motivation and the theoretical grounding of our method. In terms of the questions:
>
> >Q1: No equivariant tasks (i.e., segmentation) were studied within the paper, which is one of the more relevant tasks where such a method should be evaluated.
>
>
> **New ShapeNet part segmentation experiment**
>
> We thank the reviewer for encouraging us to evaluate our method on the segmentation task.  Following the recommendation, we conducted experiments on the ShapeNet part segmentation benchmark in the revised manuscript (Appendix G.10). Our prior maximization adaptive canonicalization method was naturally suited to classification tasks. Extending it to the segmentation task is not trivial, as the segmentation task requires predicting a label for each point in the point cloud,  and applying prior maximization to each point would be computationally inefficient. Therefore,  in order to adapt our adaptive canonicalization to the segmentation task, we consider the adaptive canonicalization with the minimal entropy prior summing over nodes. This modification preserves the spirit of the adaptive canonicalization while making it compatible with per-point prediction. Table 13 reports the segmentation performance. For the PointNet backbone, we see that, similar to the point cloud classification task, the entropy-based adaptive canonicalization yields advantageous segmentation performance compared to equivariant architectures and standard canonicalization baselines. For the DGCNN backbone, our method attains performance comparable to equivariant architectures while outperforming existing canonicalization methods. These results demonstrate that our approach has potential beyond classification. Note that one of the main contributions of our work is to construct continuous and symmetry-respecting models. In the entropy prior adaptive canonicalization, the continuity property is not straightforward. We plan to investigate the continuity properties of this adapted approach in future work.
>
> ---
>
> >Q2: One of the primary benefits of canonicalization-based methods is the adaptation of large pre-trained models in a zero-shot manner to avoid significant re-training cost. However, the experimental designs proposed in the paper require re-design and re-training of the downstream network. It would be great if the authors could add a discussion of this in the paper, along with potential ways to use adaptive canonicalization with pre-trained models.
>
> **Application to pretrained network**
>
> We thank the reviewer for raising this important question. We would like to point out that our setting and theoretical results also apply to pretrained networks. In the revised manuscript (p.8, l.429), we now state this explicitly and clarify that the adaptive canonicalization framework can be applied with pretrained networks on a downstream task. To illustrate this applicability, we added an experiment on truncation canonicalization with a pretrained classifier in the revised manuscript (see Appendix E.5 for contrucation and Appendix G.6 for experiment). Specifically, we take a ResNet-18 pretrained on ImageNet, freeze the backbone, and train only a linear classifier as the last layer using the training set of a new dataset(i.e., CIFAR-10 or CIFAR-100). The resulting classification performance is reported in the Table 11. We see that truncation-based prior maximization improves classification performance over the standard vanilla baseline. This implies that our method enables the model to adaptively select a canonical truncation that enhances downstream performance. By optimizing over the truncation family, the model chooses a representative transformed image that best aligns with its prior over the class. This experiment demonstrates that our adaptive canonicalization can be instantiated with a truncation symmetry and benefit from off-the-shelf pretrained models. We thank the reviewer again for encouraging us to expand on this point.

---

> ### Author Response · Authors · 2025-11-28
>
> (This is comment 2 out of 3.)
>
> >Q3: Could you discuss and contrast your proposed method with the following lines of work [1, 2]?
>
> **Equi-Tuning and $\lambda$-equitune**
>
> We thank the reviewer for bringing these two works to our attention. After carefully reading both papers, we now provide a more detailed discussion in the revised manuscript. Regarding Equi-Tuning [1],  it is about frame averaging and involves averaging the whole orbit of each datapoint. We briefly list here the comparison of our method and the frame averaging method. Frame averaging achieves equivariance to group symmetries by averaging a network's output over a set of group transformations (known as a "frame").  It is built on the classical group averaging operator, which guarantees symmetries by summing a function over all group elements. However, it requires a careful selection of a suitable frame. In addition, frame averaging uses a fixed set of transformations independent of the input or task. In contrast, our adaptive canonicalization learns the optimal transformation for each input in a data- and network-dependent way, yielding symmetry preserving continuous functions. As for $\lambda$-equitune [2], this is an earlier work related to weighted frame averaging, but it does not study continuity preservation. In the revised version, we now discuss  in more detail weighted frame averaging and $\lambda$-equitunein Appendix A.2. We thank the reviewer again for directing us to these relevant works.
>
> >Q4: How does the distribution of canonical samples look? Does it retain some structure or orientation (or specific rotations, e.g., for images, say for a horse, we get for humans, as described, with legs down), or is there no such orientation once the canonicalization is adaptive?
>
> **Canonicalized point clouds**
>
> We thank the reviewer for this insightful comment. Following the question, we took 20 examples from the chair class in the point cloud experiment and visualized them after applying the optimal transformations. Figure 4 shows the resulting canonicalized point clouds for the chair class. We observe that the examples in this class share a similar orientation after canonicalization.
>
> >Q5: Can you provide the average over multiple runs in Table 3?
>
> **Average over multiple runs on ModelNet40**
>
> We thank the reviewer for this helpful suggestion. While the ModelNet40 benchmark employs a fixed train/test split and existing works (e.g., PointNet, PointNet++, DGCNN) typically report single-run accuracy without standard deviations, we agree that reporting averages and variability over multiple runs is valuable. Even with a fixed split, results could vary across different random initializations and configurations.  In response, we updated the table to include the mean with standard deviation for our method and for the augmentation baseline, computed over five runs with different random seeds. We also added in the table caption that the results of competing methods are taken from prior work. The updated results demonstrate that our method remains stable. We are currently conducting an additional experiment with multiple train/test splits to facilitate a more comprehensive comparison. These results will be included in the final version of the paper.
>
> >Q6: Computational requirements are not discussed, i.e., what was the training budget (time and compute requirements) for A-NLSF vs other methods? It would be unfair if the proposed method requires a significantly higher training budget.
>
> **Computational requirements**
>
> We thank the reviewer for raising this question. In revision, we now report in Table 12 the training time per epoch with the number of parameters. We see that A-NLSF uses a similar number of parameters as the other methods and fewer than the spectral method. Its computational requirements are comparable to the other methods and does not rely on a significantly larger training budget than the competing methods.

---

> ### Author Response · Authors · 2025-11-28
>
> (This is comment 3 out of 3.)
>
> >Q7: How do the theoretical results in this paper compare with the results of [3]?
>
> **Comparison with Maron et al. (2019)**
>
> We thank the reviewer for encouraging us to compare our method with the work of Maron et al. (2019). In their work,  the authors study the  universal approximation property for an equivariant architecture (see Appendix A.3 for a discussion of related equivariant architectures). We view that our approach is very different. The canonicalization approach drops the requirement for the network to be equivariant, and instead uses standard symmetryless networks like standard multilayer perceptron. The way invariance is achieved by canonicalization methods is to transform the data in such a way that each datapoint in an orbit is always mapped to the same point on the orbit. We explain this construction in detail in the main text (see, e.g., Section 3.4). We thank the reviewer again for motivating us to compare our method with the universal approximation property for equivariant architectures.
>
> ---
>
> We thank the reviewer once again for their insightful comments, which helped us improve our paper, and hope to discuss any ongoing suggestions further.

---

### Official Review · Reviewer_6Rf6 · 2025-10-31

**Soundness:** 3
**Presentation:** 3
**Contribution:** 3
**Rating:** 6
**Confidence:** 4

**Summary:**

Canonicalization is widely used in equivariant network but it introduces discontinuities that cause many problems. This paper proposes an adaptive canonicalization via prior maximisation to address this problem. The paper also presents applications of the proposal method. The authors theoretically and empirically verify their results.

**Strengths:**

The authors aim to address a key problem in equivariant neural networks: that canonicalization can cause problematic discontinuities. They propose a new, interesting, and well-motivated method. They present solid theory to support their claims.

**Weaknesses:**

My concern is mostly about the experiments.

The performance improvement in experiments on OGB looks marginal. For the experiments on OGB, gains are only 0.0114 in ogbg-molhiv, 0.0061 in ogbg-molpcba, 0.0134 in ogbg-ppa, while the standard deviations in ogbg-molhiv are 0.0132/0.0152. The standard deviations on ModelNet40 are not reported.

No ablation study is presented. For anisotropic nonlinear spectral filters, I quoted, "our formulation partitions the spectrum into disjoint frequency bands," but no experiments were conducted on how to partition the spectrum. Also, "we consider the GSO as the normalized graph Laplacian," and uses an MLP, without rationale or empirical validation. For anisotropic point cloud networks, the authors use backbones DeepSet and DGCNN, but not validate the choice.

The experiemtns rely on very many hyperparameters but didn't report their sensitivity. In Appendix F, the authors mentioned graid size, sinusoidal period, noise level, hidden dimension, etc.

Experiments on more datasets are encouraged, such as ShapeNet.

The code is not submitted.

**Questions:**

Please see above.

---

> ### Author Response · Authors · 2025-11-28
>
> (This is comment 1 out of 3.)
>
> We thank the reviewer for their insightful comments and for recognizing the contribution of our method for addressing the discontinuities in canonicalization methods. In terms of weaknesses:
>
>
> >W1: The performance improvement in experiments on OGB looks marginal. For the experiments on OGB, gains are only 0.0114 in ogbg-molhiv, 0.0061 in ogbg-molpcba, 0.0134 in ogbg-ppa, while the standard deviations in ogbg-molhiv are 0.0132/0.0152. The standard deviations on ModelNet40 are not reported.
>
> **On the consistent improvements on OGB datasets**
>
> We thank the reviewer for the careful reading and for noting the magnitude of the improvements on the OGB benchmarks. Indeed, the absolute gains on OGB benchmarks appear to be numerically modest. However, we would like to point out that our method achieves consistent improvements across these three OGB datasets, which cover different molecular and protein tasks.  Such consistent improvements across multiple OGB benchmarks are non-trivial, suggesting that our adaptive canonicalization is contributing positively and improves generalization in these tasks. Therefore, we believe that on large-scale benchmarks like OGB, even small but systematic performance differences can be meaningful.
>
>
> To address the reviewer's concern, we changed the discussion to highlight the consistency and generality of the gains rather than their absolute magnitude in the revised manuscript (p.10, l.502). Together with the results on TUDataset, point-cloud classification, the newly added ShapeNet part segmentation (Appendix G.10), and the new experiments on truncation canonicalization (Appendix G.6), we hope this better clarifies that the main contribution of our work is a new theoretical approach: a general adaptive canonicalization framework that can be practically beneficial across diverse application domains. We thank the reviewer again for this helpful feedback, which helps us to improve the clarity and positioning of our work.
>
>
> **Standard deviations on ModelNet40**
>
> We thank the reviewer for the comment. While ModelNet40 benchmark employs a fixed train/test split and existing works (e.g., PointNet, PointNet++, DGCNN) typically report single-run accuracy without standard deviations, we agree with the reviewer that reporting standard deviations is valuable. Even with a fixed split, results could vary across different random initializations and configurations.  In response to the reviewer’s feedback, we updated the table to include the mean with standard deviation for our method and for the augmentation baseline, computed over five runs with different random seeds. We also added in the table caption that the results of competing methods are taken from prior work. The updated results demonstrate that our method remains stable. We are currently conducting an additional experiment with multiple train/test splits to facilitate a more comprehensive comparison. These results will be included in the final version of the paper.

---

> ### Author Response · Authors · 2025-11-28
>
> (This is comment 2 out of 3.)
>
> >W2: No ablation study is presented. For anisotropic nonlinear spectral filters, I quoted, "our formulation partitions the spectrum into disjoint frequency bands," but no experiments were conducted on how to partition the spectrum. Also, "we consider the GSO as the normalized graph Laplacian," and uses an MLP, without rationale or empirical validation. For anisotropic point cloud networks, the authors use backbones DeepSet and DGCNN, but not validate the choice.
>
> **Ablation studies**
>
> We thank the reviewer for highlighting the need for additional ablations. We agree that understanding the design choices is important, and in response, we expanded our experiments accordingly for ablation studies in the revision (Section G.4).
>
> - Spectral band partition:\
>    Our experiment uses a dyadic partitioning scheme (see Appendix E.3). Following the reviewer’s suggestion, we conducted an ablation using a uniform partitioning of the eigenvalues with the same number of bands and reported the graph classification in Table 8 in the revised paper. We observe that using the dyadic partitions performs better than using the uniform partition. We attribute this to the finer scale resolution provided by dyadic bands, which could more effectively isolate band-wise unitary actions that commute with the chosen GSO.
> - Graph shift operator:\
>   We thank the reviewer for suggesting the evaluation of alternative GSOs. Following the reviewer's comment, we conducted additional experiments using the graph Laplacian matrix. The corresponding graph classification results are reported in Table 9 in the revision. Across datasets, we observe that using the normalized graph Laplacian in our method yields better performance than the graph Laplacian. We attribute this to the properties of the normalized Laplacian: (i) the normalized Laplacian removes degree-related scaling effects, leading to a comparable spectral domain across graphs with different degree distributions, and (ii) its eigenvalues lie in the fixed interval $[0,2]$, providing a controlled and interpretable frequency range, and making dyadic partitioning better aligned across different graphs.
> - Point cloud backbones:\
>   We thank the reviewer for the valuable feedback. We agree that it is important to assess the impact of the backbone choice on the performance of our anisotropic point cloud networks. Following the reviewer’s suggestion, we extended our experiments to include two additional and widely used point cloud backbones, PointNet++  (Qi et al., 2017) and RSCNN  (Liu et al., 2019), in addition to PointNet and DGCNN. The corresponding ablation results are reported in Table 10 of the revised manuscript. Indeed, the choice of backbone does influence the overall performance. However, we observe that our method consistently improves the classification performance across these four backbones. Moreover, when comparing methods built on the same backbone (e.g., PointNet or DGCNN), our approach outperforms equivariant models, data augmentation, and standard canonicalization. This indicates that our method is robust across different point cloud backbones and can further benefit from stronger backbones when they are available. We thank the reviewer again for this insightful comment, which helps us to evaluate our method more comprehensively across different point cloud backbones.
>
> ---
>
> >W3: The experiemtns rely on very many hyperparameters but didn't report their sensitivity. In Appendix F, the authors mentioned graid size, sinusoidal period, noise level, hidden dimension, etc.
>
> **Sensitivity analysis**
>
> We thank the reviewer for raising this point and agree that understanding the sensitivity to hyperparameters is important. In response, we conducted a hyperparameter sensitivity study covering grid size, sinusoidal period, noise level, and hidden dimension. For each hyperparameter, we swept over a range of values while keeping all other settings fixed. The results of this analysis are summarized in Figure 2 in the revised manuscript. Overall, we observe that our method is reasonably robust. For grid size and sinusoidal period, performance remains stable across the tested ranges. For the noise level, small to moderate noise leads to similar performance, with a degradation only when the noise becomes large enough that it effectively corrupts the underlying structure of the data. For the hidden dimension, small dimensions impact the performance, but performance stabilizes once we enter a standard regime of model capacity.

---

> ### Author Response · Authors · 2025-11-28
>
> (This is comment 3 out of 3.)
>
>
>
> >W4: Experiments on more datasets are encouraged, such as ShapeNet.
>
> **New ShapeNet part segmentation and truncation canonicalization experiment**
>
> We thank the reviewer for encouraging us to evaluate our method on more datasets.  We agree that expanding our experimental study could strengthen the paper. Following the recommendation, in the revised version, we (i) conducted experiments on the ShapeNet part segmentation benchmark (Appendix G.10), and (ii) added an image truncation transformation as an additional application of our adaptive canonicalization (Appendix E.5 and Appendix G.6).
>
> - ShapeNet part segmentation:\
>     Our prior maximization adaptive canonicalization method was naturally suited to classification tasks. Extending it to the segmentation task is not trivial, as the segmentation task requires predicting a label for each point in the point cloud,  and applying prior maximization to each point would be computationally inefficient. Therefore, in order to adapt our adaptive canonicalization to the segmentation task, we consider the adaptive canonicalization with the minimal entropy prior summing over nodes. This modification preserves the spirit of the adaptive canonicalization while making it compatible with per-point prediction. Table 13 reports the segmentation performance. For the PointNet backbone, we see that, similar to the point cloud classification task, the entropy-based adaptive canonicalization yields advantageous segmentation result compared to equivariant architectures and standard canonicalization baselines. For the DGCNN backbone, our method attains performance comparable to equivariant architectures while outperforming existing canonicalization methods. These results demonstrate that our approach has potential beyond classification. Note that one of the main contributions of our work is to construct continuous and symmetry-respecting models. In the entropy prior adaptive canonicalization, the continuity property is not straightforward. We plan to investigate the continuity properties of this adapted approach in future work.
> - Truncation canonicalization:\
>     One of the advantages of our framework is that it is not restricted to symmetries defined via group actions, and allows working with more general augmentations for transforming datapoints. To illustrate this generality, we added an image truncation transformation as an additional application. The idea is that many image classes are invariant under removing uninformative regions: as long as the object of interest remains in the field of view, the class label should not change. We consider, in this case, the truncation is our "symmetry" and we apply prior maximization over this family of transformations to select a canonical truncation (see Appendix E.5 in the revision). To demonstrate the applicability of this setup, we conducted an experiment on truncation canonicalization with a pretrained classifier (see Appendix G.6 in the revision). Specifically, we take a ResNet-18 pretrained on ImageNet, freeze the backbone, and train only the classifier using the training set of a new dataset (i.e., CIFAR-10 or CIFAR-100). The resulting classification performance is reported in the Table 11. We see that truncation-based prior maximization improves classification performance over the standard vanilla baseline.  This experiment demonstrates that our adaptive canonicalization can be instantiated with a truncation symmetry and benefit from off-the-shelf pretrained models.
>
> Through these additional experiments and the new truncation-based application, we hope to show that our method is general and can be instantiated with a broad range of tasks and datasets. We thank the reviewer again for the helpful suggestion.
>
> >W5: The code is not submitted.
>
> **Code**
>
> We thank the reviewer for raising this point and agree that code availability is important. At this stage, we are still cleaning and organizing the implementation. We will release the code upon publication.
>
> ---
>
> We thank the reviewer once again for their insightful and constructive comments, which helped us improve our paper, and hope to discuss any ongoing suggestions further.

---

### Official Review · Reviewer_dHXs · 2025-11-01

**Soundness:** 3
**Presentation:** 2
**Contribution:** 2
**Rating:** 4
**Confidence:** 3

**Summary:**

This paper introduces new results on the continuity of canonicalization functions used for architecture agnostic equivariance. Based on this analysis, it proposes an optimization based method for canonicalization that respects continuity. The method is validated experimentally on spectral graph processing and point cloud classification.

**Strengths:**

- Understanding the continuity of canonicalization methods is an interesting problem given that these methods are becoming more widely used
- The theoretical results are sound and rigorous as far as I could check
- The experimental performance of the proposed method on the tasks presented is competitive
- The detailed appendix is appreciated

**Weaknesses:**

- I think generally the paper should do a better job at discussing related works beyond the introduction and appendix
    - In many canonicalization frameworks, the canonicalization is learned and already depends implicitly on the prediction network, since it is trained using gradients from the prediction network. We can define the combination of the canonicalization architecture, initialization, and optimization process as the "adaptive canonicalization", which then becomes a function of the prediction function. In that sense, I don't think "adaptive canonicalization" is really a novel concept.
    - Likewise "prior maximization adaptive canonicalization" is closely related to the optimization canonicalization method introduced in Kaba et al. 2023. Similar ideas have also been used and proposed in subsequent papers such as Schmidt & Stober, 2024, Shumaylov et al. 2024, Singhal et al. 2025. I think it is necessary to acknowledge these works better and not overstate the novelty of these ideas
- The framework is developed with a high degree of generality, which does not refer to group actions until section 2.4. However, in the end the only applications considered are really only group actions. I feel like the analysis should either be simplified to consider the specific case of group actions (with possibly the more general version in appendix) or applications outside group actions should be examined. In the current state, the amount of formalism seems overkill given the problem considered.
- I think the construction for continuous canonicalization introduced in 2.3 (proved in theorem 9) is closely related to the weighted frames of Dym et al. 2024. Can the authors comment on that and discuss connections in that section of the paper?
  - If the proposed canonicalization scheme is not a type of weighted frame, it would be great if the authors could better highlight the core ingredient that makes the scheme continuous.
- More generally, I would appreciate if the authors could discuss their theoretical results more and try to distill them into insightful conclusions
- I think section 3 on applications should be significantly rewritten to improve ease of understanding and reduce the amount of mathematical formalism. In its current state, this section that introduces applications will be quite difficult to understand for practitioners.
- Minor corrections
    - Typos:
        - line 137: "vanish at infinity is" - > "vanish at infinity if"

**Questions:**

- An important part of the implementation that was not clear to me is how the optimization is performed for each group. Solving these optimization problems is expected to be costly in general and to introduce significant overhead. This should be discussed more transparently
- Likewise, I didn't understand which priors are maximized in practice. Can the authors elaborate and clarify that in the main text.

---

> ### Author Response · Authors · 2025-11-28
>
> (This is comment 1 out of 6.)
>
> We thank the reviewer for their insightful comments, which have helped us improve the manuscript, and for recognizing the soundness and rigor of our theoretical results. In terms of the weaknesses and questions:
>
> >W1: I think generally the paper should do a better job at discussing related works beyond the introduction and appendix
> > - In many canonicalization frameworks, the canonicalization is learned and already depends implicitly on the prediction network, since it is trained using gradients from the prediction network. We can define the combination of the canonicalization architecture, initialization, and optimization process as the "adaptive canonicalization", which then becomes a function of the prediction function. In that sense, I don't think "adaptive canonicalization" is really a novel concept.
> > - Likewise "prior maximization adaptive canonicalization" is closely related to the optimization canonicalization method introduced in Kaba et al. 2023. Similar ideas have also been used and proposed in subsequent papers such as Schmidt & Stober, 2024, Shumaylov et al. 2024, Singhal et al. 2025. I think it is necessary to acknowledge these works better and not overstate the novelty of these ideas
>
> **Detailed comparison with existing literature**
>
> We thank the reviewer for this helpful comment. We acknowledge that our initial submission did not sufficiently compare our method to all existing approaches and agree with the reviewer that a thorough comparison to prior studies is essential to position our work. In the revised manuscript, we expanded the related work discussion accordingly (please see the revised PDF and the comments below), giving credit to previous works where due, and providing a point-to-point comparison. In view of the new comparisons, we maintain that our method is novel.
>
> Below, we present a detailed comparison with the suggested references and clarify that our adaptive canonicalization framework is novel.
>
>
> * **[1] Sékou-Oumar Kaba, Arnab Kumar Mondal, Yan Zhang, Yoshua Bengio, and Siamak Ravanbakhsh. Equivariance with learned canonicalization functions. In International Conference on Machine Learning, pp. 15546–15566. PMLR, 2023.**
>
>     We added the following discussion in short to the main paper (p.3, l.131) and wrote an extended comparison in the Appendix A.1.1.
>
>     The paper [1] introduced energy-based canonicalization. The idea is to learn an energy function $s$ over samples and group elements, and define the canonicalization as minimizing this energy with respect to the group, given a fixed datapoint. Specifically, the energy minimization in Equation (4) in [1] is related to our prior maximization adaptive canonicalization. Training $s$ end-to-end with the task network can indeed be seen as canonicalization that depends on the task network, if one considers the full end-to-end method. However, [1] does not give the same continuity guarantees that our approach gives. We list key differences between the works below:
>     1. In [1], the energy $s$ is not the task neural network like in our analysis, but rather some other trainable neural network.
>     2. The approach in [1] does not give continuity guarantees as opposed to our approach.
>     3. [1] considers symmetries based on group actions, while we consider a more general setting of canonicalization transformation (or augmentation) that need not be based on groups, which makes our approach much more applicable across different domains.
>     4. The trainable canonicalization method in [1] has a tendency to learn an “average behavior,” not specializing to a specific orientation but rather performing reasonably well across all orientations of the data. This de facto reduces the expressive power of the method. In contrast, our approach leads to a task neural network that specializes on one important orientation for each datapoint.

---

> ### Author Response · Authors · 2025-11-28
>
> (This is comment 2 out of 6.)
>
> * **(cont.) [1] Sékou-Oumar Kaba, Arnab Kumar Mondal, Yan Zhang, Yoshua Bengio, and Siamak Ravanbakhsh. Equivariance with learned canonicalization functions. In International Conference on Machine Learning, pp. 15546–15566. PMLR, 2023.**
>
>     In more details: First, in [1], there is one canonical form for each datapoint and network, while in our approach, each output channel of the network defines a different canonical form of the datapoint. This allows our approach to preserve continuity as opposed to [1], where the continuity of the end-to-end predictor is not studied. One might be tempted to view our prior maximization approach as a special case of [1] with $s$ taken as the task neural network. However, this would require restricting the classification setting to a single class (which is not a practically meaningful scenario) and limiting our framework to the special case where symmetry preserving prior maximization is based on a group action. In contrast, our setting is multiclass, and our "symmetries" are much more general than group actions. Moreover, with our general formulation of the method, we get guarantees for continuity, while [1] did not address the continuity of the predicting function. Their universal approximation result allows approximating continuous invariant functions by potentially discontinuous functions, whereas our universal approximation results approximate continuous functions by continuous functions.  Even if one chooses to interpret (a restricted version of) our prior maximization method as a special case of [1], this specific case is not proposed or studied in [1], so we believe our method remains novel. More broadly, our general setting of adaptive canonicalization, which reveals which properties are sufficient for continuous canonicalization, differs from the approach and analysis in [1] and is not encompassed by it.
>
>     Another difference in [1] is that when training is initialized, the canonicalizing energy $s$ is random and independent of the task network. This leads each datapoint to be randomly transformed, so the task neural network initially has to perform well at all orientations of the data. This can lead the task network to ultimately learn an "average behavior," not specializing in any special orientation but rather performing reasonably well on all orientations of the data. In other words, the limited set of trainable parameters has to simultaneously specialize in many orientations, which reduces the network’s expressive power. In contrast, in our prior maximization approach, from the beginning of training,  the network only pays a price for not performing well on the single best orientation per datapoint (on which the network performs the best). This encourages the network to specialize on one canonical orientation per datapoint, and not learn an average behavior. Hence, in our approach, all trainable parameters of the task network can focus on performing well only on the sole canonical orientation of each datapoint.
>
> * **[2] Arnab Kumar Mondal, Siba Smarak Panigrahi, Oumar Kaba, Sai Rajeswar Mudumba, and Siamak Ravanbakhsh. Equivariant adaptation of large pretrained models. Advances in Neural Information Processing Systems, 36:50293–50309, 2023.**\
> **[3] Johann Schmidt and Sebastian Stober. Robust canonicalization through bootstrapped data re-alignment. arXiv preprint arXiv:2510.08178, 2025.**
>
>
>     We add a comparison to the above papers in the Appendix A.1.2.
>
>     As discussed above, in the energy-based canonicalization framework of [1], the canonicalizing energy $s$ is random at initialization, which can lead the task network to learn an average behavior, not specialized in one orientation for each datapoint.  To address this effect, [2]  biases the canonical transformation of each datapoint to be close to the identity, assuming that the datapoints in the training set already have a small orientation variance. On top of that, [3] iteratively reduces the orientation variance of the training set by iteratively reorienting datapoints that lead to a large loss. We note that these approaches are rather different from our prior maximization method, and they do not try to address the continuity problem in canonicalization.

---

> ### Author Response · Authors · 2025-11-28
>
> (This is comment 3 out of 6.)
>
> * **[4] Zakhar Shumaylov, Peter Zaika, James Rowbottom, Ferdia Sherry, Melanie Weber, and Carola-Bibiane Schönlieb. Lie algebra canonicalization: Equivariant neural operators under arbitrary lie groups. In The Thirteenth International Conference on Learning Representations, 2025.**
>
>      We add a comparison to the above papers in the Appendix A.1.3.
>
>     The energy-based canonicalization [1] was further explored and extended in [4] on symmetries defined by general Lie group actions. Similarly to our work, [4] also discusses continuity preservation, but their approach is different from ours. In the work of [4], they define the notion of weighted canonicalization, which is a similar concept to the weighted frame introduced by Dym et al., (2024). Here, to each datapoint there is an assigned probability measure over the orbit of the datapoint. Namely, the distribution is over the space of data instead of over the group like in the weighted frames of Dym et al., (2024). In contrast, in our setting, each datapoints has a single canonical form, which is more efficient. With respect to energy minimization, this approach is not very different from [1]. The main difference is in the minimization algorithm, which minimizes over the Lie algebra instead of the Lie group. It is important to note that their work does not train the energy end-to-end with the neural network. Hence, in their work, the canonicalization depends on the whole training set, but not on the task network, which is quite different from our approach. Moreover, in their setup, one has to learn an approximation of the data distribution, which is invariant to the group action. This is a highly nontrivial approach to implement. In contrast, our prior maximization is simple and direct.
>
>
> * **[5] Utkarsh Singhal, Ryan Feng, Stella X. Yu, and Atul Prakash. Test-time canonicalization by foundation models for robust perception. In Forty-second International Conference on Machine Learning, 2025.**\
> **[6] Johann Schmidt and Sebastian Stober. Tilt your head: Activating the hidden spatial-invariance of classifiers. In Forty-first International Conference on Machine Learning, 2024.**
>
>     We add a comparison to the above papers in the Appendix A.1.4.
>
>     Another recent extension of canonicalization, [5], explores a set of transformations at test time and uses the scoring functions of large pre-trained foundation models like CLIP or SAM to select the most "canonical" representation upon which downstream inference is performed. Note that the work in [5] does not involve network retraining, and uses the foundation models as is.  While it achieves strong empirical performance, its canonicalization mapping is not guaranteed to be continuous and continuity is not discussed.  Therefore, small input changes may cause abrupt switches in the selected canonical view.
>
>
>     A related line of work is inverse transformation search [6], which also performs test-time optimization over transformations to exploit invariances. Their method focuses on the standard action of the special linear group on images (rotations, scalings, and shear transformations), i.e., purely group-based symmetries, and similar to [5], does not address continuity. In addition, their method does not train the model simultaneously with the canonicalization. While their energy-induced confidence is similar to our prior-maximization formulation in the classification setting, it does not lead to a continuity guarantee. In contrast, our work rigorously develops sufficient conditions on the canonicalizer for the canonicalized network to be continuous and have a universal approximation property.
>
> We thank the reviewer again for motivating us to broaden and refine our literature review. This led us to clarify more carefully how our general adaptive canonicalization framework and the accompanying continuity and universal approximation guarantees help distinguish our contribution and clarify its position in the literature.

---

> ### Author Response · Authors · 2025-11-28
>
> (This is comment 4 out of 6.)
>
> > W2: The framework is developed with a high degree of generality, which does not refer to group actions until section 2.4. However, in the end the only applications considered are really only group actions.
>
> **Scope of the framework beyond group actions**
>
> We respectfully disagree. Only the point clouds application is based on the standard equivariance setting via group representations. However, the graph neural network application is not based on a vanilla equivariance setting. Here, $\kappa_u$ does not even map the data space $G$ into itself, so it is not a group action. Moreover, the dataspace $G$ is not even a vector space. The “symmetry” $\kappa_u$ is noninvertible and maps graphs to spectral coefficients in some predefined bands.
>
> We note that we developed our theory with this level of generality as this is required in our applications. We put a lot of thought into how to develop a general enough theory that would encompass all of our example applications, while being the least general possible. We also gave an example in the appendix: *Discrete to Discrete Images with Rotations and Other Image Transformations*. The rotation symmetries here are also not based on a group action, as $\kappa_{u’}(\kappa_u(g))\neq \kappa_{u’u}(g)$.
>
> Still, there is some component related to a group (though not an action) in these examples. In the revised paper we now give a new example of prior maximization with no group whatsoever in the loop. Here, the data are images, and the transformation family are truncations.  Namely, the adaptive canonicalization finds the truncated image which maximizes each class probability. Truncation is not a group representation (it is not even invertible), and not based on any group. The application is presented in Appendix E.5 and Appendix G.6 in the revised paper.
>
> >   I feel like the analysis should either be simplified to consider the specific case of group actions (with possibly the more general version in appendix) or applications outside group actions should be examined. In the current state, the amount of formalism seems overkill given the problem considered.
>
> **On the level of generality and formalism**
>
> We respectfully disagree. While indeed including additional assumption can often simplify constructions and proofs, in our paper it will have the opposite effect.
>
> In more detail: even ignoring the fact that some of our applications are not based on the standard group action setting, we strongly believe that developing theory with less assumptions is better for clarity and for future discovery.  We believe that in order to understand a property (e.g. continuity of adaptive canonicalization) one should understand which minimal set of assumptions are required to logically imply the property. Adding any additional irrelevant assumptions, e.g. on group action, will just hide the true reason behind the truth of the property. Irrelevant assumptions just increase the clutter, can lead to misconceptions and hinder future discoveries.
>
> One could justify adding additional assumptions if they make the proofs simpler and clearer. However, in our case, the proofs are already very direct and short, and additional irrelevant assumptions would just make them longer.  Currently, all proofs are just a few lines long. We don’t believe they can be simplified much further.
>
> Note that the other reviewers state that the generality of the theory is a positive aspect of the paper. Hence, since the opinion that we should add additional assumptions to the theory is a minority opinion among the reviewers, and since this is also not the authors’ opinion, we opt to keep the theory general.

---

> ### Author Response · Authors · 2025-11-28
>
> (This is comment 5 out of 6.)
>
> >W3: I think the construction for continuous canonicalization introduced in 2.3 (proved in theorem 9) is closely related to the weighted frames of Dym et al. 2024. Can the authors comment on that and discuss connections in that section of the paper?
> > - If the proposed canonicalization scheme is not a type of weighted frame, it would be great if the authors could better highlight the core ingredient that makes the scheme continuous.
>
> **Detailed comparison with weighted frame averaging**
>
> We thank the reviewer for encouraging us to compare our method with weighted frame averaging (Dym et al., 2024). In the revised manuscript, we now include a brief comparison in the main paper (p.3, l. 137) and provide a more detailed discussion in the Appendix A.2, which we also present below.
>
> Dym et al. (2024) propose a very different approach to addressing the continuity of the end-to-end predicting function in frame averaging methods. They first prove that in many well-known cases, continuous canonicalization is impossible.  This does not contradict our work, as in Dym et al. (2024) the canonicalization is a function solely of the datapoint, and not the task network. Then, they define a variant of frame averaging, called weighted frame averaging, in which to each datapoint there is an associated probability distribution over the group, and the frame averaging is performed with respect to this measure.  This construction yields continuity guarantees.  However, their focus is fundamentally different from ours: they study frame averaging while we focus on canonicalization. Moreover, the weighted frame averaging is a function only of the data, not the neural network, as opposed to our method.   We next compare our framework to weighted frame averaging in more detail.
>
> - In our method, data need not come from a vector space of a fixed dimension (see e.g., the application for graphs). In contrast, weighted frame averaging requires working with data that comes from a vector space of a fixed dimension.
> - In our work, symmetries need not be based on group representations. Our notion of symmetry is called a transformation family, and our "symmetries" are not required to be based on a group action or even to be invertible. See, for example, the image truncation transformation in Appendix E.5. On the other hand, the symmetries in weighted frame averaging are required to be representations of compact groups.
> - In the framework of weighted frame averaging, the requirement that the weighted frame is robust is sufficient for continuity of the canonicalized function. However, this requirement is quite strong, and constructing robust frames could be challenging. In contrast, we achieve continuity of the canonicalized function even though the maximizer in prior maximization need not be continuous in any sense (as a function of the datapoint). The maximizer need not even be uniquely defined. Hence, practitioners can use our method out of the box on new domains with new symmetries without having to prove any nontrivial mathematical results.  In weighted frame averaging, employing the method in a new setting typically requires carefully constructing a problem-specific weighted frame and proving that it is robust. It often involves nontrivial mathematical work and there is no general recipe for doing so. It requires mathematical ingenuity.  In our case, once an architecture and a symmetry setting are chosen, prior-maximization adaptive canonicalization is straightforward to implement and does not require additional sophisticated proofs from the practitioner. The only assumption that needs to be checked is continuity of the chosen transformations, which is usually easy to verify.
> - Robust weighted frames often require averaging the predicting network over many transformations of the input. In practice, our prior maximization approach performs well with fewer random argmax candidates. For comparison, frame averaging for rotations of $n$ points in a 3D point cloud requires an order of $n^2$ transformations (where $n=1024$ in the ModelNet40  dataset for example), while, in practice, prior maximization performs well with a total of 50 random transformations.
>
> These differences imply that our framework is more flexible in terms of the data types and symmetry structures it can accommodate, while also imposing a lighter mathematical burden on practitioners who wish to apply it. This gives much more freedom for future practitioners, and may lead to a wider adaptation of the method.

---

> ### Author Response · Authors · 2025-11-28
>
> (This is comment 6 out of 6.)
>
> >W4: More generally, I would appreciate if the authors could discuss their theoretical results more and try to distill them into insightful conclusions
>
> **Theoretical insights**
>
> We thank the reviewer for this comment. Our main theoretical contribution is that we can show that continuous prior maximization gives functions continuous in $\mathcal{G}$ (Theorem 9). This is the main distinction between our prior maximization and standard canonicalization. Continuity is important for robustness, since without it, small input changes may cause abrupt switches in the selected canonical view, and hence also in the network's prediction. In addition, the universal approximation of symmetry preserving functions in Theorem 13 demonstrates that one can approximate continuous symmetry preserving functions by continuous canonicalized networks.
>
> ---
>
> >W5: I think section 3 on applications should be significantly rewritten to improve ease of understanding and reduce the amount of mathematical formalism. In its current state, this section that introduces applications will be quite difficult to understand for practitioners.
>
> **Revised application appendix for practitioners**
>
> Thank you for this comment. We agree with the reviewer that presenting the applications accessible to practitioners is important. In response, we overhauled the appendix about the applications. In the revised paper, this appendix is now written as a self-contained tutorial aimed at practitioners (Appendix E). Due to space limitations, we are unable to include such a detailed section in the main paper, and therefore refer the readers to the appendix for the detailed tutorial.
>
> ---
>
> >W6: Typos: line 137: "vanish at infinity is" - > "vanish at infinity if"
>
> **Typo**
>
> Thank you, we fixed it in the revision.
>
> ---
>
> >Q1: An important part of the implementation that was not clear to me is how the optimization is performed for each group. Solving these optimization problems is expected to be costly in general and to introduce significant overhead. This should be discussed more transparently
>
> **Approximate maximization**
>
> We thank the reviewer for raising this important point. This was indeed only explained in the appendix of the original paper. In the revised manuscript (p.9, l.461), we now put a description of the maximization algorithm. There, we explain that we approximate the prior maximization by sampling a finite set of transformations from a probability measure on the transformation space, evaluating the prior for all candidates, and retaining the transformation that yields the largest prior value with the one-vs-rest classification objective. We then refine the selected candidate locally by running a few steps of gradient descent from the best sampled transformation. In Appendix E.2 of the revision, we further elaborate on the details of this random maximization procedure. In the same appendix,  we also added an approximation analysis of our random maximization method. Our analysis is based on the tail bounds developed by [a], which address approximating a maximum over a probability space by a maximum over random i.i.d. samples therein. These results apply directly to our setting and give a guarantee that our random maximization method approximates the true maximum. In addition, at the end of Appendix E.2 in the revision, we note that when the prior maximization has error $e$ (which is a random variable), this leads to an additive term $e$ in the universal approximation. We thank the reviewer again for this helpful comment.
>
>
> [a] Matthieu Cordonnier, Nicolas Keriven, Nicolas Tremblay, and Samuel Vaiter. Convergence of message-passing graph neural networks with generic aggregation on large random graphs. Journal of Machine Learning Research, 25(406):1–49, 2024.
>
> ---
>
> >Q2: Likewise, I didn't understand which priors are maximized in practice. Can the authors elaborate and clarify that in the main text.
>
>
> **Used prior**
>
> We thank the reviewer for commenting on this point. In our classification experiment, the prior is applied to the scalar class score (i.e., the one-vs-rest predicted probability), and we set $h_d(x)=x$, so that transformations which increase this probability are preferred during prior maximization. Extending our framework to regression and other tasks will require designing task-appropriate priors and additional formalism, which we view as an important direction for future work. We currently develop such analysis for a future publication.
>
> ---
>
> We thank the reviewer once again for their insightful and constructive comments, which helped us improve our paper, and hope to discuss any ongoing suggestions further.

---

### Official Review · Reviewer_KLEU · 2025-11-04

**Soundness:** 3
**Presentation:** 4
**Contribution:** 3
**Rating:** 8
**Confidence:** 3

**Summary:**

The paper introduces adaptive canonicalization, where the canonical map depends on both the function/network and the input: a family $\rho_f : G \to K$ such that models have the form $g \mapsto f(\rho_f(g))$. Under an equicontinuity condition on ${f \circ \rho_f(g)}$, the authors prove a universal approximation theorem: if a base class $\mathcal N$ is universal on $C_0(K,\mathbb R^D)$, then ${\theta \circ \rho_\theta : \theta \in \mathcal N}$ is universal for adaptive canonicalized functions. They instantiate this via prior maximization, where a transformation family $\kappa_u$ and monotone priors $H=(h_d)$ are used to pick a canonical representative by maximizing $h_d(f_d(\kappa_u(g)))$ over $u$, making $\rho_f$ depend on $f$. They show prior maximization yields continuous, symmetry-preserving models and, under a symmetry-preserving construction, that $f \circ \rho_f$ can represent all continuous symmetry-preserving functions, again with universal approximation guarantees. Two applications are given: (i) A-NLSF spectral GNNs that resolve eigenbasis ambiguities and allow anisotropic filters, and (ii) adaptive canonicalization for point clouds (AC-PointNet / AC-DGCNN) handling SO(3) while retaining directional information. Experiments on graph benchmarks (TUDataset, OGB) and ModelNet40 show consistent improvements over data augmentation, standard canonicalization, and equivariant architectures.

**Strengths:**

1. **Elegant and general theory.** Adaptive canonicalization and prior maximization are simple to state yet powerful, giving continuity and universal approximation while avoiding classical discontinuity pitfalls.

2. **Strong link to symmetry theory.** The symmetry-preserving construction and characterization of all continuous symmetry-preserving functions via $f\circ\rho_f$ provide a satisfying representation-theoretic story.

3. **Practical instantiations.** A-NLSF neatly resolves spectral eigenbasis ambiguities and yields anisotropic filters; AC-PointNet/AC-DGCNN give rotation-invariant yet directionally aware point-cloud models, with solid improvements over data augmentation, standard canonicalization, and VN-based equivariants.

4. **Good positioning in current literature.** The work is clearly framed against discontinuity results for canonicalization and complements other directions like probabilistic frames and probabilistic symmetry breaking.

**Weaknesses:**

1. **Approximate maximization not analyzed.** The theory assumes exact $\arg\max_u$ for prior maximization; in practice one uses discretization or iterative search. The impact of $\delta$-suboptimal maximizers on continuity and approximation guarantees is not quantified.

2. **Focus on classification tasks.** The prior design and applications are tailored to one-vs-rest classification; how to extend to regression or more structured outputs is left open.

3. **Limited variety of groups in experiments.** The empirical section covers spectral band symmetries and SO(3) for point clouds; it would be interesting to see at least one example with more complex/non-compact groups (e.g., SE(3) in continuous domains), though this is not required for acceptance.

**Questions:**

1. In practice you maximize $h_d(f_d(\kappa_u(g)))$ approximately. If the chosen $u$ is within $\delta$ of the true maximum, can you bound the resulting deviation in $f\circ\rho_f$ or in the universal approximation error (e.g., as an additive $\mathcal O(\delta)$ term)?

2. What specific $h_d$ do you use (logit, probability, margin)? Do different monotone choices substantially change performance or optimization stability, and do any non-monotone choices appear useful despite not fitting the theory?

3. Many applications involve non-compact groups (e.g., SE(3)). Under what additional conditions (e.g., priors that decay at infinity, restricted parameter ranges) do you expect the prior maximization construction to remain continuous and symmetry preserving?

4. Since $\rho_f$ depends on the evolving network $f$, did you observe any instabilities (e.g., rapid switching of canonical representatives) during training? Have you tried freezing $\rho_f$ after some epochs and training only the classifier/regressor, and if so, how does that affect accuracy?

5. Do you foresee obstacles to applying adaptive canonicalization to other domains (e.g., SE(2) image rotations/scalings, sequence symmetries), or is it mostly an implementation effort at that point?

---

> ### Author Response · Authors · 2025-11-28
>
> (This is comment 1 out of 2.)
>
> We thank the reviewer for their valuable comments and for recognizing the generality of our method as well as the importance of designing the canonicalization method with continuity properties. In terms of the weaknesses and questions:
>
> >W1: Approximate maximization not analyzed. The theory assumes exact $\arg\max_u$ for prior maximization; in practice one uses discretization or iterative search. The impact of $\delta$-suboptimal maximizers on continuity and approximation guarantees is not quantified.
>
> >Q1: In practice you maximize $h_d(f_d(\kappa_u(g)))$ approximately. If the chosen $u$ is within $\delta$ of the true maximum, can you bound the resulting deviation in $f \circ \rho_f$  or in the universal approximation error (e.g., as an additive $\mathcal{O}(\delta)$ term)?
>
>
>
> **Approximate maximization**
>
>
> We thank the reviewer for this insightful comment. We added to the revised paper an approximation analysis of our random maximization method (Appendix E.2). Here we briefly summarize the main idea. Our analysis builds on the tail bounds developed by [a], which address approximating a maximum over a probability space by a maximum over random i.i.d. samples therein. These results apply directly to our setting and gives a guarantee that our random maximization method approximates the true maximum. In addition, at the end of Appendix E.2 in the revision, we note that when the prior maximization has error $e$ (which is a random variable), this leads to an additive term $e$ in universal approximation.
>
> [a] Matthieu Cordonnier, Nicolas Keriven, Nicolas Tremblay, and Samuel Vaiter. Convergence of message-passing graph neural networks with generic aggregation on large random graphs. Journal of Machine Learning Research, 25(406):1–49, 2024.
>
> ---
>
> >W2: Focus on classification tasks. The prior design and applications are tailored to one-vs-rest classification; how to extend to regression or more structured outputs is left open.
>
> **Extension beyond classification tasks**
>
> We thank the reviewer for this helpful comment and agree that extending our framework beyond classification is a valuable direction. Our current formulation is indeed tailored to classification, and extending the continuous properties to regression or more structured outputs is not trivial. It requires additional theoretical analysis to develop continuous and symmetry preserving models in these settings, which we view as important and promising future work. We are actively working on this extension and plan to address it in a follow-up work. As a first step beyond classification tasks, we included new results on point cloud segmentation in Appendix G.10 in the revised paper, illustrating that our adaptive canonicalization framework has potential beyond classification. We plan to investigate the continuity properties of such adaptive canonicalization methods in future work.
>
> ---
>
> >Q4: Since $\rho_f$ depends on the evolving network $f$, did you observe any instabilities (e.g., rapid switching of canonical representatives) during training? Have you tried freezing $\rho_f$ after some epochs and training only the classifier/regressor, and if so, how does that affect accuracy?
>
> **On training stability**
>
> Thank you for the thoughtful question. In the revised manuscript, we now include an empirical analysis of the stability of the adaptive canonicalization in Appendix G.8. Specifically, in the point cloud classification experiment, we fix a subset of 1500 randomly chosen training examples. At each epoch, we measure the mean geodesic distance on SO(3) between consecutive epochs. Figure 3 reports the mean geodesic distance between epochs. We observe that this distance decreases during the training and then remains stable, indicating that the canonical representatives stabilize with no rapid switching.
>
>
> Regarding the suggestion of freezing $\rho_f$, we did not explore this variant in our experiment. In our view, freezing $\rho_f$ would introduce additional complications. Specifically, if one were to freeze an intermediate $f_0$ for the maximization module while continuing to train $f$, then at test time one would need to save in memory the partially trained network $f_0$ for the maximization $\rho_{f_0}$ and use the fully trained network $f$ for predicting. For these reasons, we chose not to consider this variant in our experiments. We hope this clarifies our rationale and thank the reviewer again for the suggestion.

---

> ### Author Response · Authors · 2025-11-28
>
> (This is comment 2 out of 2.)
>
>
>
> >W3: Limited variety of groups in experiments. The empirical section covers spectral band symmetries and SO(3) for point clouds; it would be interesting to see at least one example with more complex/non-compact groups (e.g., SE(3) in continuous domains), though this is not required for acceptance.
>
> >Q5: Do you foresee obstacles to applying adaptive canonicalization to other domains (e.g., SE(2) image rotations/scalings, sequence symmetries), or is it mostly an implementation effort at that point?
>
> >Q3: Many applications involve non-compact groups (e.g., SE(3)). Under what additional conditions (e.g., priors that decay at infinity, restricted parameter ranges) do you expect the prior maximization construction to remain continuous and symmetry preserving?
>
> **Generality beyond group symmetries**
>
> We would like to point out that our framework is not restricted to symmetries defined via group actions, and allows working with more general augmentations for transforming datapoints.   Specifically, our notion of symmetry is a transformation family, and our "symmetries" are not even required to be invertible or based on a group action. Moreover, the "symmetry space" $\mathcal{U}$ is not required to be compact in our analysis, and we don't assume that $\mathcal{U}$ is compact in our theorems.  For example, $\mathcal{U}$ is not compact in our graph neural network application, as it is a union of infinitely many galaxies.
>
> To illustrate that compactness is not required in our theory, we added a new comment to Appendix C, about the applicability of our theory for noncompact groups (p. 26 , l. 1383): *"The above examples can be naturally extended to additional image transformation, like translations and dilations. Notably, translations and dilations do not form a compact group, but still they satisfy the conditions in our theory, which requires no compactness assumptions."*
>
> To further illustrate that the symmetry need not be based on a group, we added a new application to the revised paper. Specifically, in Appendix E.5 we consider an image truncation transformation as an additional application of our adaptive canonicalization. The idea is that many image classes are invariant under removing uninformative regions: as long as the object of interest remains in the field of view, the class label should not change. We consider, in this case, the truncation as our "symmetry" and we apply prior maximization over this family of transformations to select a canonical truncation. Note that truncation is not a group actions, and it is not even invertible.
>
> To demonstrate the applicability of this setup, we added an experiment on truncation canonicalization with a pretrained classifier in Appendix G.6 in the revised manuscript. Specifically, we take a ResNet-18 pretrained on ImageNet, freeze the backbone, and train only a linear classifier as the last layer using the training set of a new dataset (i.e., CIFAR-10 or CIFAR-100). The resulting classification performance is reported in the Table 11 in the revision. We see that truncation-based prior maximization improves classification performance over the standard vanilla baseline.  This experiment also demonstrates that our adaptive canonicalization can be applied to off-the-shelf pretrained models.
>
> We note that our approach is also applicable to SE(2) and SE(3). While a detailed study of these cases would be an interesting direction for future work, we do not foresee fundamental obstacles to applying adaptive canonicalization in these settings.
>
>
> ---
>
> >Q2: What specific $h_d$ do you use (logit, probability, margin)? Do different monotone choices substantially change performance or optimization stability, and do any non-monotone choices appear useful despite not fitting the theory?
>
> **Used prior**
>
> We thank the reviewer for commenting on this point. In our classification experiment, the prior is applied to the scalar class score (i.e., the one-vs-rest predicted probability), and we set $h_d(x)=x$, so that transformations which increase this probability are preferred during prior maximization. Extending our framework to regression and other tasks will require designing task-appropriate priors and additional formalism, which we view as an important direction for future work and are currently investigating for a future paper.
>
>
> ---
>
> We thank the reviewer once again for their insightful comments, which helped us improve our paper, and hope to discuss any ongoing suggestions further.

---

### Author Response · Authors · 2025-11-30
**Summary for Area Chairs under the updated review process**

We thank the new Area Chairs for taking on the additional workload of our submission under the updated review process, and we also thank the reviewers for their valuable feedback and constructive comments, which helped us improve the paper. Below we briefly summarize our contributions and the main revisions.


Canonicalization is a widely used strategy in equivariant machine learning, enforcing symmetry in neural networks by mapping each input to a standard form. Yet, it often introduces discontinuities that can affect stability of the model. To address this, we introduce adaptive canonicalization, a general framework in which the canonicalization depends both on the input and the network. Our method yields continuous and symmetry-respecting models that admit universal approximation properties.


While continuity had previously been developed for frame averaging in equivariant machine learning, we develop the continuity properties for canonicalization, which is a more efficient approach and leads to more expressive models. Importantly, our framework goes beyond the standard group-action setting and covers more general symmetry transformations. The resulting theory is direct and is formulated so that practitioners can apply it in a variety of settings without a heavy theoretical overhead.

In the revised version, we have:
- Clarified our novel contributions and positioned them more explicitly within the existing literature. In particular, we added several missing references that were initially omitted and explained in detail why the proposed results remain genuinely novel despite this broader context.
- Expanded the experimental results with additional studies, including new ShapeNet part segmentation experiments and a truncation-based canonicalization experiment, further illustrating the practical impact and generality of our framework.

We hope that this summary, together with the revised manuscript and our detailed response to reviewers, provides a clear picture of the contribution of our paper.

We would like to thank the reviewing team and the organizing committee again for their time and effort under the current circumstances.

---

### Meta-Review · Area_Chair_5gVd · 2026-01-05

**Summary:**

The paper has received mostly positive reviews, with all reviewers initially recommending acceptance or strong acceptance, except for Reviewer dHXs, who recommended weak rejection. Based on the authors’ rebuttal, I conclude that the concerns raised by Reviewer dHXs have been satisfactorily addressed. Further details are provided below. I therefore recommend acceptance of the paper.

**Reviewer Concerns:**

Reviewer dHXs: I believe that the presentation-related concerns raised by Reviewer dHXs have been adequately addressed. In addition, the authors provided a thorough discussion of related work and clearly articulated the novelty of their proposed method. I also agree with the authors’ rebuttal regarding the generality of the theory presented in the paper, a point that was further supported by the positive assessments of other reviewers, including Reviewer KLEU, Reviewer 6Rf6, and Reviewer 48pC. Overall, I am convinced that all concerns have been satisfactorily addressed.

**Reviewer Scores:**

Based on the authors’ rebuttal, I believe that Reviewer dHXs would have raised their score to recommend acceptance. Since the concerns raised by the other reviewers were largely addressed, I also believe that they would have maintained their positive assessments.

---

### Decision · Program_Chairs · 2026-01-26

Accept (Poster)